# Zeroth-Order Methods for Nondifferentiable, Nonconvex, and Hierarchical Federated Optimization

**Yuyang Qiu**
Dept. of Industrial and Systems Engg.
Rutgers University
yuyang.qiu@rutgers.edu

**Uday V. Shanbhag**
Dept. of Industrial and Manufacturing Engg.
Pennsylvania State University
udaybag@psu.edu

**Farzad Yousefian**
Dept. of Industrial and Systems Engg.
Rutgers University
farzad.yousefian@rutgers.edu

## Abstract

Federated learning (FL) has emerged as an enabling framework for communication-efficient decentralized training. We study three broadly applicable problem classes in FL: (i) Nondifferentiable nonconvex federated optimization; (ii) Federated bilevel optimization; (iii) Federated minimax problems. Notably, in an implicit sense, both (ii) and (iii) are instances of (i). However, the hierarchical problems in (ii) and (iii) are often complicated by the absence of a closed-form expression for the implicit objective function. Unfortunately, research on these problems has been limited and afflicted by reliance on strong assumptions, including the need for differentiability and L-smoothness of the implicit function. We address this shortcoming by making the following contributions. In (i), by leveraging convolution-based smoothing and Clarke's subdifferential calculus, we devise a randomized smoothing-enabled zeroth-order FL method and derive communication and iteration complexity guarantees for computing an approximate Clarke stationary point. To contend with (ii) and (iii), we devise a unified randomized implicit zeroth-order FL framework, equipped with explicit communication and iteration complexities. Importantly, our method utilizes delays during local steps to skip making calls to the inexact lower-level FL oracle. This results in significant reduction in communication overhead when addressing hierarchical problems. We empirically validate the theory on nonsmooth and hierarchical ML problems.

## 1 Introduction

Federated learning (FL) has recently emerged as a promising enabling framework for learning predictive models from a multitude of distributed, privacy-sensitive, and possibly, heterogeneous datasets. This is accomplished through the use of efficiently devised periodic communications between a central server and a collection of clients. The FL algorithmic framework allows for addressing several key obstacles in the development and implementation of standard machine learning methods in a distributed and parallel manner. For instance, the conventional parallel stochastic gradient descent (SGD) method requires the exchange of information among the computing nodes at every single time step, resulting in excessive communication overhead. In contrast, FL methods including FedAvg [34] and Local SGD [46] overcome this onerous communication bottleneck by provably attaining the linear speedup of parallel SGD by using a significantly fewer communication rounds [18, 54, 24, 7]. These guarantees have been further complemented by recent efforts [26,

37th Conference on Neural Information Processing Systems (NeurIPS 2023).

38] where the presence of both data heterogeneity (i.e., variability of local datasets) and device heterogeneity (i.e., variability of edge devices in computational power, memory, and bandwidth) have been addressed. Despite recent advances, much needs to be understood about designing communication-efficient decentralized methods for resolving three broadly applicable problem classes, each of which is presented next.

(a) *Nondifferentiable nonconvex locally constrained FL.* Consider the prototypical FL setting:

$$\min_{x} \quad \left\{ f(x) \triangleq \frac{1}{m} \sum_{i=1}^{m} \mathbb{E}_{\xi_i \in \mathcal{D}_i}[\,\tilde{f}_i(x, \xi_i)\,] \mid x \in X \triangleq \bigcap_{i=1}^{m} X_i \right\}, \qquad (\mathbf{FL}_{nn})$$

where $f$ is a nonsmooth nonconvex function and is associated with a group of $m$ clients indexed by $i \in [m] \triangleq \{1, \ldots, m\}$, $\mathcal{D}_i$ denotes the local dataset, $\tilde{f}_i : \mathbb{R}^n \times \mathcal{D}_i \to \mathbb{R}$ is the local loss function, and $X_i \subseteq \mathbb{R}^n$ is an easy-to-project local constraint set. Notably, local datasets may vary across clients, allowing for data heterogeneity. We also consider client-specific local sets to induce personalization.

(b) *Nondifferentiable nonconvex bilevel FL.* Overlaying a bilevel term in ($\mathbf{FL}_{nn}$) leads to

$$\min_{x \in X \triangleq \bigcap_{i=1}^{m} X_i} \left\{ f(x) \mid y(x) \in \arg\min_{y \in \mathbb{R}^{\tilde{n}}} \quad \frac{1}{m} \sum_{i=1}^{m} \mathbb{E}_{\zeta_i \in \tilde{\mathcal{D}}_i}[\tilde{h}_i(x, y, \zeta_i)] \right\}, \qquad (\mathbf{FL}_{bl})$$

where $f(\bullet) \triangleq \frac{1}{m} \sum_{i=1}^{m} \mathbb{E}_{\xi_i \in \mathcal{D}_i}[\,\tilde{f}_i(\bullet, y(\bullet), \xi_i)\,]$ denotes the implicit objective function and $y(\bullet) : \mathbb{R}^n \to \mathbb{R}^{\tilde{n}}$ is a single-valued map returning the unique solution to the lower-level problem at $x$.

(c) *Nondifferentiable nonconvex minimax FL.* Finally, we consider the minimax setting, defined as

$$\min_{x \in X \triangleq \bigcap_{i=1}^{m} X_i} \max_{y \in \mathbb{R}^{\tilde{n}}} \quad \frac{1}{m} \sum_{i=1}^{m} \mathbb{E}_{\xi_i \in \mathcal{D}_i}[\,\tilde{f}_i(x, y, \xi_i)\,]. \qquad (\mathbf{FL}_{mm})$$

where we assume that $y(x) \in \arg\max_{y \in \mathbb{R}^{\tilde{n}}} \frac{1}{m} \sum_{i=1}^{m} \mathbb{E}_{\zeta_i \in \tilde{\mathcal{D}}_i}[\tilde{f}_i(x, y, \xi_i)]$ is unique for all $x$. Let $f(\bullet) \triangleq \frac{1}{m} \sum_{i=1}^{m} \mathbb{E}_{\xi_i \in \mathcal{D}_i}[\,\tilde{f}_i(\bullet, y(\bullet), \xi_i)\,]$ denote the implicit objective function. Indeed, problem ($\mathbf{FL}_{bl}$) subsumes this minimax formulation when we choose $\tilde{h}_i := -\tilde{f}_i$ and $\tilde{\mathcal{D}}_i := \mathcal{D}_i$.

Notably, in an implicit sense, both (b) and (c) are instances of problem (a). However, these hierarchical problems are often complicated by the absence of a closed-form expression for the implicit objective, denoted by $f(\bullet)$. Indeed, $f(\bullet)$ is often nonsmooth, nonconvex, and unavailable. As such, the absence of both zeroth and first-order information of $f(\bullet)$ in problems (b) and (c) makes the design and analysis of FL methods for these problems more challenging than that for (a).

**Gaps.** To the best of our knowledge, there are no known efficient FL algorithms that can contend with both **nonsmoothness** and **nonconvexity** in an unstructured sense. Generalizations that can accommodate either a **bilevel** or a **minimax** structure also remain unaddressed in FL.

**Goal.** *To develop a unified FL framework accommodating nondifferentiable nonconvex settings with extensions allowing for bilevel or minimax interactions.* We now describe the proposed framework.

## 1.1 A Smoothed Sampled Zeroth-order Framework

We consider a smoothed framework for contending with constrained, nonsmooth, and nonconvex regimes. Specifically, given that $f$ is an expectation-valued function and $X$ is a closed and convex set, both of which are defined in ($\mathbf{FL}_{nn}$), a smoothed unconstrained approximation is given as follows.

$$\left\{ \begin{array}{l} \min \frac{1}{m} \sum_{i=1}^{m} \mathbb{E}_{\xi_i}[\,\tilde{f}_i(x, \xi_i)\,] \\ \text{subject to } x \in \cap_{i=1}^{m} X_i. \end{array} \right\} \equiv \left\{ \min \frac{1}{m} \sum_{i=1}^{m} [\,\mathbb{E}_{\xi_i}[\,\tilde{f}_i(x, \xi_i)\,] + \underbrace{\mathbb{I}_{X_i}(x)}_{\text{Indicator function}} \,] \right\}$$

$$\stackrel{\text{Smoothing}}{\approx} \left\{ \min \frac{1}{m} \sum_{i=1}^{m} \underbrace{\mathbb{E}_{u_i \in \mathbb{B}}[\,\mathbb{E}_{\xi_i}[\,\tilde{f}_i(x + \eta u_i, \xi_i)\,]]}_{\text{Convolution smoothing}} + \underbrace{\mathbb{I}_{X_i}^{\eta}(x)}_{\text{Moreau smoothing}} \right\}. \qquad (\mathbf{FL}_{nn}^{\eta})$$

If $f$ is as defined in ($\mathbf{FL}_{nn}$) and $d(x) \triangleq \frac{1}{m} \sum_{i=1}^{m} \mathbb{I}_{X_i}(x)$, then $\mathbf{f}$ and its smoothing $\mathbf{f}^{\eta}$ are defined as

$$\mathbf{f}(x) \triangleq f(x) + d(x) \text{ and } \mathbf{f}^{\eta}(x) \triangleq f^{\eta}(x) + d^{\eta}(x), \qquad (1)$$

where $f^{\eta}(x) \triangleq \frac{1}{m} \sum_{i=1}^{m} [\,\mathbb{E}_{u_i \in \mathbb{B}}[\,\mathbb{E}_{\xi_i}[\,\tilde{f}_i(x + \eta u_i, \xi_i)\,]]]$ and $d^{\eta}(x) \triangleq \frac{1}{m} \sum_{i=1}^{m} \mathbb{I}_{X_i}^{\eta}(x)$.

(i) *Clarke-stationarity*. Consider the original problem ($\mathbf{FL}_{nn}$). Under the assumption that the objective of ($\mathbf{FL}_{nn}$) is Lipschitz continuous, then Clarke-stationarity of $x$ w.r.t. ($\mathbf{FL}_{nn}$) requires that $x$ satisfies $0 \in \partial f(x) + \mathcal{N}_{\cap_{i=1}^m X_i}(x)$, where $\partial f(x)$ represents the Clarke generalized gradient [3] of $f$ at $x$. However, a negative result has been provided regarding the efficient computability of an $\epsilon$-stationary solution in nonsmooth nonconvex regimes [56]. Consequently, we focus on the smoothed counterpart ($\mathbf{FL}_{nn}^\eta$), a smooth nonconvex problem. In fact, under suitable conditions, it can be shown that stationary point of ($\mathbf{FL}_{nn}^\eta$) is a $2\eta$-Clarke stationary point of the original problem, i.e.

$$[\, 0 \in \partial \mathbf{f}^\eta(x) \,] \implies [\, 0 \in \partial_{2\eta} \mathbf{f}(x) \,], \qquad (2)$$

where $\partial_{2\eta}\mathbf{f}(x)$ represents the $2\eta$-Clarke generalized gradient of $\mathbf{f}$ at $x$.

(ii) *Meta-scheme for efficient resolution of* ($\mathbf{FL}_{nn}^\eta$). We develop zeroth-order stochastic gradient schemes for resolving ($\mathbf{FL}_{nn}^\eta$). This requires a zeroth-order gradient estimator for $f^\eta(x)$, denoted by $\frac{1}{m}\sum_{i=1}^m g_i^\eta(x,\xi_i,v_i)$ where $v_i \in \eta\mathbb{S}$ for $i \in [m]$ and $\mathbb{S}$ denotes the surface of the unit ball. Note that the Moreau smoothing of the indicator function of $X_i$, denoted by $\mathbb{I}_{X_i}^\eta(x)$, admits a gradient, defined as $\nabla_x \mathbb{I}_{X_i}^\eta(x) = \frac{1}{\eta}(x - \mathcal{P}_{X_i}(x))$, where $\mathcal{P}_{X_i}(x) \triangleq \arg\min_{y \in X_i}\|y - x\|^2$. The resulting *meta-scheme* is defined next.

$$x_{k+1} = x_k - \gamma\left(\frac{1}{m}\sum_{i=1}^m\left(g_i^\eta(x_k,\xi_{i,k},v_{i,k}) + \frac{1}{\eta}(x_k - \mathcal{P}_{X_i}(x_k))\right)\right), \ k \geq 0. \qquad \textbf{(Meta-ZO)}$$

(iii) *Inexact implicit generalizations for* ($\mathbf{FL}_{bl}$) *and* ($\mathbf{FL}_{mm}$). In addressing the bilevel problem, unlike in ($\mathbf{FL}_{nn}$), the clients in ($\mathbf{FL}_{bl}$) may not have access to the exact evaluation of the implicit local objective $\tilde{f}_i(\bullet, y(\bullet), \xi_i)$. This makes a direct extension of FL schemes challenging. This is because the evaluation of the implicit local function necessitates exact resolution of the lower-level problem. We address this challenge by developing *inexact* implicit variants of the zeroth-order scheme, where clients compute only an $\varepsilon$-approximation of $y(x)$, denoted by $y_\varepsilon(x)$ in a federated fashion. This inexact framework, described next, is crucial in addressing hierarchy in bilevel FL formulations. Let $f_i^\eta(\bullet)$ denote the smoothed implicit local function. We estimate $\nabla_x f_i^\eta(x)$ by approximating the expectation by sampling in $(a.1)$, as follows, while in $(a.2)$, we replace $y(x)$ by an inexact form $y_\varepsilon(x)$. This leads to the introduction of $g_i^{\eta,\varepsilon}(x,\xi,v)$ as captured below.

$$\nabla_x f_i^\eta(x) = \underbrace{\mathbb{E}_{\xi_i,v}\left[g_i^\eta(x,\xi_i,v)\right]}_{\text{cannot be tractably evaluated}} \overset{(a.1)}{\approx} \underbrace{g_i^\eta(x,\xi_{i,k},v_{T_r})}_{\text{intractable since } y(x) \text{ is unavailable}} \overset{(a.2)}{\approx} g_i^{\eta,\varepsilon}(x,\xi_{i,k},v_{T_r}),$$

where $k$ is the local time index, $T_r$ is the global time index of communication round $r$ ($k \geq T_r$), and $g_i^{\eta,\varepsilon}(x,\xi,v) \triangleq \frac{n}{\eta}(\tilde{f}_i(x + v, y_\varepsilon(x + v), \xi) - \tilde{f}_i(x, y_\varepsilon(x), \xi))\frac{v}{\|v\|}$ denotes an inexact implicit zeroth-order gradient. Note that at each round of communication at the upper level, $y_\varepsilon(x)$ can be computed using calls to a standard FL method, e.g., FedAvg, in the lower level. Notably, such calls to an FL oracle should be made only at the global step to preserve the communication efficiency of the scheme. It follows that $g_i^{\eta,\varepsilon}(x,\xi_{i,k},v_{T_r}) = \nabla_x f_i^\eta(x) + \tilde{e}_{i,\varepsilon}$ where the approximation error $\tilde{e}_{i,\varepsilon}$ is a possibly biased random variable. This bias can be then controlled by carefully by updating the accuracy level $\varepsilon$ at each communication round, as we will address in this work.

## 1.2 Contributions

Our goal lies in extending (**Meta-ZO**) to federated nonsmooth nonconvex optimization and then provide generalizations to bilevel and minimax regimes. In each instance, we intend to provide iteration and communication-complexity bounds for computing an $\epsilon$-accurate $\eta$-Clarke stationary point of the original problem. Accordingly, we make the following contributions.

(i) *FL for nondifferentiable nonconvex problems*. To address ($\mathbf{FL}_{nn}$) with heterogeneous datasets, we develop a Randomized Zeroth-Order Locally-Projected Federated Averaging method (FedRZO$_{\mathtt{nn}}$). We derive iteration complexity of $\mathcal{O}\left(\frac{1}{m\epsilon^2}\right)$ and communication complexity of $\mathcal{O}\left(m^{3/4}K^{3/4}\right)$ for computing an approximate Clarke stationary point. Such guarantees appear to be new in the context of resolving nondifferentiable nonconvex FL problems, e.g. in training of ReLU neural networks (see Table 2). This is distinct from existing zeroth-order methods, including FedZO [12], that rely on differentiability and $L$-smoothness of the local loss functions.

(ii) *Federated bilevel optimization*. In addressing ($\mathbf{FL}_{bl}$), we develop FedRZO$_{\mathtt{bl}}$, an inexact implicit extension of FedRZO$_{\mathtt{nn}}$. By skipping local calls to the lower-level FL oracle, FedRZO$_{\mathtt{bl}}$ is a

Table 1: Communication complexity for nonsmooth nonconvex, bilevel, and minimax FL problems. $K$ and $\tilde{K}$ denote the maximum number of iteration used in upper level and lower level, respectively.

| heterogeneous upper level (**this work**) | lower level (standard FL schemes are employed) | | total for bilevel and minimax (**this work**) |
|---|---|---|---|
| $\mathcal{O}\left((mK)^{\frac{3}{4}}\right)$ (Prop. 1, Thm. 1) | i.i.d. (Local SGD [26]) | $\mathcal{O}(m)$ | $\mathcal{O}\left(m^{\frac{7}{4}}K^{\frac{3}{4}}\right)$ |
| | i.i.d. (FedAC [54]) | $\mathcal{O}\left(m^{\frac{1}{3}}\right)$ | $\mathcal{O}\left(m^{\frac{13}{12}}K^{\frac{3}{4}}\right)$ |
| | heterogeneous (LFD [20]) | $\mathcal{O}\left(m^{\frac{1}{3}}\tilde{K}^{\frac{1}{3}}\right)$ | $\mathcal{O}\left((mK)^{\frac{11}{12}}\right)$ |

novel communication-efficient FL scheme with single-timescale local steps, resulting in significant reduction in communication overhead. Table 1 summarizes the communication complexity of this scheme. In all cases, we assume heterogeneous data at the upper level. In the lower level, depending on which conventional FL scheme is employed, we obtain the communication complexity accordingly.

(iii) *Federated minimax optimization.* FedRZO$_{\text{bl}}$ can be employed for addressing (**FL**$_{mm}$) where $\tilde{h}_i := -\tilde{f}_i$. As such, the complexity results in (ii) hold for solving (nondifferentiable nonconvex)-(strongly concave) FL minimax problems. Such results are new for this class of FL problems.

**Remark 1.** There has been recent progress in addressing bilevel and minimax problems in FL, including Local SGDA and FedNest [48, 43]. Our work in (ii) and (iii) has two main distinctions with existing FL methods, described as follows. (1) We do not require the differentiability and $L$-smoothness of the implicit objective function. This assumption may fail to hold, e.g., in constrained hierarchical FL problems. (2) The existing FL methods for bilevel and minimax problems assume that the lower-level problem is unconstrained. In fact, even in centralized regimes, addressing hierarchical problems where the lower-level constraints depend on $x$ have remained challenging. For example, consider the problem $\min_{x\in[-1,1]} \max_{y\in[-1,1], \, x+y\leq 0} x^2 + y$ that admits a unique solution $(x^*, y^*) = (0.5, -0.5)$. Now consider a reversal of min and max in this problem, i.e., $\max_{y\in[-1,1]} \min_{x\in[-1,1], \, x+y\leq 0} x^2 + y$, admitting the unique solution $(x^*, y^*) = (-1, 1)$. As a consequence, the well-known primal-dual gradient methods, that have been extensively employed for addressing minimax problems with independent constraint sets, may fail to converge to a saddle-point in minimax problems with coupling constrains. Our proposed algorithmic framework allows for accommodating these challenging problems in FL.

## 2 Related work

Table 2: Comparison of our scheme with other FL schemes for nonconvex settings

| Ref. | Nonconvex | Metric | Rate | Comm. rounds | Assumption |
|---|---|---|---|---|---|
| [53] | Smooth | $\|\nabla_x f(x)\|^2$ | $\mathcal{O}\left(\frac{G^2}{\sqrt{mK}}\right)$ | $\mathcal{O}\left(m^{3/4}K^{3/4}\right)$ | Bounded gradients, $L$-smooth functions |
| [50] | Smooth | $\|\nabla_x f(x)\|^2$ | $\mathcal{O}\left(\frac{1}{\sqrt{mK}}\right)$ | $\mathcal{O}\left(m^{3/2}K^{1/2}\right)$ | $L$-smooth functions |
| [18] | Smooth, PL-cond | $f(x) - f^*$ | $\mathcal{O}\left(\frac{1}{mK}\right)$ | $\mathcal{O}\left(m^{1/3}K^{1/3}\right)$ | L-smooth functions, PL-condition |
| **This work** | Nonsmooth | $\|\nabla_x \mathbf{f}^\eta(x)\|^2$ | $\mathcal{O}\left(\frac{1}{\sqrt{mK}}\right)$ | $\mathcal{O}\left(m^{3/4}K^{3/4}\right)$ | Lipschitz functions |

**(i) Nondifferentiable nonconvex optimization.** Nonsmooth and nonconvex optimization has been studied extensively with convergence guarantees to Clarke-stationary points via gradient sampling [1, 2] and difference-of-convex approaches [5]. Most complexity and rate guarantees necessitate smoothness of the nonconvex term [14, 51, 8, 9, 28] or convexity of the nonsmooth term [13], while only a few results truly consider nonsmooth nonconvex objective function [32, 4, 42]. **(ii) Nondifferentiable nonconvex federated learning.** The research on FL was initially motivated by decentralized neural networks where local functions are nondifferentiable and nonconvex [34]. Nevertheless, theoretical guarantees that emerged after FedAvg required either nonsmooth convex or smooth nonconvex local costs, under either iid [46, 58, 50, 47] or non-iid datasets [30, 26], while provable guarantees for FL methods under nonconvexity [58, 53, 19] require $L$-smoothness of local functions. Unfortunately, these assumptions do not hold either for ReLU neural networks or risk-averse learning and necessitate the use of Clarke calculus [3]. Moreover, existing work on zeroth-order FL methods in convex [31] and nonconvex settings [12] rely on the smoothness properties of the objective function. However, there appear to be no provably convergent FL schemes with complexity guarantees for computing approximate Clarke stationary points of nondifferentiable nonconvex problems. **(iii) Federated bilevel optimization.** Hyperparameter tuning [21] and its federated counterpart [23] is a crucial, and yet, computationally complex integrant of machine learning (ML) pipeline. Bilevel models

where the lower-level is a parameterized training model while the upper-level requires selecting the best configuration for the unknown hyperparameters [15, 22, 49]. Solving such hierarchical problems is challenging because of nondifferentiable nonconvex terms and absence of an analytical form for the implicit objective. These challenges exacerbate the development of provable guarantees for privacy-aware and communication-efficient schemes. **(iv) Federated minimax optimization.** Minimax optimization has assumed relevance in adversarial learning [17, 40, 44] and fairness in ML [57], amongst other efforts. Recently, FL was extended to distributed minimax problems [36, 10, 43], but relatively little exists in nonconvex-strongly concave settings [48, 43].

## 3 A Zeroth-order FL Framework for Nondifferentiable Nonconvex Settings

In this section, we introduce our framework for ($\text{FL}_{nn}$), where we impose the following assumption.

**Assumption 1.** Consider problem ($\text{FL}_{nn}$). The following hold.

(i) The function $f_i$ is Lipschitz continuous with parameter $L_0 > 0$ for all $i \in [m]$.

(ii) For any $i \in [m]$, client $i$ has access to a zeroth-order oracle $\tilde{f}_i(x, \xi_i)$ satisfying the following for every $x$ in an almost-sure sense:

(ii-a) $\mathbb{E}[\tilde{f}_i(x, \xi_i) \mid x] = f_i(x)$; (ii-b) There exists $\nu > 0$ such that $\mathbb{E}[|\tilde{f}_i(x, \xi_i) - f_i(x)|^2 \mid x] \leq \nu^2$.

(iii) The set $X_i$ is nonempty, closed, and convex for all $i \in [m]$. In addition, the following *bounded set-dissimilarity* condition holds for all $x \in \mathbb{R}^n$ and some scalars $B_1$ and $B_2$.

$$\tfrac{1}{m} \sum_{i=1}^m \text{dist}^2(x, X_i) \leq B_1^2 + B_2^2 \left\| x - \tfrac{1}{m} \sum_{i=1}^m \mathcal{P}_{X_i}(x) \right\|^2. \tag{3}$$

We note that the bounded set-dissimilarity condition is naturally analogous to the so-called *bounded gradient-dissimilarity* condition that has been employed in the literature, e.g., in [25]. In particular, when the bounded gradient-dissimilarity condition is stated for the Moreau smoothing of the indicator function of $X_i$, denoted by $\mathbb{I}_{X_i}^\eta(x)$, we reach to (3). Notably, condition (3) holds for the generated iterates by the algorithm when, for example, the iterates remain bounded.

**Nonsmooth unconstrained reformulation.** Consider an unconstrained reformulation of ($\text{FL}_{nn}$) given by $\min_{x \in \mathbb{R}^n} \mathbf{f}(x)$ (see (1)), where the nonsmoothness of $\mathbf{f}$ arises from that of $f$ and the local indicator functions $\mathbb{I}_{X_i}$. The minimization of $\mathbf{f}$ is challenging, as noted by recent findings on nonsmooth analysis where it is shown [56] that for a suitable class of nonsmooth functions, computing an $\epsilon$-stationary point, i.e., a point $\bar{x}$ for which $\text{dist}(0_n, \partial \mathbf{f}(\bar{x})) \leq \epsilon$, is impossible in finite time.

**Approximate Clarke stationarity.** To circumvent this challenge, as a weakening of $\epsilon$-stationarity, a notion of $(\delta, \epsilon)$-stationarity is introduced [56] for a vector $\bar{x}$ when $\text{dist}(0_n, \partial_\delta \mathbf{f}(\bar{x})) \leq \epsilon$, where the set

$$\partial_\delta \mathbf{f}(x) \triangleq \text{conv} \left\{ \zeta : \zeta \in \partial \mathbf{f}(y), \|x - y\| \leq \delta \right\}$$

denotes the $\delta$-Clarke generalized gradient of $\mathbf{f}$ at $x$ [16]; i.e. if $x$ is $(\delta, \epsilon)$-stationary, then there exists a convex combination of gradients in a $\delta$-neighborhood of $x$ that has a norm of at most $\epsilon$ [41].

This discussion naturally leads to the following key question: *Can we devise provably convergent FL methods for computing approximate Clarke stationary points of minimization of $\mathbf{f}$?* The aim of this section is to provide an affirmative answer to this question by proposing a zeroth-order FL method that employs smoothing. To contend with the nonsmoothness, we employ the Moreau-smoothed variant of $\mathbb{I}_X(x)$, where $X \triangleq \bigcap_{i=1}^m X_i$, and a randomized smoothed variant of $f$, as shown next.

**Randomized smoothing of loss function.** To smoothen the loss function $f$, we employ a *randomized smoothing* approach where the smoothing parameter is maintained as sufficiently small. This framework is rooted in the seminal work by Steklov [45], leading to progress in both convex [27, 52, 11] and nonconvex [37] regimes. We consider a smoothing of $f$, given by $f^\eta$ defined as $f^\eta(x) \triangleq \mathbb{E}_{u \in \mathbb{B}}[f(x + \eta u)]$, where $u$ is a random vector in the unit ball $\mathbb{B}$, defined as $\mathbb{B} \triangleq \{u \in \mathbb{R}^n \mid \|u\| \leq 1\}$. Further, we let $\mathbb{S}$ denote the surface of the ball $\mathbb{B}$, i.e., $\mathbb{S} \triangleq \{v \in \mathbb{R}^n \mid \|v\| = 1\}$ and $\eta \mathbb{B}$ and $\eta \mathbb{S}$ denote a ball with radius $\eta$ and its surface, respectively.

**Lemma 1** (Randomized spherical smoothing). Let $h : \mathbb{R}^n \to \mathbb{R}$ be a given continuous function and define $h^\eta(x) \triangleq \mathbb{E}_{u \in \mathbb{B}}[h(x + \eta u)]$. Then, the following hold.

(i) $h^\eta$ is continuously differentiable and $\nabla h^\eta(x) = \left(\frac{n}{\eta}\right) \mathbb{E}_{v \in \eta \mathbb{S}}[h(x+v)\frac{v}{\|v\|}]$ for any $x \in \mathbb{R}^n$.

Suppose $h$ is Lipschitz continuous with parameter $L_0 > 0$. Then, the following statements hold.

(ii) $|h^\eta(x) - h^\eta(y)| \leq L_0 \|x - y\|$ for all $x, y \in \mathbb{R}^n$; (iii) $|h^\eta(x) - h(x)| \leq L_0 \eta$ for all $x \in \mathbb{R}^n$; (iv) $\|\nabla h^\eta(x) - \nabla h^\eta(y)\| \leq \frac{L_0 n}{\eta}\|x - y\|$ for all $x, y \in \mathbb{R}^n$. $\qquad\square$

The discussion leads to the consideration of the following smoothed federated problem.

**Definition 1** (Unconstrained smoothed approximate problem). Given $\eta > 0$, consider an unconstrained smoothed problem given as

$$\min_{x \in \mathbb{R}^n} \mathbf{f}^\eta(x) \left\{ \triangleq \frac{1}{m}\sum_{i=1}^m \mathbf{f}_i^\eta(x) \right\}, \text{ where } \mathbf{f}_i^\eta(x) \triangleq \mathbb{E}_{\xi_i, u_i \in \mathbb{B}}[\tilde{f}_i(x + \eta u_i, \xi_i)] + \frac{1}{2\eta}\mathrm{dist}^2(x, X_i). \tag{4}$$

---

**Algorithm 1** Randomized Zeroth-Order Locally-Projected Federated Averaging (FedRZO$_{\text{nn}}$)

---

1: **input:** Server chooses a random initial point $\hat{x}_0 \in X$, stepsize $\gamma$, smoothing parameter $\eta$, synchronization indices $T_0 := 0$ and $T_r \geq 1$, where $r \geq 1$ is the communication round index
2: **for** $r = 0, 1, \ldots$ **do**
3: $\quad$ Server broadcasts $\hat{x}_r$ to all clients: $x_{i,T_r} := \hat{x}_r, \ \forall i \in [m]$
4: $\quad$ **for** $k = T_r, \ldots, T_{r+1} - 1$ **do in parallel by clients**
5: $\quad\quad$ Client $i$ generates the random replicates $\xi_{i,k} \in \mathcal{D}_i$ and $v_{i,k} \in \eta \mathbb{S}$
6: $\quad\quad g_{i,k}^\eta := \frac{n}{\eta^2}\left(\tilde{f}_i(x_{i,k} + v_{i,k}, \xi_{i,k}) - \tilde{f}_i(x_{i,k}, \xi_{i,k})\right) v_{i,k}$
7: $\quad\quad$ Client $i$ does a local update as $x_{i,k+1} := x_{i,k} - \gamma\left(g_{i,k}^\eta + \frac{1}{\eta}(x_{i,k} - \mathcal{P}_{X_i}(x_{i,k}))\right)$
8: $\quad$ **end for**
9: $\quad$ Server receives $x_{i,T_{r+1}}$ from all clients and aggregates, i.e., $\hat{x}_{r+1} := \frac{1}{m}\sum_{i=1}^m x_{i,T_{r+1}}$
10: **end for**

---

To address (4), we propose FedRZO$_{\text{nn}}$ given by Algorithm 1. Here, client $i$ employs a zeroth-order stochastic gradient of the form $g_{i,k}^\eta \triangleq \frac{n}{\eta^2}\left(\tilde{f}_i(x_{i,k} + v_{i,k}, \xi_{i,k}) - \tilde{f}_i(x_{i,k}, \xi_{i,k})\right) v_{i,k}$, augmented by the gradient of the Moreau smoothed function. The random sample $v_{i,k} \in \eta \mathbb{S}$ is locally generated by each client $i$, allowing for randomized smoothing. This is indeed in view of Lemma 1 (i) that facilitates the development of a randomized zeroth-order gradient.

We define $\bar{x}_k \triangleq \frac{\sum_{i=1}^m x_{i,k}}{m}$ as an auxiliary sequence to denote the averaged iterates of the clients.

**Definition 2.** Consider Algorithm 1. Let $H > 0$ denote an upper bound on the number of local steps per round, i.e., $H \geq \max_{r=0,1,\ldots}|T_{r+1} - T_r|$. Throughout, we assume that $H$ is finite.

**Proposition 1.** Consider Algorithm 1. Let Assumption 1 hold.

(i) **[Error bound]** Suppose $\gamma \leq \min\left\{\frac{\eta}{4L_0 n}, \frac{1}{4H}, \frac{\eta}{12\sqrt{3}B_2(L_0 n + 1)H}\right\}$. Let $k^*$ denote an integer drawn uniformly at random from $\{0, \ldots, K\}$ and $\mathbf{f}^{\eta,*} \triangleq \inf_x \mathbf{f}^\eta(x)$. Then,

$$\mathbb{E}\left[\|\nabla \mathbf{f}^\eta(\bar{x}_{k^*})\|^2\right] \leq \frac{8(\mathbb{E}[\mathbf{f}^\eta(\bar{x}_0)] - \mathbf{f}^{\eta,*})}{\gamma(K+1)} + \frac{12\gamma L_0 n^3}{\eta m}\left(\frac{2\nu^2}{\eta^2} + L_0^2\right)$$
$$+ \frac{36H^2\gamma^2(L_0 n + 1)^2}{\eta^2}\left(\frac{6n^2\nu^2 + 2B_1^2}{\eta^2} + (3 + 4B_2^2)L_0^2 n^2\right).$$

(ii) **[Iteration complexity]** Let $\gamma := \sqrt{\frac{m}{K}}$ and $H := \left\lceil \sqrt[4]{\frac{K}{m^3}} \right\rceil$ where $\eta > 0$. Let $\epsilon > 0$ be an arbitrary scalar and $K_\epsilon$ denote the number of iterations such that $\mathbb{E}\left[\|\nabla f^\eta(\bar{x}_{k^*})\|^2\right] \leq \epsilon$. Then, the iteration complexity is $K_\epsilon := \mathcal{O}\left(\left(\frac{L_0 n^3 \nu^2}{\eta^3} + \frac{L_0^3 n^3}{\eta} + \frac{L_0^2 n^4 \nu^2}{\eta^4} + \frac{L_0^2 n^2 B_1^2}{\eta^4} + \frac{B_2^2 L_0^4 n^4}{\eta^2}\right)^2 \frac{1}{m\epsilon^2}\right)$.

(iii) **[Communication complexity]** Suppose $K_\epsilon \geq m^3$. Then, the number of communication rounds to achieve the accuracy level in (ii) is $R := \mathcal{O}\left((mK_\epsilon)^{3/4}\right)$.

We now formally relate the original nonsmooth problem and its smoothed counterpart.

**Proposition 2.** Consider problem (4) and let Assumption 1 hold.

(i) Assume that $X_i = \mathbb{R}^n$ for all $i \in [m]$. Then, for any $\eta > 0$, we have $\nabla f^\eta(x) \in \partial_{2\eta} f(x)$.

(ii) Assume that the sets $X_i$ are identical for all $i \in [m]$. Let $\delta > 0$ be an arbitrary scalar. If $\nabla \mathbf{f}^\eta(x) = 0$ and $\eta \leq \frac{\delta}{\max\{2, nL_0\}}$, then $0_n \in \partial_\delta (f + \mathbb{I}_X)(x)$.

## 4 Extensions to Bilevel and Minimax FL

**4.1 Nondifferentiable nonconvex bilevel FL.** In this section, we consider the federated bilevel optimization problem defined earlier as (**FL**$_{bl}$). We consider the following smoothed implicit problem.

**Definition 3** (Unconstrained smoothed implicit problem)**.** Given $\eta > 0$, consider an unconstrained smoothed implicit problem given as

$$\min_{x \in \mathbb{R}^n} \ \mathbf{f}^\eta(x) \left\{ \triangleq \frac{1}{m} \sum_{i=1}^m \left( \mathbb{E}_{\xi_i, u \in \mathbb{B}}[\tilde{f}_i(x + \eta u, y(x + \eta u), \xi_i)] + \frac{1}{2\eta} \text{dist}^2(x, X_i) \right) \right\}. \quad (5)$$

**Assumption 2.** Consider problem (**FL**$_{bl}$). Let the following assumptions hold.

(i) For all $i \in [m]$, $\tilde{f}_i(\bullet, y, \xi_i)$ is $L_{0,x}^f(\xi_i)$-Lipschitz for any $y$ and $\tilde{f}_i(x, \bullet, \xi_i)$ is $L_{0,y}^f(\xi_i)$-Lipschitz for any $x$, where $L_{0,x}^f \triangleq \max_{i=1,\ldots,m} \sqrt{\mathbb{E}[(L_{0,x}^f(\xi_i))^2]} < \infty$ and $L_{0,y}^f \triangleq \max_{i=1,\ldots,m} \sqrt{\mathbb{E}[(L_{0,y}^f(\xi_i))^2]} < \infty$.

(ii) For all $i \in [m]$, for any $x$, $h_i(x, \bullet)$ is $L_{1,y}^h$-smooth and $\mu_h$-strongly convex. Further, for any $y$, the map $\nabla_y h_i(\bullet, y)$ is Lipschitz continuous with parameter $L_{0,x}^{\nabla h}$.

(iii) The sets $X_i$, for $i \in [m]$, satisfy Assumption 1 (iii).

The outline of FedRZO$_{bl}$ is presented in Algorithm 2. We make the following remarks: (i) At each global step, the server makes two calls to a lower-level FL oracle to inexactly compute $y(\hat{x}_r + v_{T_r})$ and $y(\hat{x}_r)$. These lower-level FL calls are performed by the same clients, on the lower-level FL problem. (ii) The inexactness error is carefully controlled by terminating the lower-level FL oracle after $\mathcal{O}(r)$ number of iterations, where $r$ denotes the upper-level communication round index. (iii) FedRZO$_{bl}$ skips the calls to the lower-level FL oracle during the local steps. To accommodate this, unlike in FedRZO$_{nn}$, here we employ a global randomized smoothing denoted by $v_{T_r}$ during the communication round $r$ in the upper level.

---

**Algorithm 2** Randomized Implicit Zeroth-Order Federated Averaging (FedRZO$_{bl}$)

1: **input:** Server chooses a random $\hat{x}_0 \in X$, stepsize $\gamma$, smoothing parameter $\eta$, synchronization indices $T_0 := 0$ and $T_r \geq 1$, where $r \geq 1$ is the upper-level communication round index
2: **for** $r = 0, 1, \ldots$ **do**
3:     Server generates a random replicate $v_{T_r} \in \eta \mathbb{S}$
4:     Server calls FedAvg to receive $y_{\varepsilon_r}(\hat{x}_r + v_{T_r})$ and $y_{\varepsilon_r}(\hat{x}_r)$, denoting the inexact evaluations of $y(\hat{x}_r + v_{T_r})$ and $y(\hat{x}_r)$, respectively.
5:     Server broadcasts $\hat{x}_r$, $\hat{x}_r + v_{T_r}$, $y_{\varepsilon_r}(\hat{x}_r)$, and $y_{\varepsilon_r}(\hat{x}_r + v_{T_r})$ to all clients; $x_{i,T_r} := \hat{x}_r$, $\forall i$
6:     **for** $k = T_r, \ldots, T_{r+1} - 1$ **do in parallel by clients**
7:         Client $i$ generates the random replicates $\xi_{i,k} \in \mathcal{D}_i$
8:         $g_{i,k}^{\eta,\varepsilon_r} := \frac{n}{\eta^2} \left( \tilde{f}_i(x_{i,k} + v_{T_r}, y_{\varepsilon_r}(\hat{x}_r + v_{T_r}), \xi_{i,k}) - \tilde{f}_i(x_{i,k}, y_{\varepsilon_r}(\hat{x}_r), \xi_{i,k}) \right) v_{T_r}$
9:         Client $i$ does a local update as $x_{i,k+1} := x_{i,k} - \gamma \left( g_{i,k}^{\eta,\varepsilon_r} + \frac{1}{\eta}(x_{i,k} - \mathcal{P}_{X_i}(x_{i,k})) \right)$
10:    **end for**
11:    Server receives $x_{i,T_{r+1}}$ from all clients and aggregates, i.e., $\hat{x}_{r+1} := \frac{1}{m} \sum_{i=1}^m x_{i,T_{r+1}}$
12: **end for**

---

**Theorem 1** (FedRZO$_{bl}$ when using an arbitrary inexact FL method for lower-level)**.** Consider Algorithm 2. Let Assumption 2 hold, $k^*$ be chosen uniformly at random from $0, \ldots, K := T_R - 1$, and $\gamma \leq \min \left\{ \frac{\max\{2, \sqrt{0.1\Theta_3}, 4B_2\sqrt{3\Theta_2}, 4B_2\sqrt{3\Theta_3}\}^{-1}}{4H}, \frac{\eta}{24(L_0^{\text{imp}} n + 1)} \right\}$. Let $\varepsilon_r$ denote the inexactness in obtaining the lower-level solution, i.e., $\mathbb{E}\left[\|y_{\varepsilon_r}(x) - y(x)\|^2 \mid x\right] \leq \varepsilon_r$ for $x \in \cup_{r=0}^R \{\hat{x}_r, \hat{x}_r + v_{T_r}\}$.

(i) **[Error bound]** We have

$$\mathbb{E}\left[\|\nabla \mathbf{f}^\eta(\bar{x}_{k^*})\|^2\right] \le 8(\gamma K)^{-1}(\mathbb{E}\left[\mathbf{f}^\eta(x_0)\right] - \mathbf{f}^{\eta,*}) + \frac{8\gamma\Theta_1}{m} + 8H^2\gamma^2\max\{\Theta_2,\Theta_3\}\Theta_5$$

$$+ 8\left(H^2\gamma^2\max\{\Theta_2,\Theta_3\}\Theta_4 + \Theta_3\right)H\frac{\sum_{r=0}^{R-1}\varepsilon_r}{K},$$

where $\Theta_1 := \frac{3(L_0^{\text{imp}}n+1)n^2}{2\eta}(L_0^{\text{imp}})^2$, $\Theta_2 := \frac{5(L_0^{\text{imp}}n+1)^2}{8\eta^2}$, $\Theta_3 := \left(\frac{L_{0,x}^{\nabla h}}{\mu_h}\right)^2\frac{20n^2}{\eta^2}(L_{0,y}^f)^2$,

$\Theta_4 := \frac{96n^2}{\eta^2}(L_{0,y}^f)^2$, and $\Theta_5 := \frac{48B_1^2}{\eta^2} + (96B_2^2+1)(L_0^{\text{imp}})^2n^2$.

(ii) **[Iteration complexity]** Let $\gamma := \sqrt{\frac{m}{K}}$ and $H := \left\lceil\sqrt[4]{\frac{K}{m^3}}\right\rceil$ where $\eta > 0$. Let $\epsilon > 0$ be an arbitrary scalar and $K_\epsilon$ denote the number of iterations such that $\mathbb{E}\left[\|\nabla\mathbf{f}^\eta(\bar{x}_{k^*})\|^2\right] \le \epsilon$. Also, suppose we employ an FL method in the lower level that achieves a sublinear convergence rate with a linear speedup in terms of the number of clients, i.e., $\varepsilon_r := \tilde{\mathcal{O}}(\frac{1}{m\tilde{T}_{\tilde{R}_r}})$ where $\tilde{R}_r$ denotes the number of communication rounds performed in the lower-level FL method when it is called in round $r$ of FedRZO$_{\text{b1}}$ and $\tilde{T}_{\tilde{R}_r}$ denotes the number of iterations performed in the lower-level FL scheme to do $\tilde{R}_r$ rounds of upper-level communication. Further, suppose $\tilde{T}_{\tilde{R}_r} := \tilde{\mathcal{O}}\left(m^{-1}(r+1)^{\frac{2}{3}}\right)$. Then, the iteration complexity of FedRZO$_{\text{b1}}$ (upper level) is $K_\epsilon := \tilde{\mathcal{O}}\left((\Theta_1^2 + \max\{\Theta_2,\Theta_3\}^2\Theta_5^2 + \max\{\Theta_2,\Theta_3\}^2\Theta_4^2 + \Theta_3^2)\frac{1}{m\epsilon^2}\right)$.

(iii) **[Communication complexity]** Suppose $K_\epsilon \ge m^3$. Then, the number of communication rounds in FedRZO$_{\text{b1}}$ (upper-level only) to achieve the accuracy level in (ii) is $R := \mathcal{O}\left((mK_\epsilon)^{3/4}\right)$.

**Remark 2.** (i) Importantly, Theorem 1 is equipped with explicit communication complexity $R := \mathcal{O}\left((mK_\epsilon)^{3/4}\right)$, matching that of single-level nonsmooth nonconvex problems in Proposition 1. This implies that as long as the lower-level FL oracle has a rate of $\varepsilon_r := \tilde{\mathcal{O}}(\frac{1}{m\tilde{T}_{\tilde{R}_r}})$, the inexactness does not affect the communication complexity bounds of the method in the upper level.

(ii) As noted in the assumptions, in the upper level, we allow for heterogeneity. To elaborate on overall communication complexity of FedRZO$_{\text{b1}}$, we provide detailed complexity results in Table 1 for three cases, where we employ Local SGD [26], FedAC [54], and LFD [20] for the lower-level scheme. All these schemes meet the linear speedup condition in Theorem 1. Notably, among these schemes, the last scheme allows for the presence of heterogeneity. As an example, we present in Algorithm 3, the outline of FedAvg, if employed in step 4 of Algorithm 2.

---

**Algorithm 3** FedAvg $(x, r, y_{0,r}, m, \tilde{\gamma}, \tilde{H}, \tilde{T}_{\tilde{R}})$ for lower level

---

1: **input:** $x, r$, server chooses a random initial point $\hat{y}_0 := y_{0,r} \in \mathbb{R}^{\tilde{n}}$, $a_r := \max\{m, 4\kappa_h, r\} + 1$
   where $\kappa_h := \frac{L_{1,y}^h}{\mu_h}$, $\tilde{\gamma} := \frac{1}{\mu_h a_r}$, $\tilde{T}_{\tilde{R}_r} := 2a_r\ln(a_r)$, and $\tilde{H} := \lceil\frac{\tilde{T}_{\tilde{R}_r}}{m}\rceil$
2: **for** $\tilde{r} = 0, 1, \dots, \tilde{R}_r - 1$ **do**
3:     Server broadcasts $\hat{y}_{\tilde{r}}$ to all agents: $y_{i,\tilde{T}_{\tilde{r}}} := \hat{y}_{\tilde{r}}, \forall i$
4:     **for** $t = \tilde{T}_{\tilde{r}}, \dots, \tilde{T}_{\tilde{r}+1} - 1$ **do in parallel by agents**
5:         Agent $i$ does a local update as $y_{i,t+1} := y_{i,t} - \tilde{\gamma}\nabla_y h_i(x, y_{i,t}, \tilde{\xi}_{i,t})$
6:         Agent $i$ sends $x_{i,T_{r+1}}$ to the server
7:     **end for**
8:     Server aggregates, i.e., $\hat{y}_{\tilde{r}+1} := \frac{1}{m}\sum_{i=1}^m y_{i,T_{\tilde{r}+1}}$
9: **end for**

---

(iii) We use $L_0^{\text{imp}}(\xi_i)$ to denote the Lipschitz continuity constant of the random local implicit function $\tilde{f}_i(x, y(x), \xi_i)$, and let $L_0^{\text{imp}} \triangleq \max_{i=1,\dots,m}\sqrt{\mathbb{E}[(L_0^{\text{imp}}(\xi_i))^2]} < \infty$. As shown in supplementary material, $L_0^{\text{imp}}(\xi_i)$ can be obtained explicitly in terms of problem parameters.

**Remark 3.** A technical challenge in designing Algorithm 2 is that an inexact evaluation of $y(x)$ must be avoided during the local steps. This is because we consider bilevel problems of the form (**FL**$_{bl}$)

where both levels are distributed. Because of this, the inexact evaluation of $y(x)$ by each client in the local step in the upper level would require significant communications that is undesirable in the FL framework. We carefully address this challenge by introducing **delayed inexact computation** of $y(x)$. In step 8 of Algorithm 2, we note how $y_\varepsilon$ is evaluated at $\hat{x}_r + v_{T_r}$ which is a different than the vector used by the client, i.e., $x_{i,k} + v_{T_r}$. At each communication round in the upper level, we only compute $y(x)$ inexactly twice in the global step and then use this delayed information in the local steps. This delayed inexact computation of $y$ renders a challenge in the convergence analysis which makes the design and analysis of Algorithm 2 a non-trivial extension of Algorithm 1.

**4.2 Nondifferentiable nonconvex-strongly concave minimax FL.** Next, we consider the decentralized federated minimax problem of the form ($\textbf{FL}_{mm}$) introduced earlier. This problem is indeed a zero-sum game and can be viewed as an instance of the non-zero sum game ($\textbf{FL}_{bl}$) where $\tilde{h}_i := -\tilde{f}_i$.

**Corollary 1.** Consider Algorithm 2 for solving ($\textbf{FL}_{mm}$). Let Assumption 2 hold for $\tilde{h}_i := -\tilde{f}_i$ and $\mathcal{D}_i := \tilde{\mathcal{D}}_i$. Then, all the results in Theorem 1 hold true.

# 5 Experiments

We present three sets of experiments to validate the performance of the proposed algorithms. In Section 5.1, we implement Algorithm 1 on ReLU neural networks (NNs) and compare it with some recent FL methods. In Sections 5.2 and 5.3 we implement Algorithm 2 on federated hyperparameter learning and a minimax formulation in FL. Throughout, we use the MNIST dataset. Additional experiments on a higher dimensional dataset (i.e., Cifar-10) are presented in supplementary material.

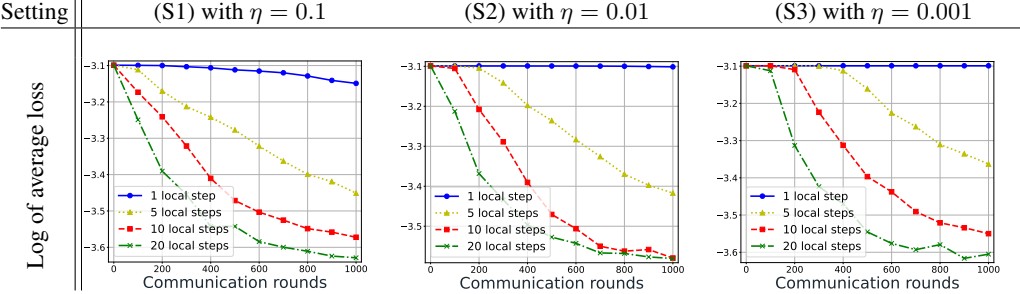

Figure 1: Performance of FedRZO$_{nn}$ on a single-layer ReLU NN in terms of communication rounds for different no. of local steps and different values of the smoothing parameter $\eta$. FedRZO$_{nn}$ benefits from larger number of local steps and shows robustness with respect to the choice of $\eta$.

**5.1 Federated training of ReLU NNs.** We implement FedRZO$_{nn}$ for federated training in a single-layer ReLU NN with $N_1$ neurons. This is a nondifferentiable nonconvex optimization problem, aligning with ($\textbf{FL}_{nn}$) and taking the form $\min_{x:=(Z,w)\in\mathcal{X}} \frac{1}{2m}\sum_{i=1}^m \sum_{\ell\in\mathcal{D}_i}(v_{i,\ell} - \sum_{q=1}^{N_1} w_q\sigma(Z_{\bullet,q}U_{i,\ell}))^2 + \frac{\lambda}{2}\left(\|Z\|_F^2 + \|w\|^2\right)$, where $m$ denotes the number of clients, $Z\in\mathbb{R}^{N_1\times N_0}$, $w\in\mathbb{R}^{N_1}$, $N_0$ is the feature dimension, $U_{i,\ell}\in\mathbb{R}^{N_0}$ and $v_{i,\ell}\in\{-1,1\}$ are the $\ell$th input and output training sample of client $i$, respectively, $\sigma(x):=\max\{0,x\}$, and $\lambda$ is the regularization parameter.

**Setup.** We distribute the training dataset among $m := 5$ clients and implement FedRZO$_{nn}$ for the FL training with $N_1 := 4$ neurons under three different settings for the smoothing parameter, $\eta\in\{0.1,0.01,0.001\}$, $\gamma := 10^{-5}$, and $\lambda := 0.01$. We study the performance of the method under different number of local steps with $H\in\{1,5,10,20\}$.

**Results and insights.** Figure 1 presents the first set of numerics for FedRZO$_{nn}$ under the aforementioned settings. In terms of communication rounds, we observe that the performance of the method improves by using a larger number of local steps. In fact, in the case where $H := 1$, FedRZO$_{nn}$ is equivalent to a parallel zeroth-order SGD that employs communication among clients at each iteration, resulting in a poor performance, motivating the need for the FL framework. In terms of $\eta$, while we observe robustness of the scheme in terms of the original loss function, we also note a slight improvement in the empirical speed of convergence in early steps, as $\eta$ increases. This is indeed aligned with the dependence of convergence bound in Proposition 1 on $\eta$.

**Comparison with other FL methods.** While we are unaware of other FL methods for addressing nondifferentiable nonconvex problems, we compare FedRZO$_{\tt nn}$ with other FL methods including FedAvg [34], FedProx [29], FedMSPP [55], and Scaffnew [35] when applied on a NN with a smooth rectifier. Details of these experiments are provided in the supplementary material.

**5.2 Federated hyperparameter learning.** To validate FedRZO$_{\tt bl}$, we consider the following FL hyperparameter learning problem for binary classification using logistic loss.

$$\min_{x \in X, \ y \in \mathbb{R}^n} \quad f(x,y) \triangleq \tfrac{1}{m} \sum_{i=1}^m \sum_{\ell \in \mathcal{D}_i} \log \left( 1 + \exp(-v_{i,\ell} U_{i,\ell}^T y) \right)$$

$$\text{subject to} \quad y \in \arg\min_{y \in \mathbb{R}^n} \quad h(x,y) \triangleq \tfrac{1}{m} \sum_{i=1}^m (\sum_{\tilde{\ell} \in \tilde{D}_i} \log \left( 1 + \exp(-v_{i,\tilde{\ell}} U_{i,\tilde{\ell}}^T y) \right) + x_i \tfrac{\|y\|^2}{2}),$$

where $m$ is number of clients, $x$ denotes the regularization parameter for client $i$, $U_{i,\ell} \in \mathbb{R}^n$ and $v_{i,\ell} \in \{-1, 1\}$ are the $\ell$th input and output testing sample of client $i$, respectively, $U_{i,\ell'} \in \mathbb{R}^n$ and $v_{i,\ell'} \in \{-1, 1\}$ are the $\ell'$th input and output training sample of client $i$, respectively. The constraint set $X$ is considered as $X := \{x \in \mathbb{R}^m \mid x \geq \underline{\mu}\mathbf{1}_m\}$, where $\underline{\mu} > 0$. This problem is an instance of (**FL**$_{bl}$), where the lower-level problem is $\ell_2$-regularized and the regularization parameter is a decision variable of the upper-level FL problem. The convergence results are presented in Fig. 2 (left).

**5.3 Fair classification learning.** Here, we study the convergence of FedRZO$_{\tt bl}$ in minimax FL. We consider solving an FL minimax formulation of the fair classification problem [39] of the form

$$\min_{x \in \mathbb{R}^n} \max_{y \in \mathbb{R}^c} \quad \tfrac{1}{m} \sum_{i=1}^m \sum_{c=1}^C \sum_{\ell \in \mathcal{D}_{i,c}} (v_{i,\ell} - \sum_{q=1}^{N_1} w_q \sigma(Z_{\bullet,q} U_{i,\ell}))^2 - \tfrac{\lambda}{2} \|y\|^2,$$

where $c$ denotes the class index and $\mathcal{D}_{i,c}$ denotes the a portion of local dataset associated with client $i$ that is comprised of class $c$ samples. The loss function follows the same formulation in Section 5.1, where an ReLU neural network is employed. This problem is nondifferentiable and nonconvex-strongly concave, fitting well with the assumptions in our work in addressing minimax FL problems. The performance of our algorithm is presented in Figure 2 (right).

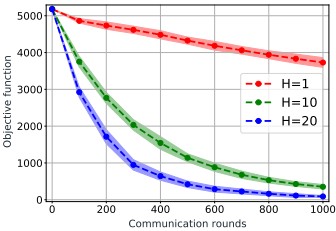 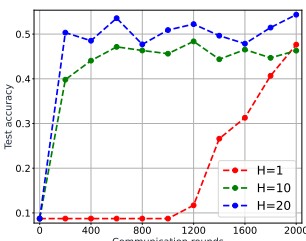

Figure 2: (Left) Convergence of FedRZO$_{\tt bl}$ in hyperparameter FL for $\ell_2$ regularized logistic loss, where we plot the loss function on test data for different values of local steps with $95\%$ CIs. (Right) Convergence of FedRZO$_{\tt bl}$ in minimax FL, where we present test results in solving a nondifferentiable nonconvex-strongly concave FL minimax formulation of the fair classification problem [39].

# 6 Concluding Remarks

Federated learning has assumed growing relevance in ML. However, most practical problems are characterized by the presence of local objectives, jointly afflicted by nonconvexity and nondifferentiability, precluding resolution by most FL schemes, which can cope with nonconvexity in only smooth settings. We resolve this gap via a zeroth-order communication-efficient FL framework that can contend with both nondifferentiability and nonsmoothness with rate and complexity guarantees for computing approximate Clarke-stationary points. Extensions to nonconvex bilevel and nonconvex-strongly concave minimax settings are developed via inexact generalizations.

# 7 Acknowledgments

We acknowledge the funding support from the U.S. Department of Energy under grant #DE-SC0023303, and the U.S. Office of Naval Research under grants #N00014-22-1-2757 and #N00014-22-1-2589. We also would like to thank the four anonymous referees for their constructive suggestions.

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
