# Supplementary Material

The supplementary material is organized as follows. First, we prove Proposition 1 and Theorem 1. We then provide some details on the overall communication complexity of FedRZO$_{\texttt{bl}}$. Lastly, we present some additional numerical experiments.

## A  Proof of Proposition 1

In this section we prove Proposition 1, and some preliminary lemmas. Before going into the proof details, we introduce some basic notation for clarity.

### A.1  Notation in proofs

**Definition 4.** Let the function $d_i(x)$ be defined as $d_i^\eta(x) \triangleq \frac{1}{2\eta}\mathrm{dist}^2(x, X_i)$. Let $x_{i,k}$ be given by Algorithm 1 for all $i \in [m]$ and $k \geq 0$. Let us define the following terms:

$$g_{i,k}^\eta \triangleq \frac{n}{\eta^2}\left(\tilde{f}_i(x_{i,k} + v_{i,k}, \xi_{i,k}) - \tilde{f}_i(x_{i,k}, \xi_{i,k})\right)v_{i,k}, \quad \nabla_{i,k}^{\eta,d} \triangleq \frac{1}{\eta}(x_{i,k} - \mathcal{P}_{X_i}(x_{i,k})),$$

$$\bar{g}_k^\eta \triangleq \frac{\sum_{i=1}^m g_{i,k}^\eta}{m}, \quad \bar{\nabla}_k^{\eta,d} \triangleq \frac{\sum_{i=1}^m \nabla_{i,k}^{\eta,d}}{m}, \quad \bar{x}_k \triangleq \frac{\sum_{i=1}^m x_{i,k}}{m}, \quad \bar{e}_k \triangleq \frac{\sum_{i=1}^m \|x_{i,k} - \bar{x}_k\|^2}{m}.$$

Here, $\bar{x}_k$ is an auxiliary sequence that denotes the average iterates of the clients at any iteration $k$ and $\bar{e}_k$ denotes an average consensus error at that iteration.

We will make use of the following notation for the history of the method. Let $\mathcal{F}_0 = \{\hat{x}_0\}$ and

$$\mathcal{F}_k \triangleq \cup_{i=1}^m \cup_{t=0}^{k-1}\{\xi_{i,t}, v_{i,t}\} \cup \{\hat{x}_0\}, \quad \text{for all } k \geq 1.$$

Throughout, we assume that the $i$th client generates i.i.d. random samples $\xi_i \in \mathcal{D}_i$ and $v_i \in \eta\mathbb{S}$. These samples are assumed to be independent across clients.

We use $\mathcal{O}(\cdot)$ to denote the big O notation and $\tilde{\mathcal{O}}(\cdot)$ to denote a variation of $\mathcal{O}(\cdot)$ that ignores polylogarithmic and constant numerical factors.

### A.2  Preliminary results

**Proof of Lemma 1.**

*Proof.* (i) By definition, we have that

$$h^\eta(x) = \mathbb{E}_{u \in \mathbb{B}}[h(x + \eta u)] = \int_\mathbb{B} h(x + \eta u)p(u)du.$$

Let $p(u)$ denote the probability density function of $u$. Since $u$ is uniformly distributed on the unit ball $\mathbb{B}$, we have that $p(u) = \frac{1}{\mathrm{Vol}_n(\mathbb{B})}$ for any $u \in \mathbb{B}$, where $\mathrm{Vol}_n(\mathbb{B})$ denotes volume of $n$ dimensional unit ball. We obtain

$$h^\eta(x) = \int_\mathbb{B} h(x + \eta u)p(u)du = \frac{\int_\mathbb{B} h(x + \eta u)du}{\mathrm{Vol}_n(\mathbb{B})}.$$

Next we compute the derivative $\nabla_x h^\eta(x)$ by leveraging Stoke's theorem. Let us define $\tilde{p}(v) = \frac{1}{\mathrm{Vol}_{n-1}(\eta\mathbb{S})}$ for all $v$. We have

$$\nabla_x h^\eta(x) = \nabla_x \left[\frac{\int_\mathbb{B} h(x + \eta u)du}{\mathrm{Vol}_n(\mathbb{B})}\right] \overset{\text{Stoke's theorem}}{=} \left[\frac{\int_{\eta\mathbb{S}} h(x + v)\frac{v}{\|v\|}dv}{\mathrm{Vol}_n(\mathbb{B})}\right]$$

$$= \left[\frac{\int_{\eta\mathbb{S}} h(x + v)\frac{v}{\|v\|}dv}{\mathrm{Vol}_{n-1}(\eta\mathbb{S})}\right]\frac{\mathrm{Vol}_{n-1}(\eta\mathbb{S})}{\mathrm{Vol}_n(\mathbb{B})} = \left[\int_{\eta\mathbb{S}} h(x + v)\frac{v}{\|v\|}\tilde{p}(v)dv\right]\frac{n}{\eta} = \frac{n}{\eta}\mathbb{E}_{v \in \eta\mathbb{S}}\left[h(x + v)\frac{v}{\|v\|}\right].$$

(ii) We have

$$|h^\eta(x) - h^\eta(y)| = |\mathbb{E}_{u\in\mathbb{B}}[h(x+\eta u)] - \mathbb{E}_{u\in\mathbb{B}}[h(y+\eta u)]| \overset{\text{Jensen's ineq.}}{\leq} \mathbb{E}_{u\in\mathbb{B}}[|h(x+\eta u) - h(y+\eta u)|]$$
$$\leq \mathbb{E}_{u\in\mathbb{B}}[L_0\|x-y\|] = L_0\|x-y\|.$$

(iii)

$$|h^\eta(x) - h(x)| = \left|\int_{\mathbb{B}} (h(x+\eta u) - h(x))p(u)du\right|$$
$$\leq \int_{\mathbb{B}} |(h(x+\eta u) - h(x))|\, p(u)du$$
$$\leq L_0 \int_{\mathbb{B}} \eta\|u\|p(u)du \leq L_0\eta \int_{\mathbb{B}} p(u)du = L_0\eta.$$

(iv)

$$\|\nabla_x h^\eta(x) - \nabla_x h^\eta(y)\| = \left\|\tfrac{n}{\eta}\mathbb{E}_{v\in\eta\mathbb{S}}\left[h(x+v)\tfrac{v}{\|v\|}\right] - \tfrac{n}{\eta}\mathbb{E}_{v\in\mathbb{S}}\left[h(y+v)\tfrac{v}{\|v\|}\right]\right\|$$
$$\leq \tfrac{n}{\eta}\mathbb{E}_{v\in\eta\mathbb{S}}\left[\left\|(h(x+v) - h(y+v))\tfrac{v}{\|v\|}\right\|\right]$$
$$\leq \tfrac{L_0 n}{\eta}\|x-y\|\mathbb{E}_{v\in\eta\mathbb{S}}\left[\tfrac{\|v\|}{\|v\|}\right] = \tfrac{L_0 n}{\eta}\|x-y\|.$$

$\square$

**Remark 4** (Compact local representation of Algorithm 1). Let us define $\mathcal{I} \triangleq \{K_1, K_2, \ldots\}$ where $K_r \triangleq T_r - 1$ for $r \geq 1$. The following equation, for $k \geq 0$, compactly represents the update rules of Algorithm 1.

$$x_{i,k+1} := \begin{cases} \frac{1}{m}\sum_{j=1}^m \left(x_{j,k} - \gamma\left(g_{j,k}^\eta + \nabla_{j,k}^{\eta,d}\right)\right), & k \in \mathcal{I} \\ x_{i,k} - \gamma\left(g_{i,k}^\eta + \nabla_{i,k}^{\eta,d}\right), & k \notin \mathcal{I}. \end{cases} \tag{6}$$

**Lemma 2.** *Consider Algorithm 1 and Definition 4. For all $k \geq 0$, we have*

$$\bar{x}_{k+1} = \bar{x}_k - \gamma\left(\bar{g}_k^\eta + \bar{\nabla}_k^{\eta,d}\right).$$

*Proof. Case 1:* When $k \in \mathcal{I}$, from equation (6) we can write

$$x_{i,k+1} = \tfrac{1}{m}\sum_{j=1}^m \left(x_{j,k} - \gamma\left(g_{j,k}^\eta + \nabla_{j,k}^{\eta,d}\right)\right)$$
$$= \tfrac{1}{m}\sum_{j=1}^m x_{j,k} - \gamma\tfrac{1}{m}\sum_{j=1}^m(g_{j,k}^\eta + \nabla_{j,k}^{\eta,d}) = \bar{x}_k - \gamma\left(\bar{g}_k^\eta + \bar{\nabla}_k^{\eta,d}\right),$$

where the last equation is implied by the definition of $\bar{x}_k$, $\bar{g}_k^\eta$ and $\bar{\nabla}_k^{\eta,d}$. Averaging on both sides over $i = 1, \ldots, m$, we obtain $\bar{x}_{k+1} = \bar{x}_k - \gamma\left(\bar{g}_k^\eta + \bar{\nabla}_k^{\eta,d}\right)$.

*Case 2:* When $k \notin \mathcal{I}$, from equation (6) we can write

$$x_{i,k+1} = x_{i,k} - \gamma g_{i,k}^\eta + \nabla_{i,k}^{\eta,d}.$$

Summing over $i = 1, \ldots, m$ on both sides and then dividing both sides by $m$, we obtain

$$\tfrac{1}{m}\sum_{i=1}^m x_{i,k+1} = \tfrac{1}{m}\sum_{i=1}^m x_{i,k} - \gamma\tfrac{1}{m}\sum_{i=1}^m(g_{i,k}^\eta + \nabla_{i,k}^{\eta,d}).$$

Invoking Definition 4, we obtain the desired result. $\square$

**Lemma 3** (Properties of zeroth-order stochastic local gradient). Let Assumption 1 hold. Consider Algorithm 1. Then, the following relations hold for all $k \geq 0$ and all $i \in [m]$ in an almost-sure sense.

(i) $\mathbb{E}\left[g_{i,k}^\eta + \nabla_{i,k}^{\eta,d} \mid \mathcal{F}_k\right] = \nabla\mathbf{f}_i^\eta(x_{i,k})$.

(ii) $\mathbb{E}\left[\|g_{i,k}^{\eta} + \nabla_{i,k}^{\eta,d}\|^2 \mid \mathcal{F}_k\right] \leq \frac{12n^2\nu^2}{\eta^2} + 6L_0^2 n^2 + 2\|\nabla_{i,k}^{\eta,d}\|^2.$

(iii) $\mathbb{E}\left[\|(g_{i,k}^{\eta} + \nabla_{i,k}^{\eta,d}) - \nabla\mathbf{f}_i^{\eta}(x_{i,k})\|^2 \mid \mathcal{F}_k\right] \leq \frac{6n^2\nu^2}{\eta^2} + 3L_0^2 n^2.$

*Proof.* (i) From the definition of the zeroth-order stochastic gradient, we can write

$$\mathbb{E}\left[g_{i,k}^{\eta} \mid \mathcal{F}_k\right] = \mathbb{E}_{v_{i,k}}\left[\mathbb{E}_{\xi_{i,k}}\left[\frac{n}{\eta^2}\left(\tilde{f}_i(x_{i,k} + v_{i,k}, \xi_{i,k}) - \tilde{f}_i(x_{i,k}, \xi_{i,k})\right)v_{i,k} \mid \mathcal{F}_k \cup \{v_{i,k}\}\right]\right]$$

$$= \mathbb{E}_{v_{i,k}}\left[\frac{n}{\eta^2}f_i(x_{i,k} + v_{i,k})v_{i,k} \mid \mathcal{F}_k\right] \overset{\text{Lemma 1(i)}}{=} \nabla f_i^{\eta}(x_{i,k}).$$

Adding the preceding equality to $\nabla_{i,k}^{\eta,d}$, noting that $\nabla_{i,k}^{\eta,d} = \nabla d_i^{\eta}(x_{i,k})$, and using equation (4), we obtain the relation in (i).

(ii) From the definition of $g_{i,k}^{\eta}$ and that $\|v_{i,k}\| = \eta$, we have

$$\mathbb{E}\left[\left\|g_{i,k}^{\eta}\right\|^2 \mid \mathcal{F}_k \cup \{v_{i,k}\}\right] = \left(\tfrac{n^2}{\eta^4}\right)\mathbb{E}\left[|\tilde{f}_i(x_{i,k} + v_{i,k}, \xi_{i,k}) - \tilde{f}_i(x_{i,k}, \xi_{i,k})|^2 \mid \mathcal{F}_k \cup \{v_{i,k}\}\right]\|v_{i,k}\|^2$$

$$= \left(\tfrac{n^2}{\eta^2}\right)\mathbb{E}\left[|\tilde{f}_i(x_{i,k} + v_{i,k}, \xi_{i,k}) - \tilde{f}_i(x_{i,k}, \xi_{i,k})|^2 \mid \mathcal{F}_k \cup \{v_{i,k}\}\right].$$

Consider the term $|\tilde{f}_i(x_{i,k} + v_{i,k}, \xi_{i,k}) - \tilde{f}_i(x_{i,k}, \xi_{i,k})|^2$. Adding and subtracting $f_i(x_{i,k} + v_{i,k})$ and $f_i(x_{i,k})$, we can write

$$|\tilde{f}_i(x_{i,k} + v_{i,k}, \xi_{i,k}) - \tilde{f}_i(x_{i,k}, \xi_{i,k})|^2 \leq 3|\tilde{f}_i(x_{i,k} + v_{i,k}, \xi_{i,k}) - f_i(x_{i,k} + v_{i,k})|^2$$

$$+ 3|f_i(x_{i,k} + v_{i,k}) - f_i(x_{i,k})|^2 + 3|f_i(x_{i,k}) - \tilde{f}_i(x_{i,k}, \xi_{i,k})|^2.$$

From the two preceding relations and invoking Assumption 1 (ii), we obtain

$$\mathbb{E}\left[\left\|g_{i,k}^{\eta}\right\|^2 \mid \mathcal{F}_k \cup \{v_{i,k}\}\right] \leq \tfrac{n^2}{\eta^2}\left(6\nu^2 + 3|f_i(x_{i,k} + v_{i,k}) - f_i(x_{i,k})|^2\right).$$

Taking expectations with respect to $v_{i,k}$ on both sides and invoking Assumption 1 (i), we have

$$\mathbb{E}\left[\left\|g_{i,k}^{\eta}\right\|^2 \mid \mathcal{F}_k\right] \leq \tfrac{n^2}{\eta^2}\left(6\nu^2 + 3L_0^2\eta^2\right). \tag{7}$$

From the preceding relation, we can write

$$\mathbb{E}\left[\|g_{i,k}^{\eta} + \nabla_{i,k}^{\eta,d}\|^2 \mid \mathcal{F}_k\right] \leq 2\mathbb{E}\left[\|g_{i,k}^{\eta}\|^2 \mid \mathcal{F}_k\right] + 2\|\nabla_{i,k}^{\eta,d}\|^2 \leq \tfrac{12n^2\nu^2}{\eta^2} + 6L_0^2 n^2 + 2\|\nabla_{i,k}^{\eta,d}\|^2.$$

(iii) Invoking the bounds given by (i) and (7), and using equation (4), we have

$$\mathbb{E}\left[\|(g_{i,k}^{\eta} + \nabla_{i,k}^{\eta,d}) - \nabla\mathbf{f}_i^{\eta}(x_{i,k})\|^2 \mid \mathcal{F}_k\right] = \mathbb{E}\left[\|g_{i,k}^{\eta} + \nabla_{i,k}^{\eta,d} - \nabla f_i^{\eta}(x_{i,k}) - \nabla_{i,k}^{\eta,d}\|^2 \mid \mathcal{F}_k\right]$$

$$= \mathbb{E}\left[\|g_{i,k}^{\eta}\|^2 + \|\nabla f_i^{\eta}(x_{i,k})\|^2 \mid \mathcal{F}_k\right] - 2\mathbb{E}\left[g_{i,k}^{\eta}{}^T \nabla f_i^{\eta}(x_{i,k}) \mid \mathcal{F}_k\right]$$

$$= \mathbb{E}\left[\|g_{i,k}^{\eta}\|^2 \mid \mathcal{F}_k\right] + \mathbb{E}\left[\|\nabla f_i^{\eta}(x_{i,k})\|^2 \mid \mathcal{F}_k\right] - 2\mathbb{E}\left[\|\nabla f_i^{\eta}(x_{i,k})\|^2 \mid \mathcal{F}_k\right] \leq \tfrac{6n^2\nu^2}{\eta^2} + 3L_0^2 n^2.$$

$\square$

**Lemma 4** (Aggregated zeroth-order gradient). Let Assumption 1 hold. Consider Algorithm 1. Then, for all $k \geq 0$, the following results hold almost surely:

(i) $\mathbb{E}\left[\|\bar{g}_k^{\eta} + \bar{\nabla}_k^{\eta,d}\|^2 \mid \mathcal{F}_k\right] \leq \tfrac{n^2}{m}\left(\tfrac{6\nu^2}{\eta^2} + 3L_0^2\right) + 2\|\nabla\mathbf{f}^{\eta}(\bar{x}_k)\|^2 + \tfrac{2(L_0 n + 1)^2}{\eta^2}\bar{e}_k.$

(ii) $\mathbb{E}\left[\nabla\mathbf{f}^{\eta}(\bar{x}_k)^T(\bar{g}_k^{\eta} + \bar{\nabla}_k^{\eta,d}) \mid \mathcal{F}_k\right] \geq \tfrac{1}{2}\|\nabla\mathbf{f}^{\eta}(\bar{x}_k)\|^2 - \tfrac{(L_0 n + 1)^2}{2mn^2}\sum_{i=1}^{m}\|x_{i,k} - \bar{x}_k\|^2.$

*Proof.* (i) Using the definition of $\bar{g}_k^\eta$, we can write

$$\mathbb{E}\left[\|\bar{g}_k^\eta + \bar{\nabla}_k^{\eta,d}\|^2 \mid \mathcal{F}_k\right] = \mathbb{E}\left[\left\|\frac{1}{m}\sum_{i=1}^m (g_{i,k}^\eta + \nabla_{i,k}^{\eta,d})\right\|^2 \mid \mathcal{F}_k\right] \tag{8}$$

$$=\mathbb{E}\left[\left\|\frac{1}{m}\sum_{i=1}^m \left(g_{i,k}^\eta + \nabla_{i,k}^{\eta,d} - \nabla \mathbf{f}_i^\eta(x_{i,k}) + \nabla \mathbf{f}_i^\eta(x_{i,k})\right)\right\|^2 \mid \mathcal{F}_k\right]$$

$$\overset{\text{Lemma 3 (i)}}{=} \mathbb{E}\left[\left\|\frac{1}{m}\sum_{i=1}^m \left(g_{i,k}^\eta + \nabla_{i,k}^{\eta,d} - \nabla \mathbf{f}_i^\eta(x_{i,k})\right)\right\|^2 \mid \mathcal{F}_k\right] + \left\|\frac{1}{m}\sum_{i=1}^m \nabla \mathbf{f}_i^\eta(x_{i,k})\right\|^2$$

$$\overset{\text{Lemma 3 (i)}}{=} \frac{1}{m^2}\sum_{i=1}^m \mathbb{E}\left[\left\|g_{i,k}^\eta + \nabla_{i,k}^{\eta,d} - \nabla \mathbf{f}_i^\eta(x_{i,k})\right\|^2 \mid \mathcal{F}_k\right]$$

$$+ \left\|\nabla \mathbf{f}^\eta(\bar{x}_k) + \frac{1}{m}\sum_{i=1}^m \left(\nabla \mathbf{f}_i^\eta(x_{i,k}) - \nabla \mathbf{f}_i^\eta(\bar{x}_k)\right)\right\|^2$$

$$\overset{\text{Lemma 3 (iii)}}{\leq} \frac{n^2}{m}\left(\frac{6\nu^2}{\eta^2} + 3L_0^2\right) + 2\left\|\nabla \mathbf{f}^\eta(\bar{x}_k)\right\|^2 + 2\left\|\frac{1}{m}\sum_{i=1}^m \left(\nabla \mathbf{f}_i^\eta(x_{i,k}) - \nabla \mathbf{f}_i^\eta(\bar{x}_k)\right)\right\|^2.$$

Note that given vectors $y_i \in \mathbb{R}^n$ for $i \in [m]$, we have $\|\frac{1}{m}\sum_{i=1}^m y_i\|^2 \leq \frac{1}{m}\sum_{i=1}^m \|y_i\|^2$. Utilizing this inequality together with Lemma 1 (iv), we obtain

$$\mathbb{E}\left[\|\bar{g}_k^\eta + \bar{\nabla}_k^{\eta,d}\|^2 \mid \mathcal{F}_k\right] \leq \frac{n^2}{m}\left(\frac{6\nu^2}{\eta^2} + 3L_0^2\right) + 2\left\|\nabla \mathbf{f}^\eta(\bar{x}_k)\right\|^2 + \frac{2(L_0 n+1)^2}{\eta^2 m}\sum_{i=1}^m \|x_{i,k} - \bar{x}_k\|^2.$$

Recalling the definition of $\bar{e}_k$, we obtain the required bound.

(ii) We have

$$\mathbb{E}\left[\nabla \mathbf{f}^\eta(\bar{x}_k)^T(\bar{g}_k^\eta + \bar{\nabla}_k^{\eta,d}) \mid \mathcal{F}_k\right] = \nabla \mathbf{f}^\eta(\bar{x}_k)^T \mathbb{E}\left[\frac{1}{m}\sum_{i=1}^m (g_{i,k}^\eta + \bar{\nabla}_{i,k}^{\eta,d}) \mid \mathcal{F}_k\right]$$

$$\overset{\text{Lemma 3 (i)}}{=} \nabla \mathbf{f}^\eta(\bar{x}_k)^T\left(\frac{1}{m}\sum_{i=1}^m \nabla \mathbf{f}_i^\eta(x_{i,k})\right)$$

$$= \nabla \mathbf{f}^\eta(\bar{x}_k)^T \frac{1}{m}\sum_{i=1}^m \left(\nabla \mathbf{f}_i^\eta(x_{i,k}) - \nabla \mathbf{f}_i^\eta(\bar{x}_k) + \nabla \mathbf{f}_i^\eta(\bar{x}_k)\right)$$

$$= \nabla \mathbf{f}^\eta(\bar{x}_k)^T \frac{1}{m}\sum_{i=1}^m \left(\nabla \mathbf{f}_i^\eta(x_{i,k}) - \nabla \mathbf{f}_i^\eta(\bar{x}_k)\right) + \left\|\nabla \mathbf{f}^\eta(\bar{x}_k)\right\|^2$$

$$\geq -\frac{1}{2}\|\nabla \mathbf{f}^\eta(\bar{x}_k)\|^2 - \frac{1}{2}\left\|\frac{1}{m}\sum_{i=1}^m \left(\nabla \mathbf{f}_i^\eta(x_{i,k}) - \nabla \mathbf{f}_i^\eta(\bar{x}_k)\right)\right\|^2 + \left\|\nabla \mathbf{f}^\eta(\bar{x}_k)\right\|^2$$

$$\geq \frac{1}{2}\|\nabla \mathbf{f}^\eta(\bar{x}_k)\|^2 - \frac{1}{2m}\sum_{i=1}^m \left\|\nabla \mathbf{f}_i^\eta(x_{i,k}) - \nabla \mathbf{f}_i^\eta(\bar{x}_k)\right\|^2$$

$$\overset{\text{Lemma 1 (iv)}}{\geq} \frac{1}{2}\|\nabla \mathbf{f}^\eta(\bar{x}_k)\|^2 - \frac{(L_0 n+1)^2}{2mn^2}\sum_{i=1}^m \|x_{i,k} - \bar{x}_k\|^2.$$

The bound is obtained by recalling the definition of $\bar{e}_k$. $\qquad\square$

**Lemma 5.** Let Assumption 1 hold. Consider Algorithm 1. For any $k$, we have

$$\frac{1}{m}\sum_{i=1}^m \|\nabla_{i,k}^{\eta,d}\|^2 \leq 8\bar{e}_k + \frac{2B_1^2}{\eta^2} + 4B_2^2\|\nabla \mathbf{f}^\eta(\bar{x}_k)\|^2 + 4B_2^2 L_0^2 n^2.$$

*Proof.* Invoking Definition 4, we can write

$$\frac{1}{m}\sum_{i=1}^m \|\nabla_{i,k}^{\eta,d}\|^2 = \frac{1}{m}\sum_{i=1}^m \|\nabla_{i,k}^{\eta,d} - \nabla d_i^\eta(\bar{x}_k) + \nabla d_i^\eta(\bar{x}_k)\|^2$$

$$\leq \frac{2}{m}\sum_{i=1}^m \|\nabla_{i,k}^{\eta,d} - \nabla d_i^\eta(\bar{x}_k)\|^2 + \frac{2}{m}\sum_{i=1}^m \|\nabla d_i^\eta(\bar{x}_k)\|^2$$

$$= \frac{2}{m}\sum_{i=1}^m \|(x_{i,k} - \bar{x}_k) - (\mathcal{P}_{X_i}(x_{i,k}) - \mathcal{P}_{X_i}(\bar{x}_k))\|^2 + \frac{2}{m\eta^2}\sum_{i=1}^m \text{dist}^2(\bar{x}_k, X_i)$$

$$\leq 8\bar{e}_k + \frac{2}{m\eta^2}\sum_{i=1}^m \text{dist}^2(\bar{x}_k, X_i),$$

where the last inequality is obtained using the nonexpansiveness of the projection operator. Employing Assumption 1 (iii), we obtain

$$\frac{1}{m}\sum_{i=1}^m \|\nabla_{i,k}^{\eta,d}\|^2 \leq 8\bar{e}_k + \frac{2B_1^2}{\eta^2} + \frac{2B_2^2}{\eta^2}\left\|\bar{x}_k - \frac{1}{m}\sum_{i=1}^m \mathcal{P}_{X_i}(\bar{x}_k)\right\|^2$$

$$\leq 8\bar{e}_k + \frac{2B_1^2}{\eta^2} + 2B_2^2\left\|\frac{1}{m}\sum_{i=1}^m \frac{1}{\eta}\left(\bar{x}_k - \mathcal{P}_{X_i}(\bar{x}_k)\right)\right\|^2$$

$$= 8\bar{e}_k + \frac{2B_1^2}{\eta^2} + 2B_2^2\left\|\frac{1}{m}\sum_{i=1}^m \nabla d_i^\eta(\bar{x}_k)\right\|^2$$

$$= 8\bar{e}_k + \frac{2B_1^2}{\eta^2} + 2B_2^2\left\|\frac{1}{m}\sum_{i=1}^m \left(\nabla \mathbf{f}_i^\eta(\bar{x}_k) - \nabla f_i^\eta(\bar{x}_k)\right)\right\|^2$$

$$\leq 8\bar{e}_k + \frac{2B_1^2}{\eta^2} + 4B_2^2\|\nabla \mathbf{f}^\eta(\bar{x}_k)\|^2 + 4B_2^2\|\nabla f^\eta(\bar{x}_k)\|^2.$$

Next, we find an upper bound on $\|\nabla f^\eta(\bar{x}_k)\|^2$. Using the definition of $f(x)$ and invoking Lemma 1, for any $x$ we have

$$\|\nabla f^\eta(x)\|^2 \leq \frac{1}{m}\sum_{i=1}^m \|\nabla f_i^\eta(x)\|^2 = \left(\frac{n^2}{\eta^2}\right)\frac{1}{m}\sum_{i=1}^m \left\|\mathbb{E}_{v_i\in\eta\mathbb{S}}[f_i(x+v_i)\frac{v_i}{\|v_i\|}]\right\|^2$$

$$= \left(\frac{n^2}{\eta^2}\right)\frac{1}{m}\sum_{i=1}^m \left\|\mathbb{E}[(f_i(x+v)-f_i(x))\frac{v_i}{\|v_i\|}]\right\|^2$$

$$\overset{\text{Jensen's ineq.}}{\leq} \left(\frac{n^2}{\eta^2}\right)\frac{1}{m}\sum_{i=1}^m \mathbb{E}[|f_i(x+v)-f_i(x)|^2]$$

$$\overset{\text{Assumption 1 (i)}}{\leq} \left(\frac{n^2}{\eta^2}\right)\frac{1}{m}\sum_{i=1}^m \mathbb{E}[L_0^2\|v_i\|^2] = n^2 L_0^2.$$

From the two preceding inequalities, we obtain the result. $\qquad\square$

**Lemma 6.** Suppose for $T_r + 1 \leq k \leq T_{r+1}$, the nonnegative sequences $\{a_k\}$ and $\{\theta_k\}$ satisfy a recursive relation of the form $a_k \leq (k-T_r)\gamma^2\sum_{t=T_r}^{k-1}(\beta a_t+\theta_t)$, where $a_{T_r}=0$ and $\beta>0$. Then, $a_k \leq H\gamma^2\sum_{t=T_r}^{k-1}(\beta H\gamma^2+1)^{k-t-1}\theta_t$ for all $T_r+1\leq k\leq T_{r+1}$. Moreover, if $0<\gamma\leq\frac{1}{\sqrt{\beta H}}$, then $a_k \leq 3H\gamma^2\sum_{t=T_r}^{k-1}\theta_t$.

*Proof.* This can be shown using induction by unrolling $a_k \leq (k-T_r)\gamma^2\sum_{t=T_r}^{k-1}(\beta a_t+\theta_t)$ recursively. The proof is omitted. $\qquad\square$

**Lemma 7** (Bound on average consensus violation). Consider Algorithm 1. Let Assumption 1 hold and let $H \geq 1$ be given by Definition 2. Then, for any communication round $r > 0$, for all $T_r \leq k \leq T_{r+1} - 1$ we have

$$\mathbb{E}[\bar{e}_k] \leq \gamma^2(k-T_r)\sum_{t=T_r}^{k-1}\left(\frac{12n^2\nu^2+4B_1^2}{\eta^2}+(6+8B_2^2)L_0^2n^2+16\mathbb{E}[\bar{e}_t]+8B_2^2\mathbb{E}\left[\|\nabla\mathbf{f}^\eta(\bar{x}_t)\|^2\right]\right).$$

Moreover, if $0 < \gamma \leq \frac{1}{4H}$, then

$$\mathbb{E}[\bar{e}_k] \leq 3H^2\gamma^2\left(\frac{12n^2\nu^2+4B_1^2}{\eta^2}+(6+8B_2^2)L_0^2n^2\right)+24B_2^2H\gamma^2\sum_{t=T_r}^{k-1}\mathbb{E}\left[\|\nabla\mathbf{f}^\eta(\bar{x}_t)\|^2\right].$$

*Proof.* In view of Algorithm 1, for any $i$ at any communication round $r > 0$, for all $T_r \leq k \leq T_{r+1}-1$ we have $x_{i,k+1} = x_{i,k} - \gamma(g_{i,k}^\eta + \nabla_{i,k}^{\eta,d})$. Equivalently, we can write

$$x_{i,k} = x_{i,k-1} - \gamma(g_{i,k-1}^\eta + \nabla_{i,k-1}^{\eta,d}), \quad \text{for all } T_r+1\leq k\leq T_{r+1}.$$

This implies that

$$x_{i,k} = x_{i,T_r} - \gamma\sum_{t=T_r}^{k-1}(g_{i,t}^\eta + \nabla_{i,t}^{\eta,d}), \quad \text{for all } T_r+1\leq k\leq T_{r+1}. \tag{9}$$

Again from Algorithm 1, we have $\hat{x}_r = x_{i,T_r}$. From the definition of $\bar{x}_k$, we can write $\bar{x}_{T_r} = \hat{x}_r$. This implies that $\bar{x}_{T_r} = x_{i,T_r}$ for all $i$ and $r$. In view of Lemma 2, we have

$$\bar{x}_k = x_{i,T_r} - \gamma\sum_{t=T_r}^{k-1}(\bar{g}_t^\eta + \bar{\nabla}_t^{\eta,d}), \quad \text{for all } T_r+1\leq k\leq T_{r+1}. \tag{10}$$

Utilizing (9) and (10), for all $T_r+1\leq k\leq T_{r+1}$ we have

$$\mathbb{E}[\bar{e}_k \mid \mathcal{F}_{T_r}] = \frac{1}{m}\sum_{i=1}^m \mathbb{E}\left[\|x_{i,k}-\bar{x}_k\|^2 \mid \mathcal{F}_{T_r}\right]$$

$$= \frac{1}{m}\sum_{i=1}^m \mathbb{E}\left[\left\|\gamma\sum_{t=T_r}^{k-1}(g_{i,t}^\eta+\nabla_{i,t}^{\eta,d})-\gamma\sum_{t=T_r}^{k-1}(\bar{g}_t^\eta+\bar{\nabla}_t^{\eta,d})\right\|^2 \mid \mathcal{F}_{T_r}\right]$$

$$\leq \frac{\gamma^2(k-T_r)}{m}\sum_{i=1}^m\sum_{t=T_r}^{k-1}\mathbb{E}\left[\left\|(g_{i,t}^\eta+\nabla_{i,t}^{\eta,d})-(\bar{g}_t^\eta+\bar{\nabla}_t^{\eta,d})\right\|^2 \mid \mathcal{F}_{T_r}\right],$$

where the preceding relation is implied by the inequality $\|\sum_{t=1}^T y_t\|^2 \leq T\sum_{t=1}^T\|y_t\|^2$ for any $y_t \in \mathbb{R}^n$ for $t \in [T]$. We have

$$\mathbb{E}[\bar{e}_k \mid \mathcal{F}_{T_r}] \leq \frac{\gamma^2(k-T_r)}{m}\sum_{t=T_r}^{k-1}\sum_{i=1}^m \mathbb{E}\left[\left\|(g_{i,t}^\eta+\nabla_{i,t}^{\eta,d})-(\bar{g}_t^\eta+\bar{\nabla}_t^{\eta,d})\right\|^2 \mid \mathcal{F}_{T_r}\right]$$

$$= \frac{\gamma^2(k-T_r)}{m}\sum_{t=T_r}^{k-1}\sum_{i=1}^m\left(\mathbb{E}\left[\left\|g_{i,t}^\eta+\nabla_{i,t}^{\eta,d}\right\|^2 \mid \mathcal{F}_{T_r}\right]+\mathbb{E}\left[\left\|\bar{g}_t^\eta+\bar{\nabla}_t^{\eta,d}\right\|^2 \mid \mathcal{F}_{T_r}\right]\right)$$

$$-2\frac{\gamma^2(k-T_r)}{m}\sum_{t=T_r}^{k-1}\sum_{i=1}^m \mathbb{E}\left[(g_{i,t}^\eta+\nabla_{i,t}^{\eta,d})^T(\bar{g}_t^\eta+\bar{\nabla}_t^{\eta,d}) \mid \mathcal{F}_{T_r}\right],$$

Observing that

$$\frac{1}{m} \sum_{i=1}^{m} \mathbb{E}\left[(g_{i,t}^{\eta} + \nabla_{i,t}^{\eta,d})^T (\bar{g}_t^{\eta} + \bar{\nabla}_t^{\eta,d}) \mid \mathcal{F}_{T_r}\right] = \mathbb{E}\left[\left\|\bar{g}_t^{\eta} + \bar{\nabla}_t^{\eta,d}\right\|^2 \mid \mathcal{F}_{T_r}\right],$$

we obtain

$$\mathbb{E}\left[\bar{e}_k \mid \mathcal{F}_{T_r}\right] \leq \frac{\gamma^2 (k - T_r)}{m} \sum_{t=T_r}^{k-1} \sum_{i=1}^{m} \mathbb{E}\left[\left\|g_{i,t}^{\eta} + \nabla_{i,t}^{\eta,d}\right\|^2 \mid \mathcal{F}_{T_r}\right].$$

From the law of total expectation, for any $T_r \leq t \leq k - 1$, we can write

$$\mathbb{E}\left[\left\|g_{i,t}^{\eta} + \nabla_{i,t}^{\eta,d}\right\|^2 \mid \mathcal{F}_{T_r}\right] = \mathbb{E}\left[\mathbb{E}\left[\left\|g_{i,t}^{\eta} + \nabla_{i,t}^{\eta,d}\right\|^2 \mid \mathcal{F}_{T_r} \cup \left(\cup_{i=1}^{m} \cup_{t=T_r}^{t-1} \{\xi_{i,t}, v_{i,t}\}\right)\right]\right]$$

$$= \mathbb{E}\left[\mathbb{E}\left[\left\|g_{i,t}^{\eta} + \nabla_{i,t}^{\eta,d}\right\|^2 \mid \mathcal{F}_t\right]\right]$$

$$\overset{\text{Lemma 3 (ii)}}{\leq} \frac{12 n^2 \nu^2}{\eta^2} + 6 L_0^2 n^2 + 2\|\nabla_{i,t}^{\eta,d}\|^2.$$

From the two preceding relations, we obtain

$$\mathbb{E}\left[\bar{e}_k \mid \mathcal{F}_{T_r}\right] \leq \frac{\gamma^2 (k - T_r)}{m} \sum_{t=T_r}^{k-1} \sum_{i=1}^{m} \left(\frac{12 n^2 \nu^2}{\eta^2} + 6 L_0^2 n^2 + 2\|\nabla_{i,t}^{\eta,d}\|^2\right).$$

Invoking Lemma 5, we obtain

$$\mathbb{E}\left[\bar{e}_k \mid \mathcal{F}_{T_r}\right] \leq \gamma^2 (k - T_r) \sum_{t=T_r}^{k-1} \left(\frac{12 n^2 \nu^2 + 4 B_1^2}{\eta^2} + (6 + 8 B_2^2) L_0^2 n^2 + 16 \bar{e}_t + 8 B_2^2 \|\nabla \mathbf{f}^{\eta}(\bar{x}_t)\|^2\right).$$

Taking expectations on both sides, we obtain the first result. The second bound is obtained by invoking Lemma 6. $\qquad\square$

### A.3 Proof of Proposition 1

*Proof.* (i) Recall from Lemma 1 that each of the local functions $f_i^{\eta}$ is $\frac{L_0 n}{\eta}$-smooth. Also, $d_i^{\eta}$ is $\frac{1}{\eta}$-smooth. As such, $\mathbf{f}^{\eta}$ is $\left(\frac{L_0 n + 1}{\eta}\right)$-smooth. Invoking Lemma 2, we may obtain

$$\mathbf{f}^{\eta}(\bar{x}_{k+1}) \leq \mathbf{f}^{\eta}(\bar{x}_k) - \gamma \nabla \mathbf{f}^{\eta}(\bar{x}_k)^T \left(\bar{g}_k^{\eta} + \bar{\nabla}_{i,k}^{\eta,d}\right) + \frac{L_0 n}{2\eta} \gamma^2 \|\bar{g}_k^{\eta} + \bar{\nabla}_{i,k}^{\eta,d}\|^2.$$

Taking expectations on both sides, we obtain

$$\mathbb{E}\left[\mathbf{f}^{\eta}(\bar{x}_{k+1})\right] \leq \mathbb{E}\left[\mathbf{f}^{\eta}(\bar{x}_k)\right] - \gamma \mathbb{E}\left[\nabla \mathbf{f}^{\eta}(\bar{x}_k)^T \left(\bar{g}_k^{\eta} + \bar{\nabla}_{i,k}^{\eta,d}\right)\right] + \frac{L_0 n}{2\eta} \gamma^2 \mathbb{E}\left[\|\bar{g}_k^{\eta} + \bar{\nabla}_{i,k}^{\eta,d}\|^2\right].$$

Invoking Lemma 4, we obtain

$$\mathbb{E}\left[\mathbf{f}^{\eta}(\bar{x}_{k+1})\right] \leq \mathbb{E}\left[\mathbf{f}^{\eta}(\bar{x}_k)\right] - \gamma \left(\frac{1}{2} \mathbb{E}\left[\|\nabla \mathbf{f}^{\eta}(\bar{x}_k)\|^2\right] - \frac{(L_0 n + 1)^2}{2\eta^2} \mathbb{E}\left[\bar{e}_k\right]\right)$$

$$+ \frac{L_0 n}{2\eta} \gamma^2 \left(\frac{n^2}{m} \left(\frac{6 \nu^2}{\eta^2} + 3 L_0^2\right) + 2 \mathbb{E}\left[\|\nabla \mathbf{f}^{\eta}(\bar{x}_k)\|^2\right] + \frac{2(L_0 n + 1)^2}{\eta^2} \mathbb{E}\left[\bar{e}_k\right]\right).$$

Using $\gamma \leq \frac{\eta}{4 L_0 n}$ and rearranging the terms, we have

$$\frac{\gamma}{4} \mathbb{E}\left[\|\nabla \mathbf{f}^{\eta}(\bar{x}_k)\|^2\right] \leq \mathbb{E}\left[\mathbf{f}^{\eta}(\bar{x}_k)\right] - \mathbb{E}\left[\mathbf{f}^{\eta}(\bar{x}_{k+1})\right] + \frac{\gamma^2 L_0 n^3}{2\eta m} \left(\frac{6 \nu^2}{\eta^2} + 3 L_0^2\right) + \frac{3 \gamma (L_0 n + 1)^2}{4\eta^2} \mathbb{E}\left[\bar{e}_k\right].$$

Summing both sides over $k = 1 = 0, \ldots, K$, then dividing both sides by $\frac{\gamma(K+1)}{4}$, and using the definition of $k^*$, we obtain

$$\mathbb{E}\left[\|\nabla \mathbf{f}^{\eta}(\bar{x}_{k^*})\|^2\right] \leq \frac{4(\mathbb{E}[\mathbf{f}^{\eta}(\bar{x}_0)] - \mathbb{E}[\mathbf{f}^{\eta}(\bar{x}_{K+1})])}{\gamma(K+1)} + \frac{2\gamma L_0 n^3}{\eta m} \left(\frac{6 \nu^2}{\eta^2} + 3 L_0^2\right) + \frac{3(L_0 n + 1)^2}{\eta^2 (K+1)} \sum_{k=0}^{K} \mathbb{E}\left[\bar{e}_k\right].$$

From Lemma 7 and the definition of $k^*$, we obtain, we have

$$\frac{1}{K+1} \sum_{k=0}^{K} \mathbb{E}\left[\bar{e}_k\right] \leq 3 H^2 \gamma^2 \left(\frac{12 n^2 \nu^2 + 4 B_1^2}{\eta^2} + (6 + 8 B_2^2) L_0^2 n^2\right)$$

$$+ 24 B_2^2 H \gamma^2 \frac{1}{K+1} \sum_{k=0}^{K} \sum_{t=T_r}^{k-1} \mathbb{E}\left[\|\nabla \mathbf{f}^{\eta}(\bar{x}_t)\|^2\right]$$

$$\leq 3 H^2 \gamma^2 \left(\frac{12 n^2 \nu^2 + 4 B_1^2}{\eta^2} + (6 + 8 B_2^2) L_0^2 n^2\right) + 24 B_2^2 H^2 \gamma^2 \mathbb{E}\left[\|\nabla \mathbf{f}^{\eta}(\bar{x}_{k^*})\|^2\right],$$

where in the preceding relation, we used

$$\sum_{k=0}^{K}\sum_{t=T_r}^{k-1}\mathbb{E}\left[\|\nabla\mathbf{f}^{\eta}(\bar{x}_t)\|^2\right] \le H\sum_{k=0}^{K}\mathbb{E}\left[\|\nabla\mathbf{f}^{\eta}(\bar{x}_k)\|^2\right].$$

Thus, invoking $\gamma \le \frac{\eta}{12\sqrt{3}B_2(L_0 n+1)H}$, from the preceding relations we obtain

$$\mathbb{E}\left[\|\nabla\mathbf{f}^{\eta}(\bar{x}_{k^*})\|^2\right] \le \frac{4(\mathbb{E}[\mathbf{f}^{\eta}(\bar{x}_0)]-\mathbb{E}[\mathbf{f}^{\eta,*}])}{\gamma(K+1)} + \frac{2\gamma L_0 n^3}{\eta m}\left(\frac{6\nu^2}{\eta^2}+3L_0^2\right)$$
$$+ \frac{9H^2\gamma^2(L_0 n+1)^2}{\eta^2}\left(\frac{12n^2\nu^2+4B_1^2}{\eta^2}+(6+8B_2^2)L_0^2 n^2\right)+0.5\mathbb{E}\left[\|\nabla\mathbf{f}^{\eta}(\bar{x}_{k^*})\|^2\right].$$

Rearranging the terms, we obtain the inequality in part (i).

(ii) Substituting $\gamma := \sqrt{\frac{m}{K}}$ and $H := \sqrt[4]{\frac{K}{m^3}}$ in the preceding bound, we obtain

$$\mathbb{E}\left[\|\nabla\mathbf{f}^{\eta}(\bar{x}_{k^*})\|^2\right] \le \frac{8(\mathbb{E}[\mathbf{f}^{\eta}(\bar{x}_0)]-\mathbf{f}^{\eta,*})+\frac{12L_0 n^3}{\eta}\left(\frac{2\nu^2}{\eta^2}+L_0^2\right)+\frac{36(L_0 n+1)^2}{\eta^2}\left(\frac{6n^2\nu^2+2B_1^2}{\eta^2}+(3+4B_2^2)L_0^2 n^2\right)}{\sqrt{mK}}.$$

This leads to iteration complexity of $\mathcal{O}\left(\left(\frac{L_0 n^3\nu^2}{\eta^3}+\frac{L_0^3 n^3}{\eta}+\frac{L_0^2 n^4\nu^2}{\eta^4}+\frac{L_0^2 n^2 B_1^2}{\eta^4}+\frac{B_2^2 L_0^4 n^4}{\eta^2}\right)^2\frac{1}{m\epsilon^2}\right).$

(iii) From the choice of $H$ is (ii), we obtain $R=\mathcal{O}(\frac{K}{H})=\mathcal{O}\left(\frac{K}{\sqrt[4]{K/m^3}}\right)=\mathcal{O}\left((mK)^{3/4}\right).$

$\square$

**Proof of Proposition 2.**

*Proof.* (i) The proof for this part follows from Proposition 2.2 in [33].

(ii) From $\nabla\mathbf{f}^{\eta}(x)=0$, we have that $\nabla f^{\eta}(x)+\frac{1}{\eta}(x-\mathcal{P}_X(x))=0$. This implies that $\|x-\mathcal{P}_X(x)\| \le \eta\|\nabla f^{\eta}(x)\|$. Next, we obtain a bound as follows.

$$\|\nabla f^{\eta}(x)\|^2 \le \frac{1}{m}\sum_{i=1}^{m}\|\nabla f_i^{\eta}(x)\|^2 = \left(\frac{n^2}{\eta^2}\right)\frac{1}{m}\sum_{i=1}^{m}\left\|\mathbb{E}_{v_i\in\eta\mathbb{S}}[f_i(x+v_i)\frac{v_i}{\|v_i\|}]\right\|^2$$
$$= \left(\frac{n^2}{\eta^2}\right)\frac{1}{m}\sum_{i=1}^{m}\left\|\mathbb{E}[(f_i(x+v)-f_i(x))\frac{v_i}{\|v_i\|}]\right\|^2$$
$$\overset{\text{Jensen's ineq.}}{\le}\left(\frac{n^2}{\eta^2}\right)\frac{1}{m}\sum_{i=1}^{m}\mathbb{E}[|f_i(x+v)-f_i(x))|^2]$$
$$\overset{\text{Assumption 1 (i)}}{\le}\left(\frac{n^2}{\eta^2}\right)\frac{1}{m}\sum_{i=1}^{m}\mathbb{E}[L_0^2\|v_i\|^2] = n^2 L_0^2.$$

Thus, the infeasibility of $x$ is bounded as $\|x-\mathcal{P}_X(x)\| \le \eta n L_0$. Recall that the $\delta$-Clarke generalized gradient of $\mathbb{I}_X$ at $x$ is defined as

$$\partial_{\delta}\mathbb{I}_X(x) \triangleq \text{conv}\{\zeta : \zeta\in\mathcal{N}_X(y), \|x-y\|\le\delta\},$$

where $\mathcal{N}_X(\bullet)$ denotes the normal cone of $X$. In view of $\|x-\mathcal{P}_X(x)\| \le \eta n L_0$, for $y:=\mathcal{P}_X(x)$ and $\eta \le \frac{\delta}{\max\{2,nL_0\}}$, we have $\|x-y\|\le\delta$. Next we show that for $\zeta:=\frac{1}{\eta}(x-\mathcal{P}_X(x))$ we have $\zeta\in\mathcal{N}_X(y)$. From the projection theorem, we may write

$$(x-\mathcal{P}_X(x))^T(\mathcal{P}_X(x)-z)\ge 0, \quad \text{for all } z\in X.$$

This implies that $\zeta^T(y-z)\ge 0$ for all $z\in X$. Thus, we have $\zeta\in\mathcal{N}_X(y)$ which implies $\frac{1}{\eta}(x-\mathcal{P}_X(x))\in\partial_{\delta}\mathbb{I}(x)$. From (i) and that $2\eta\le\delta$, we have $\nabla f^{\eta}(x)\in\partial_{\delta}f(x)$. Adding the preceding relations and invoking $\nabla\mathbf{f}^{\eta}(x)=0$, we obtain $0\in\partial_{\delta}(f+\mathbb{I}_X)(x)$.

$\square$

**Remark 5.** We note that the approximate Clarke stationary point is also referred to as Goldstein stationary point. (e.g. [32])

# B  Proof of Theorem 1

Here, we prove Theorem 1 and provide some preliminary lemmas and their proofs.

## B.1  Notation in proofs

**Definition 5.** *Let $x_{i,k}$ be given by Algorithm 2 for all $i \in [m]$ and $k \geq 0$. Let $d_i(x), \nabla^{\eta,d}_{i,k}, \bar{\nabla}^{\eta,d}_k, \bar{x}_k$, and $\bar{e}_k$ be given by Definition 4. Let us define an average delay term as $\hat{e}_k \triangleq \frac{\sum_{i=1}^m \|x_{i,k} - \hat{x}_k\|^2}{m}$.*

**Definition 6.** *Let us define the following terms.*

$$g^{\eta}_{i,k} \triangleq \frac{n}{\eta^2} \left( \tilde{f}_i(x_{i,k} + v_{T_r}, y(x_{i,k} + v_{T_r}), \xi_{i,k}) - \tilde{f}_i(x_{i,k}, y(x_{i,k}), \xi_{i,k}) \right) v_{T_r},$$

$$\hat{g}^{\eta}_{i,k} \triangleq \frac{n}{\eta^2} \left( \tilde{f}_i(x_{i,k} + v_{T_r}, y(\hat{x}_r + v_{T_r}), \xi_{i,k}) - \tilde{f}_i(x_{i,k}, y(\hat{x}_r), \xi_{i,k}) \right) v_{T_r},$$

$$g^{\eta,\varepsilon_r}_{i,k} \triangleq \frac{n}{\eta^2} \left( \tilde{f}_i(x_{i,k} + v_{T_r}, y_{\varepsilon_r}(\hat{x}_r + v_{T_r}), \xi_{i,k}) - \tilde{f}_i(x_{i,k}, y_{\varepsilon_r}(\hat{x}_r), \xi_{i,k}) \right) v_{T_r},$$

$$\omega^{\eta}_{i,k} \triangleq \hat{g}^{\eta}_{i,k} - g^{\eta}_{i,k}, \qquad w^{\eta}_{i,k} \triangleq g^{\eta,\varepsilon_r}_{i,k} - \hat{g}^{\eta}_{i,k}, \qquad \bar{g}^{\eta}_k \triangleq \frac{1}{m} \sum_{i=1}^m g^{\eta}_{i,k}, \qquad \bar{g}^{\eta,\varepsilon_r}_k \triangleq \frac{1}{m} \sum_{i=1}^m g^{\eta,\varepsilon_r}_{i,k}$$

$$\bar{\omega}^{\eta}_k \triangleq \frac{1}{m} \sum_{i=1}^m \omega^{\eta}_{i,k}, \qquad \bar{w}^{\eta}_k \triangleq \frac{1}{m} \sum_{i=1}^m w^{\eta}_{i,k}.$$

**Remark 6.** *In view of Definition 6, we have $g^{\eta,\varepsilon_r}_{i,k} = g^{\eta}_{i,k} + \omega^{\eta}_{i,k} + w^{\eta}_{i,k}$. The term $g^{\eta}_{i,k}$ denotes a zeroth-order stochastic gradient of the local implicit objective of client $i$ at iteration $k$, $\hat{g}^{\eta}_{i,k}$ denotes a variant of $g^{\eta}_{i,k}$ where $y(\bullet)$ are obtained at delayed updates, and $g^{\eta,\varepsilon_r}_{i,k}$ denotes the inexact variant of $g^{\eta}_{i,k}$ where $y(\bullet)$ is only inexactly evaluated with prescribed accuracy $\varepsilon_r$. While in Algorithm 2, only $g^{\eta,\varepsilon_r}_{i,k}$ is employed at the local steps, we utilize the equations $g^{\eta,\varepsilon_r}_{i,k} = g^{\eta}_{i,k} + \omega^{\eta}_{i,k} + w^{\eta}_{i,k}$ and $\bar{g}^{\eta,\varepsilon_r}_k = \bar{g}^{\eta}_k + \bar{\omega}^{\eta}_k + \bar{w}^{\eta}_k$ to analyze the method.*

We define the history of Algorithm 2, for $T_r \leq k \leq T_{r+1} - 1$ and $r \geq 1$ as

$$\mathcal{F}_k \triangleq \left( \cup_{i=1}^m \cup_{t=0}^{k-1} \{\xi_{i,t}\} \right) \cup \left( \cup_{j=0}^{r-1} \{v_{T_j}\} \right) \cup \left( \cup_{j=0}^r \mathcal{F}^2_j \right) \cup \{\hat{x}_0\},$$

and for $1 \leq k \leq T_1 - 1$ as $\mathcal{F}_k \triangleq \left( \cup_{i=1}^m \cup_{t=0}^{k-1} \{\xi_{i,t}\} \right) \cup \{v_{T_0}\} \cup \mathcal{F}^2_0 \cup \{\hat{x}_0\}$, and $\mathcal{F}_0 \triangleq \{\hat{x}_0\}$. Here, $\mathcal{F}^2_j$ denotes the collection of all random variables generated in the two calls to the lower-level FL method (e.g., FedAvg) during the $j$th round of Algorithm 2.

## B.2  Lipschitz continuity of the implicit function

As mentioned in Remark 2, the results in Theorem 1 are characterized in terms of the Lipschitz continuity parameter of the random local implicit functions, denoted by $L^{\text{imp}}_0(\xi_i)$. One may wonder whether the local implicit functions are indeed Lipschitz continuous and whether $L^{\text{imp}}_0(\xi_i)$ can be obtained in terms of the problem parameters in ($\textbf{FL}_{bl}$). In the following lemma, we address both of these questions.

**Lemma 8** (Properties of the implicit function). Consider the implicit function $f(x)$ and mapping $y(x)$ given by ($\textbf{FL}_{bl}$). Let Assumption 2 hold. Then, the following statements hold.

(i) The mapping $y(\bullet)$ is $\left( \frac{L^{\nabla h}_{0,x}}{\mu_h} \right)$-Lipschitz continuous, i.e., for any $x_1, x_2 \in \mathbb{R}^n$, we have

$$\|y(x_1) - y(x_2)\| \leq \left( \frac{L^{\nabla h}_{0,x}}{\mu_h} \right) \|x_1 - x_2\|.$$

(ii) The random implicit function is Lipschitz continuous with constant $L^{\text{imp}}_0(\xi_i) := \frac{L^f_{0,y}(\xi_i) L^{\nabla h}_{0,x}}{\mu_h} + L^f_{0,x}(\xi_i)$, i.e., for any $x_1, x_2 \in \mathbb{R}^n$

$$|\tilde{f}_i(x_1, y(x_1), \xi_i) - \tilde{f}_i(x_2, y(x_2), \xi_i)| \leq \left( \frac{L^f_{0,y}(\xi_i) L^{\nabla h}_{0,x}}{\mu_h} + L^f_{0,x}(\xi_i) \right) \|x_1 - x_2\|.$$

(iii) The implicit function is Lipschitz continuous with parameter $l_0^{\text{imp}} := \frac{L_{0,y}^f L_{0,x}^{\nabla h}}{\mu_h} + L_{0,x}^f$, i.e., for any $x_1, x_2 \in \mathbb{R}^n$

$$|f(x_1) - f(x_2)| \leq \left(\frac{L_{0,y}^f L_{0,x}^{\nabla h}}{\mu_h} + L_{0,x}^f\right) \|x_1 - x_2\|.$$

*Proof.* (i) From the strong convexity of $h(x, \bullet)$, we can write for any $x \in \mathbb{R}^n$ and all $y_1, y_2 \in \mathbb{R}^{\tilde{n}}$,

$$\mu_h \|y_1 - y_2\|^2 \leq (y_1 - y_2)^T \left(\nabla_y h(x, y_1) - \nabla_y h(x, y_2)\right). \tag{11}$$

Substituting $x := x_1$, $y_1 := y(x_1)$, and $y_2 := y(x_1)$ in (11), we obtain

$$\mu_h \|y(x_1) - y(x_2)\|^2 \leq (y(x_1) - y(x_2))^T \left(\nabla_y h(x_1, y(x_1)) - \nabla_y h(x_1, y(x_2))\right).$$

Similarly, substituting $x := x_2$, $y_1 := y(x_1)$, and $y_2 := y(x_1)$ in (11), we obtain

$$\mu_h \|y(x_1) - y(x_2)\|^2 \leq (y(x_1) - y(x_2))^T \left(\nabla_y h(x_2, y(x_1)) - \nabla_y h(x_2, y(x_2))\right).$$

Adding the preceding two inequalities together, we have

$$2\mu_h \|y(x_1) - y(x_2)\|^2 \leq (y(x_1) - y(x_2))^T \left(\nabla_y h(x_1, y(x_1)) - \nabla_y h(x_1, y(x_2))\right.$$
$$\left. + \nabla_y h(x_2, y(x_1)) - \nabla_y h(x_2, y(x_2))\right).$$

Note that from the definition of $y(\bullet)$, we have $\nabla_y h(x_1, y(x_1)) = \nabla_y h(x_2, y(x_2)) = 0$. As such, from the preceding inequality we have

$$2\mu_h \|y(x_1) - y(x_2)\|^2 \leq (y(x_1) - y(x_2))^T \left(-\nabla_y h(x_1, y(x_1)) - \nabla_y h(x_1, y(x_2))\right.$$
$$\left. + \nabla_y h(x_2, y(x_1)) + \nabla_y h(x_2, y(x_2))\right).$$

Using the Cauchy-Schwarz inequality and the triangle inequality, we obtain

$$2\mu_h \|y(x_1) - y(x_2)\|^2 \leq \|y(x_1) - y(x_2)\| \, \|\nabla_y h(x_2, y(x_1)) - \nabla_y h(x_1, y(x_1))\|$$
$$+ \|y(x_1) - y(x_2)\| \, \|\nabla_y h(x_2, y(x_2)) - \nabla_y h(x_1, y(x_2))\|.$$

If $x_1 = x_2$, the relation in (i) holds. Suppose $x_1 \neq x_2$. Thus, $y(x_1) \neq y(x_2)$. We obtain

$$2\mu_h \|y(x_1) - y(x_2)\| \leq \|\nabla_y h(x_2, y(x_1)) - \nabla_y h(x_1, y(x_1))\|$$
$$+ \|\nabla_y h(x_2, y(x_2)) - \nabla_y h(x_1, y(x_2))\|.$$

From Assumption 2, we obtain

$$2\mu_h \|y(x_1) - y(x_2)\| \leq L_{0,x}^{\nabla h} \|x_1 - x_2\| + L_{0,x}^{\nabla h} \|x_1 - x_2\|.$$

This implies the bound in (i).

(ii) Let $L_0^{\text{imp}}(\xi_i)$ denote the Lipschitz constant of $\tilde{f}_i(x, y(x), \xi_i)$. We have

$$|\tilde{f}_i(x_1, y(x_1), \xi_i) - \tilde{f}_i(x_2, y(x_2), \xi_i)|$$
$$= |\tilde{f}_i(x_1, y(x_1), \xi_i) - \tilde{f}_i(x_1, y(x_2), \xi_i) + \tilde{f}_i(x_1, y(x_2), \xi_i) - \tilde{f}_i(x_2, y(x_2), \xi_i)|$$
$$\leq |\tilde{f}_i(x_1, y(x_1), \xi_i) - \tilde{f}_i(x_1, y(x_2), \xi_i)| + |\tilde{f}_i(x_1, y(x_2), \xi_i) - \tilde{f}_i(x_2, y(x_2), \xi_i)|$$
$$\leq L_{0,y}^f(\xi_i) \|y(x_1) - y(x_2)\| + L_{0,x}^f(\xi_i) \|x_1 - x_2\|$$
$$\leq \left(\frac{L_{0,y}^f(\xi_i) L_{0,x}^{\nabla h}}{\mu_h} + L_{0,x}^f(\xi_i)\right) \|x_1 - x_2\|$$
$$= L_0^{\text{imp}}(\xi_i) \|x_1 - x_2\|.$$

(iii) First, we show that for any $x_1, x_2, y$, we have $|f_i(x_1, y) - f_i(x_2, y)| \leq L_{0,x}^f \|x_1 - x_2\|$. Also, for any $x, y_1, y_2$, we have $|f_i(x, y_1) - f_i(x, y_2)| \leq L_{0,y}^f \|y_1 - y_2\|$. From Assumption 2, we have

$$|f_i(x_1, y) - f_i(x_2, y)| = |\mathbb{E}_{\xi_i}[\tilde{f}_i(x_1, y, \xi_i) - \tilde{f}_i(x_2, y, \xi_i)]|$$
$$\leq \mathbb{E}_{\xi_i}[|\tilde{f}_i(x_1, y, \xi_i) - \tilde{f}_i(x_2, y, \xi_i)|]$$
$$\leq \mathbb{E}_{\xi_i}[L_{0,x}^f(\xi_i)] \|x_1 - x_2\|.$$

From the Jensen's inequality, we have $(\mathbb{E}_{\xi_i}[L_{0,x}^f(\xi_i)])^2 \leq \mathbb{E}_{\xi_i}[L_{0,x}^f{}^2(\xi_i)]$. Therefore,

$$\mathbb{E}_{\xi_i}[L_{0,x}^f(\xi_i)] \leq |\mathbb{E}_{\xi_i}[L_{0,x}^f(\xi_i)]| \leq \sqrt{\mathbb{E}_{\xi_i}[(L_{0,x}^f(\xi_i))^2]} \leq L_{0,x}^f.$$

From the preceding two relations, we obtain

$$|f_i(x_1, y) - f_i(x_2, y)| \leq L_{0,x}^f \|x_1 - x_2\|. \tag{12}$$

Similarly, we obtain

$$|f_i(x, y_1) - f_i(x, y_2)| \leq L_{0,y}^f \|y_1 - y_2\|. \tag{13}$$

Next, by invoking Assumption 2, we obtain

$$
\begin{aligned}
|f(x_1) - f(x_2)| &= |f(x_1, y(x_1)) - f(x_2, y(x_2))| \\
&= |f(x_1, y(x_1)) - f(x_1, y(x_2)) + f(x_1, y(x_2)) - f(x_2, y(x_2))| \\
&\leq |f(x_1, y(x_1)) - f(x_1, y(x_2))| + |f(x_1, y(x_2)) - f(x_2, y(x_2))| \\
&\leq \frac{1}{m} \sum_{i=1}^m |f_i(x_1, y(x_1)) - f_i(x_1, y(x_2))| + \frac{1}{m} \sum_{i=1}^m |f_i(x_1, y(x_2)) - f_i(x_2, y(x_2))| \\
&\overset{(12),(13)}{\leq} L_{0,y}^f \|y(x_1) - y(x_2)\| + L_{0,x}^f \|x_1 - x_2\|.
\end{aligned}
$$

The bound in (iii) is obtained by invoking the bound in (i).

$\square$

**Remark 7.** We note that the result in Lemma 8 (i) has been studied in a more general setting in prior work, e.g., see Lemma 2.2 in [6].

### B.3  Preliminary lemmas

In the following, we analyze the error terms in Definition 6.

**Lemma 9.** Let Assumption 2 hold. Consider Algorithm 2 and Definition 6. Then, the following statements hold in an almost-sure sense for all $k \geq 0$ and $i \in [m]$.

(i) $\mathbb{E}\left[g_{i,k}^\eta \mid \mathcal{F}_k\right] = \nabla f_i^\eta(x_{i,k})$.

(ii) $\mathbb{E}\left[\|g_{i,k}^\eta\|^2\right] \leq n^2 (L_0^{\text{imp}})^2$.

(iii) $\mathbb{E}\left[\|\omega_{i,k}^\eta\|^2\right] \leq \frac{4n^2}{\eta^2} \left(\frac{L_{0,y}^f L_{0,x}^{\nabla h}}{\mu_h}\right)^2 \mathbb{E}\left[\|x_{i,k} - \hat{x}_r\|^2\right]$.

(iv) $\mathbb{E}\left[\|w_{i,k}^\eta\|^2\right] \leq \frac{4n^2}{\eta^2} \left(L_{0,y}^f\right)^2 \varepsilon_r$.

(v) $\mathbb{E}\left[\|\bar{g}_k^\eta + \bar{\nabla}_k^{\eta,d}\|^2\right] \leq \frac{n^2}{m}(L_0^{\text{imp}})^2 + 2\mathbb{E}\left[\|\nabla \mathbf{f}^\eta(\bar{x}_k)\|^2\right] + \frac{2(L_0^{\text{imp}}n+1)^2}{\eta^2}\mathbb{E}\left[\bar{e}_k\right]$.

(vi) $\mathbb{E}\left[\nabla \mathbf{f}^\eta(\bar{x}_k)^T(\bar{g}_k^\eta + \bar{\nabla}_k^{\eta,d})\right] \geq \frac{1}{2}\mathbb{E}\left[\|\nabla \mathbf{f}^\eta(\bar{x}_k)\|^2\right] - \frac{(L_0^{\text{imp}}n+1)^2}{2\eta^2}\mathbb{E}\left[\bar{e}_k\right]$.

(vii) $\mathbb{E}\left[\|\bar{\omega}_k^\eta\|^2\right] \leq \left(\frac{L_{0,x}^{\nabla h}}{\mu_h}\right)^2 \frac{4n^2}{\eta^2}(L_{0,y}^f)^2 \, \mathbb{E}[\hat{e}_k]$.

(viii) $\mathbb{E}\left[\|\bar{w}_k^\eta\|^2\right] \leq \frac{4n^2}{\eta^2}(L_{0,y}^f)^2 \varepsilon_r$.

*Proof.* (i) From Definition 6, we can write

$$
\mathbb{E}\left[g_{i,k}^\eta \mid \mathcal{F}_k\right]
$$

$$
= \mathbb{E}_{v_{T_r}}\left[\mathbb{E}_{\xi_{i,k}}\left[\frac{n}{\eta^2}\left(\tilde{f}_i(x_{i,k} + v_{T_r}, y(x_{i,k} + v_{T_r}), \xi_{i,k}) - \tilde{f}_i(x_{i,k}, y(x_{i,k}), \xi_{i,k})\right)v_{T_r} \mid \mathcal{F}_k \cup \{v_{T_r}\}\right]\right]
$$

$$
= \mathbb{E}_{v_{T_r}}\left[\frac{n}{\eta^2}f_i(x_{i,k} + v_{T_r}, y(x_{i,k} + v_{T_r}))v_{T_r} \mid \mathcal{F}_k\right] \overset{\text{Lemma 1 (i)}}{=} \nabla f_i^\eta(x_{i,k}).
$$

(ii) From the definition of $g^\eta_{i,k}$ and that $\|v_{T_r}\| = \eta$, we have

$$\mathbb{E}_{v_{T_r}}\left[\left\|g^\eta_{i,k}\right\|^2 \mid \mathcal{F}_k \cup \{\xi_{i,k}\}\right]$$

$$= \mathbb{E}_{v_{T_r}}\left[\left\|\tfrac{n}{\eta^2}\left(\tilde{f}_i(x_{i,k}+v_{T_r},y(x_{i,k}+v_{T_r}),\xi_{i,k}) - \tilde{f}_i(x_{i,k},y(x_{i,k}),\xi_{i,k})\right)v_{T_r}\right\|^2 \mid \mathcal{F}_k \cup \{\xi_{i,k}\}\right]$$

$$= \left(\tfrac{n}{\eta^2}\right)^2 \int_{\eta\mathbb{S}}\left\|\left(\tilde{f}_i(x_{i,k}+v_{T_r},y(x_{i,k}+v_{T_r}),\xi_{i,k}) - \tilde{f}_i(x_{i,k},y(x_{i,k}),\xi_{i,k})\right)v_{T_r}\right\|^2 p_v(v_{T_r})\,dv_{T_r}$$

$$= \left(\tfrac{n}{\eta^2}\right)^2 \int_{\eta\mathbb{S}}\left|\tilde{f}_i(x_{i,k}+v_{T_r},y(x_{i,k}+v_{T_r}),\xi_{i,k}) - \tilde{f}_i(x_{i,k},y(x_{i,k}),\xi_{i,k})\right|^2 \|v_{T_r}\|^2 p_v(v_{T_r})\,dv_{T_r}$$

$$\overset{\text{Assumption 2, Remark 2 (iii)}}{\leq} \left(\tfrac{n}{\eta^2}\right)^2 \int_{\eta\mathbb{S}}\left((L^{\text{imp}}_0(\xi_{i,k}))^2 \|x_{i,k}+v_{T_r}-x_{i,k}\|^2\right)\|v_{T_r}\|^2 p_v(v_{T_r})\,dv_{T_r}$$

$$\leq n^2(L^{\text{imp}}_0(\xi_{i,k}))^2 \int_{\eta\mathbb{S}} p_v(v_{T_r})\,dv_{T_r}$$

$$= n^2(L^{\text{imp}}_0(\xi_{i,k}))^2.$$

Taking expectations with respect to $\xi_{i,k}$ on both sides and invoking $L^{\text{imp}}_0 := \max_{i=1,\ldots,m}\sqrt{\mathbb{E}[(L^{\text{imp}}_0(\xi_i))^2]}$, we get $\mathbb{E}\left[\|g^\eta_{i,k}\|^2 \mid \mathcal{F}_k\right] \leq (L^{\text{imp}}_0)^2 n^2$. Then, taking expectations on both sides of the preceding inequality, we obtain $\mathbb{E}\left[\|g^\eta_{i,k}\|^2\right] \leq (L^{\text{imp}}_0)^2 n^2$.

(iii) Consider the definition of $\omega^\eta_{i,k}$. We can write

$$\mathbb{E}\left[\|\omega^\eta_{i,k}\|^2 \mid \mathcal{F}_k \cup \{\xi_{i,k}\}\right]$$

$$= \tfrac{n^2}{\eta^4}\mathbb{E}\Big[\|\big(\tilde{f}_i(x_{i,k}+v_{T_r},y(\hat{x}_r+v_{T_r}),\xi_{i,k}) - \tilde{f}_i(x_{i,k},y(\hat{x}_r),\xi_{i,k})$$

$$-\tilde{f}_i(x_{i,k}+v_{T_r},y(x_{i,k}+v_{T_r}),\xi_{i,k}) + \tilde{f}_i(x_{i,k},y(x_{i,k}),\xi_{i,k})\big)v_{T_r}\|^2 \mid \mathcal{F}_k \cup \{\xi_{i,k}\}\Big]$$

$$\leq \tfrac{2n^2}{\eta^4}\mathbb{E}\Big[\|(\tilde{f}_i(x_{i,k}+v_{T_r},y(\hat{x}_r+v_{T_r}),\xi_{i,k}) - \tilde{f}_i(x_{i,k}+v_{T_r},y(x_{i,k}+v_{T_r}),\xi_{i,k}))v_{T_r}\|^2 \mid \mathcal{F}_k \cup \{\xi_{i,k}\}\Big]$$

$$+ \tfrac{2n^2}{\eta^4}\mathbb{E}\Big[\|(-\tilde{f}_i(x_{i,k},y(\hat{x}_r),\xi_{i,k}) + \tilde{f}_i(x_{i,k},y(x_{i,k}),\xi_{i,k}))v_{T_r}\|^2 \mid \mathcal{F}_k \cup \{\xi_{i,k}\}\Big]. \tag{14}$$

Now consider the second term in the preceding relation. We have

$$\mathbb{E}\left[\|(-\tilde{f}_i(x_{i,k},y(\hat{x}_r),\xi_{i,k}) + \tilde{f}_i(x_{i,k},y(x_{i,k}),\xi_{i,k}))v_{T_r}\|^2 \mid \mathcal{F}_k \cup \{\xi_{i,k}\}\right]$$

$$= \int_{\eta\mathbb{S}}\frac{|\tilde{f}_i(x_{i,k},y(\hat{x}_r),\xi_{i,k})-\tilde{f}_i(x_{i,k},y(x_{i,k}),\xi_{i,k})|^2}{\|y(x_{i,k})-y(\hat{x}_r)\|^2}\|y(x_{i,k})-y(\hat{x}_r)\|^2\|v_{T_r}\|^2 p_v(v_{T_r})\,dv_{T_r}$$

$$\overset{\text{Assumption 2}}{\leq} (L^f_{0,y}(\xi_{i,k}))^2 \int_{\eta\mathbb{S}}\|y(x_{i,k})-y(\hat{x}_r)\|^2\|v_{T_r}\|^2 p_v(v_{T_r})\,dv_{T_r}$$

$$= \eta^2(L^f_{0,y}(\xi_{i,k}))^2\|y(x_{i,k})-y(\hat{x}_r)\|^2$$

$$\overset{\text{Lemma 8 (i)}}{\leq} \left(\frac{\eta L^f_{0,y}(\xi_{i,k})L^{\nabla h}_{0,x}}{\mu_h}\right)^2 \|x_{i,k}-\hat{x}_r\|^2.$$

We can obtain a similar bound on the first term in (14). We obtain

$$\mathbb{E}\left[\|\omega^\eta_{i,k}\|^2 \mid \mathcal{F}_k \cup \{\xi_{i,k}\}\right]$$

$$\leq \frac{2n^2}{\eta^2}\left(\frac{L^f_{0,y}(\xi_{i,k})L^{\nabla h}_{0,x}}{\mu_h}\right)^2 \|x_{i,k}-\hat{x}_r\|^2 + \frac{2n^2}{\eta^2}\left(\frac{L^f_{0,y}(\xi_{i,k})L^{\nabla h}_{0,x}}{\mu_h}\right)^2 \|x_{i,k}-\hat{x}_r\|^2$$

$$= \frac{4n^2}{\eta^2}\left(\frac{L^f_{0,y}(\xi_{i,k})L^{\nabla h}_{0,x}}{\mu_h}\right)^2 \|x_{i,k}-\hat{x}_r\|^2.$$

Taking expectations with respect to $\xi_{i,k}$ on the both sides, we get $\mathbb{E}\left[\|\omega_{i,k}^\eta\|^2 \mid \mathcal{F}_k\right] \leq \frac{4n^2}{\eta^2}\left(\frac{L_{0,y}^f L_{0,x}^{\nabla h}}{\mu_h}\right)^2 \mathbb{E}_{\xi_{i,k}}\left[\|x_{i,k} - \hat{x}_r\|^2\right]$. Then, taking expectations on the both sides of the preceding relation, we obtain $\mathbb{E}\left[\|\omega_{i,k}^\eta\|^2\right] \leq \frac{4n^2}{\eta^2}\left(\frac{L_{0,y}^f L_{0,x}^{\nabla h}}{\mu_h}\right)^2 \mathbb{E}\left[\|x_{i,k} - \hat{x}_r\|^2\right]$.

(iv) Consider the definition of $w_{i,k}^\eta$. Then we may bound $\mathbb{E}\left[\|w_{i,k}^\eta\|^2\right]$ as follows.

$$\mathbb{E}\left[\|w_{i,k}^\eta\|^2 \mid \mathcal{F}_k \cup \{\xi_{i,k}\}\right]$$
$$= \frac{n^2}{\eta^4}\mathbb{E}\left[\left\|\left(\tilde{f}_i(x_{i,k} + v_{T_r}, y_{\varepsilon_r}(\hat{x}_r + v_{T_r}), \xi_{i,k}) - \tilde{f}_i(x_{i,k}, y_{\varepsilon_r}(\hat{x}_r), \xi_{i,k})\right.\right.\right.$$
$$\left.\left.\left. - \tilde{f}_i(x_{i,k} + v_{T_r}, y(\hat{x}_r + v_{T_r}), \xi_{i,k}) + \tilde{f}_i(x_{i,k}, y(\hat{x}_r), \xi_{i,k})\right)v_{T_r}\right\|^2 \mid \mathcal{F}_k \cup \{\xi_{i,k}\}\right]$$
$$\leq \frac{2n^2}{\eta^4}\mathbb{E}\left[\|(\tilde{f}_i(x_{i,k} + v_{T_r}, y_{\varepsilon_r}(\hat{x}_r + v_{T_r}), \xi_{i,k}) - \tilde{f}_i(x_{i,k} + v_{T_r}, y(\hat{x}_r + v_{T_r}), \xi_{i,k}))v_{T_r}\|^2 \mid \mathcal{F}_k \cup \{\xi_{i,k}\}\right]$$
$$+ \frac{2n^2}{\eta^4}\mathbb{E}\left[\|(-\tilde{f}_i(x_{i,k}, y_{\varepsilon_r}(\hat{x}_r), \xi_{i,k}) + \tilde{f}_i(x_{i,k}, y(\hat{x}_r), \xi_{i,k}))v_{T_r}\|^2 \mid \mathcal{F}_k \cup \{\xi_{i,k}\}\right]$$
$$\leq \frac{4n^2}{\eta^2}\left(L_{0,y}^f(\xi_{i,k})\right)^2 \varepsilon_r.$$

The last inequality is obtained by following steps similar to those in (iii) and by invoking $\mathbb{E}\left[\|y_{\varepsilon_r}(x) - y(x)\|^2 \mid x\right] \leq \varepsilon_r$. Next, we take expectations with respect to $\xi_{i,k}$ on both sides of the preceding relation. Then, we take expectations w.r.t. $\mathcal{F}_k$ on both sides of the resulting inequality and obtain the desired bound.

(v) Using the definition of $\bar{g}_k^\eta$ in Definition 6, we can write

$$\mathbb{E}\left[\|\bar{g}_k^\eta + \bar{\nabla}_k^{\eta,d}\|^2 \mid \mathcal{F}_k\right] = \mathbb{E}\left[\left\|\frac{1}{m}\sum_{i=1}^m(g_{i,k}^\eta + \nabla_{i,k}^{\eta,d})\right\|^2 \mid \mathcal{F}_k\right]$$
$$= \mathbb{E}\left[\left\|\frac{1}{m}\sum_{i=1}^m\left(g_{i,k}^\eta + \nabla_{i,k}^{\eta,d} - \nabla\mathbf{f}_i^\eta(x_{i,k}) + \nabla\mathbf{f}_i^\eta(x_{i,k})\right)\right\|^2 \mid \mathcal{F}_k\right]$$
$$= \mathbb{E}\left[\left\|\frac{1}{m}\sum_{i=1}^m\left(g_{i,k}^\eta + \nabla_{i,k}^{\eta,d} - (\nabla f_i^\eta(x_{i,k}) + \nabla_{i,k}^{\eta,d})\right)\right\|^2 \mid \mathcal{F}_k\right] + \left\|\frac{1}{m}\sum_{i=1}^m\nabla\mathbf{f}_i^\eta(x_{i,k})\right\|^2$$
$$\overset{\text{Lemma 9 (i)}}{=} \frac{1}{m^2}\sum_{i=1}^m\mathbb{E}\left[\left\|g_{i,k}^\eta - \nabla f_i^\eta(x_{i,k})\right\|^2 \mid \mathcal{F}_k\right]$$
$$+ \left\|\nabla\mathbf{f}^\eta(\bar{x}_k) + \frac{1}{m}\sum_{i=1}^m\left(\nabla\mathbf{f}_i^\eta(x_{i,k}) - \nabla\mathbf{f}_i^\eta(\bar{x}_k)\right)\right\|^2$$
$$\leq \frac{1}{m^2}\sum_{i=1}^m\mathbb{E}\left[\left\|g_{i,k}^\eta\right\|^2 \mid \mathcal{F}_k\right] + 2\|\nabla\mathbf{f}^\eta(\bar{x}_k)\|^2 + 2\left\|\frac{1}{m}\sum_{i=1}^m\left(\nabla\mathbf{f}_i^\eta(x_{i,k}) - \nabla\mathbf{f}_i^\eta(\bar{x}_k)\right)\right\|^2.$$
$$\overset{\text{Lemma 9 (ii)}}{\leq} \frac{n^2(L_0^{\text{imp}})^2}{m} + 2\|\nabla\mathbf{f}^\eta(\bar{x}_k)\|^2 + 2\left\|\frac{1}{m}\sum_{i=1}^m\left(\nabla\mathbf{f}_i^\eta(x_{i,k}) - \nabla\mathbf{f}_i^\eta(\bar{x}_k)\right)\right\|^2.$$

Note that given vectors $y_i \in \mathbb{R}^n$ for $i \in [m]$, we have $\|\frac{1}{m}\sum_{i=1}^m y_i\|^2 \leq \frac{1}{m}\sum_{i=1}^m\|y_i\|^2$. Utilizing this inequality, Lemma 1 (iv), and that $d_i^\eta(\bullet)$ is $\frac{1}{\eta}$-smooth, we obtain

$$\mathbb{E}\left[\|\bar{g}_k^\eta + \bar{\nabla}_k^{\eta,d}\|^2 \mid \mathcal{F}_k\right] \leq \frac{n^2(L_0^{\text{imp}})^2}{m} + 2\|\nabla\mathbf{f}^\eta(\bar{x}_k)\|^2 + \frac{2(L_0^{\text{imp}}n+1)^2}{\eta^2 m}\sum_{i=1}^m\|x_{i,k} - \bar{x}_k\|^2.$$

Recalling the definition of $\bar{e}_k$ and taking expectations on both sides, we obtain the bound in (i).

(vi) We can write

$$\mathbb{E}\left[\nabla \mathbf{f}^\eta(\bar{x}_k)^T(\bar{g}_k^\eta + \bar{\nabla}_k^{\eta,d}) \mid \mathcal{F}_k\right] = \nabla \mathbf{f}^\eta(\bar{x}_k)^T \mathbb{E}\left[\frac{1}{m}\sum_{i=1}^m (g_{i,k}^\eta + \bar{\nabla}_{i,k}^{\eta,d}) \mid \mathcal{F}_k\right]$$

$$\overset{\text{Lemma 9 (i)}}{=} \nabla \mathbf{f}^\eta(\bar{x}_k)^T \left(\frac{1}{m}\sum_{i=1}^m \nabla \mathbf{f}_i^\eta(x_{i,k})\right)$$

$$= \nabla \mathbf{f}^\eta(\bar{x}_k)^T \frac{1}{m}\sum_{i=1}^m \left(\nabla \mathbf{f}_i^\eta(x_{i,k}) - \nabla \mathbf{f}_i^\eta(\bar{x}_k) + \nabla \mathbf{f}_i^\eta(\bar{x}_k)\right)$$

$$= \nabla \mathbf{f}^\eta(\bar{x}_k)^T \frac{1}{m}\sum_{i=1}^m \left(\nabla \mathbf{f}_i^\eta(x_{i,k}) - \nabla \mathbf{f}_i^\eta(\bar{x}_k)\right) + \|\nabla \mathbf{f}^\eta(\bar{x}_k)\|^2$$

$$\geq -\frac{1}{2}\|\nabla \mathbf{f}^\eta(\bar{x}_k)\|^2 - \frac{1}{2}\left\|\frac{1}{m}\sum_{i=1}^m \left(\nabla \mathbf{f}_i^\eta(x_{i,k}) - \nabla \mathbf{f}_i^\eta(\bar{x}_k)\right)\right\|^2 + \|\nabla \mathbf{f}^\eta(\bar{x}_k)\|^2$$

$$\geq \frac{1}{2}\|\nabla \mathbf{f}^\eta(\bar{x}_k)\|^2 - \frac{1}{2m}\sum_{i=1}^m \|\nabla \mathbf{f}_i^\eta(x_{i,k}) - \nabla \mathbf{f}_i^\eta(\bar{x}_k)\|^2$$

$$\overset{\text{Lemma 1 (iv)}}{\geq} \frac{1}{2}\|\nabla \mathbf{f}^\eta(\bar{x}_k)\|^2 - \frac{(L_0^{\text{imp}}n+1)^2}{2m\eta^2}\sum_{i=1}^m \|x_{i,k} - \bar{x}_k\|^2.$$

The bound is obtained by recalling the definition of $\bar{e}_k$ and taking expectations on both sides.

(vii) We have

$$\mathbb{E}\left[\|\bar{\omega}^\eta{}_k\|^2\right] \leq \frac{1}{m^2}\mathbb{E}\left[\|\sum_{i=1}^m \omega_{i,k}\|^2\right] \leq \frac{1}{m}\mathbb{E}\left[\sum_{i=1}^m \|\omega_{i,k}\|^2\right] = \frac{1}{m}\sum_{i=1}^m \mathbb{E}\left[\|\omega_{i,k}\|^2\right]$$

$$\overset{\text{(iii)}}{\leq} \frac{4n^2}{\eta^2}\left(\frac{L_{0,y}^f L_{0,x}^{\nabla h}}{\mu_h}\right)^2 \sum_{i=1}^m \frac{\mathbb{E}[\|x_{i,k} - \hat{x}_r\|^2]}{m}.$$

(viii) We can write

$$\mathbb{E}\left[\|\bar{w}^\eta{}_k\|^2\right] \leq \frac{1}{m^2}\mathbb{E}\left[\|\sum_{i=1}^m w_{i,k}^\eta\|^2\right] \leq \frac{1}{m}\mathbb{E}\left[\sum_{i=1}^m \|w_{i,k}^\eta\|^2\right] = \frac{1}{m}\sum_{i=1}^m \mathbb{E}\left[\|w_{i,k}^\eta\|^2\right]$$

$$\overset{\text{(iv)}}{\leq} \frac{4n^2}{\eta^2}\left(L_{0,y}^f\right)^2 \varepsilon_r.$$

$\square$

Next, we derive an upper bound on the average delay and average consensus violation, after performing $k$ local steps by a client in Algorithm 2. We make use of the following result.

**Lemma 10.** Suppose for $T_r + 1 \leq k \leq T_{r+1}$, for any $r \geq 0$, where $T_0 := 0$, the nonnegative sequences $\{a_k\}$, $\{b_k\}$, and $\{\theta_k\}$ satisfy a recursive relation of the form

$$\max\{a_k, b_k\} \leq (k - T_r)\gamma^2 \sum_{t=T_r}^{k-1} (\beta_1 a_t + \beta_2 b_t + \theta_t),$$

where $a_{T_r} = b_{T_r} = 0$ and $\beta_1, \beta_2 > 0$. If $0 < \gamma \leq \frac{1}{\sqrt{2\max\{\beta_1,\beta_2\}H}}$, then for $T_r + 1 \leq k \leq T_{r+1}$, we have for any $r \geq 0$,

$$a_k + b_k \leq 6H\gamma^2 \sum_{t=T_r}^{k-1} \theta_t.$$

*Proof.* From the given recursive relation, we have for any $k \geq 1$,

$$a_k \leq (k - T_r)\gamma^2 \sum_{t=T_r}^{k-1} (\beta_1 a_t + \beta_2 b_t + \theta_t)$$

and

$$b_k \leq (k - T_r)\gamma^2 \sum_{t=T_r}^{k-1} (\beta_1 a_t + \beta_2 b_t + \theta_t).$$

Summing the preceding inequalities, we have

$$a_k + b_k \leq 2(k - T_r)\gamma^2 \sum_{t=T_r}^{k-1} (\beta_1 a_t + \beta_2 b_t + \theta_t)$$

$$\leq (k - T_r)\gamma^2 \sum_{t=T_r}^{k-1} (2\max\{\beta_1, \beta_2\}(a_t + b_t) + 2\theta_t).$$

If $0 < \gamma \leq \frac{1}{\sqrt{2\max\{\beta_1,\beta_2\}H}}$, by invoking Lemma 6, from the preceding relation we have

$$a_k + b_k \leq 3H\gamma^2 \sum_{t=T_r}^{k-1} 2\theta_t = 6H\gamma^2 \sum_{t=T_r}^{k-1} \theta_t.$$

$\square$

**Lemma 11** (Bounds on average delay and average consensus violation). Let Assumption 2 hold. Consider Algorithm 2. The following holds.

(i) [Recursive bound] For any communication round $r > 0$, for all $T_r \leq k \leq T_{r+1} - 1$, we have

$$\max\left\{\mathbb{E}\left[\bar{e}_k\right], \mathbb{E}\left[\hat{e}_k\right]\right\} \leq 4\gamma^2(k - T_r) \sum_{t=T_r}^{k-1} \left( 8\mathbb{E}\left[\bar{e}_t\right] + \left(\frac{L_{0,x}^{\nabla h}}{\mu_h}\right)^2 \frac{4n^2}{\eta^2}(L_{0,y}^f)^2 \mathbb{E}[\hat{e}_k] \right.$$

$$+ 4B_2^2 \mathbb{E}\left[\|\nabla \mathbf{f}^\eta(\bar{x}_t)\|^2\right] + n^2 (L_0^{\mathrm{imp}})^2$$

$$\left. + \frac{2B_1^2}{\eta^2} + 4B_2^2(L_0^{\mathrm{imp}})^2 n^2 + \frac{4n^2}{\eta^2}(L_{0,y}^f)^2 \varepsilon_r \right).$$

(ii) [Non-recursive bound] Let $0 < \gamma \leq \left(4H \max\left\{2, \left(\frac{L_{0,x}^{\nabla h}}{\mu_h}\right) \frac{\sqrt{2}n}{\eta} L_{0,y}^f\right\}\right)^{-1}$. Then, for any $r > 0$, for all $T_r \leq k \leq T_{r+1} - 1$, we have

$$\mathbb{E}\left[\bar{e}_k\right] + \mathbb{E}\left[\hat{e}_k\right] \leq 96B_2^2 H\gamma^2 \sum_{t=T_r}^{k-1} \mathbb{E}\left[\|\nabla \mathbf{f}^\eta(\bar{x}_t)\|^2\right]$$

$$+ 24H^2\gamma^2 \left(\frac{2B_1^2}{\eta^2} + (4B_2^2 + 1)(L_0^{\mathrm{imp}})^2 n^2 + \frac{4n^2}{\eta^2}(L_{0,y}^f)^2 \varepsilon_r\right).$$

*Proof.* (i) This proof is comprised of two steps. In the first step we show that the required bound holds for $\mathbb{E}\left[\bar{e}_k\right]$. Then, in the second step we show that it holds for $\mathbb{E}\left[\hat{e}_k\right]$ as well.

(Step 1) In view of Algorithm 2, for any $i$ at any communication round $r > 0$, for all $T_r \leq k \leq T_{r+1} - 1$ we have $x_{i,k+1} = x_{i,k} - \gamma(g_{i,k}^{\eta,\varepsilon_r} + \nabla_{i,k}^{\eta,d})$. This implies that

$$x_{i,k} = x_{i,k-1} - \gamma(g_{i,k-1}^{\eta,\varepsilon_r} + \nabla_{i,k-1}^{\eta,d}), \quad \text{for all } T_r + 1 \leq k \leq T_{r+1}.$$

Unrolling the preceding relation recursively, we obtain

$$x_{i,k} = x_{i,T_r} - \gamma \sum_{t=T_r}^{k-1} (g_{i,t}^{\eta,\varepsilon_r} + \nabla_{i,t}^{\eta,d}), \quad \text{for all } T_r + 1 \leq k \leq T_{r+1}. \tag{15}$$

From Algorithm 2, we have $\hat{x}_r = x_{i,T_r}$. Invoking the definition of $\bar{x}_k$, we have $\bar{x}_{T_r} = \hat{x}_r$. This implies that $\bar{x}_{T_r} = x_{i,T_r}$ for all $i$ and $r$. Extending Lemma 2, we can write

$$\bar{x}_k = x_{i,T_r} - \gamma \sum_{t=T_r}^{k-1} (\bar{g}_t^{\eta,\varepsilon_r} + \bar{\nabla}_t^{\eta,d}), \quad \text{for all } T_r + 1 \leq k \leq T_{r+1}. \tag{16}$$

Using (15) and (16), for all $T_r + 1 \leq k \leq T_{r+1}$ we have

$$\mathbb{E}\left[\bar{e}_k\right] = \frac{1}{m} \sum_{i=1}^m \mathbb{E}\left[\|x_{i,k} - \bar{x}_k\|^2\right]$$

$$= \frac{1}{m} \sum_{i=1}^m \mathbb{E}\left[\left\|\gamma \sum_{t=T_r}^{k-1}(g_{i,t}^{\eta,\varepsilon_r} + \nabla_{i,t}^{\eta,d}) - \gamma \sum_{t=T_r}^{k-1}(\bar{g}_t^{\eta,\varepsilon_r} + \bar{\nabla}_t^{\eta,d})\right\|^2\right]$$

$$\leq \frac{\gamma^2(k-T_r)}{m} \sum_{i=1}^m \sum_{t=T_r}^{k-1} \mathbb{E}\left[\left\|(g_{i,t}^{\eta,\varepsilon_r} + \nabla_{i,t}^{\eta,d}) - (\bar{g}_t^{\eta,\varepsilon_r} + \bar{\nabla}_t^{\eta,d})\right\|^2\right],$$

where the preceding relation follows from the inequality $\|\sum_{t=1}^T y_t\|^2 \leq T \sum_{t=1}^T \|y_t\|^2$ for any $y_t \in \mathbb{R}^n$ for $t \in [T]$. Consequently,

$$\mathbb{E}\left[\bar{e}_k\right] \leq \frac{\gamma^2(k-T_r)}{m} \sum_{t=T_r}^{k-1} \sum_{i=1}^m \mathbb{E}\left[\left\|(g_{i,t}^{\eta,\varepsilon_r} + \nabla_{i,t}^{\eta,d}) - (\bar{g}_t^{\eta,\varepsilon_r} + \bar{\nabla}_t^{\eta,d})\right\|^2\right]$$

$$= \frac{\gamma^2(k-T_r)}{m} \sum_{t=T_r}^{k-1} \sum_{i=1}^m \left(\mathbb{E}\left[\left\|g_{i,t}^{\eta,\varepsilon_r} + \nabla_{i,t}^{\eta,d}\right\|^2\right] + \mathbb{E}\left[\left\|\bar{g}_t^{\eta,\varepsilon_r} + \bar{\nabla}_t^{\eta,d}\right\|^2\right]\right)$$

$$- 2\frac{\gamma^2(k-T_r)}{m} \sum_{t=T_r}^{k-1} \sum_{i=1}^m \mathbb{E}\left[(g_{i,t}^{\eta,\varepsilon_r} + \nabla_{i,t}^{\eta,d})^T(\bar{g}_t^{\eta,\varepsilon_r} + \bar{\nabla}_t^{\eta,d})\right],$$

Observing that

$$\frac{1}{m}\sum_{i=1}^{m}\mathbb{E}\left[(g_{i,t}^{\eta,\varepsilon_r}+\nabla_{i,t}^{\eta,d})^T(\bar{g}_t^{\eta,\varepsilon_r}+\bar{\nabla}_t^{\eta,d}) \mid \mathcal{F}_{T_r}\right] = \mathbb{E}\left[\left\|\bar{g}_t^{\eta,\varepsilon_r}+\bar{\nabla}_t^{\eta,d}\right\|^2\right],$$

we obtain

$$\mathbb{E}\left[\bar{e}_k\right] \leq \frac{\gamma^2(k-T_r)}{m}\sum_{t=T_r}^{k-1}\sum_{i=1}^{m}\mathbb{E}\left[\left\|g_{i,t}^{\eta,\varepsilon_r}+\nabla_{i,t}^{\eta,d}\right\|^2\right]$$

$$\leq \frac{4\gamma^2(k-T_r)}{m}\sum_{t=T_r}^{k-1}\sum_{i=1}^{m}\left(\mathbb{E}\left[\|g_{i,t}^{\eta}\|^2+\|\nabla_{i,t}^{\eta,d}\|^2+\|\omega_{i,t}^{\eta}\|^2+\|w_{i,t}^{\eta}\|^2\right]\right)$$

$$\overset{\text{Lemma 9}}{\leq} \frac{4\gamma^2(k-T_r)}{m}\sum_{t=T_r}^{k-1}\sum_{i=1}^{m}\left(n^2(L_0^{\text{imp}})^2+\|\nabla_{i,t}^{\eta,d}\|^2\right.$$

$$\left.+\left(\frac{L_{0,x}^{\nabla h}}{\mu_h}\right)^2\frac{4n^2}{\eta^2}(L_{0,y}^f)^2\mathbb{E}[\|x_{i,t}-\hat{x}_r\|^2]+\frac{4n^2}{\eta^2}(L_{0,y}^f)^2\varepsilon_r\right).$$

Invoking Lemma 5, the following bound emerges.

$$\mathbb{E}\left[\bar{e}_k\right] \leq 4\gamma^2(k-T_r)\sum_{t=T_r}^{k-1}\left(n^2(L_0^{\text{imp}})^2+8\mathbb{E}\left[\bar{e}_t\right]+\frac{2B_1^2}{\eta^2}+4B_2^2\mathbb{E}\left[\|\nabla\mathbf{f}^{\eta}(\bar{x}_t)\|^2\right]\right.$$

$$\left.+4B_2^2(L_0^{\text{imp}})^2n^2+\left(\frac{L_{0,x}^{\nabla h}}{\mu_h}\right)^2\frac{4n^2}{\eta^2}(L_{0,y}^f)^2\mathbb{E}[\hat{e}_k]+\frac{4n^2}{\eta^2}(L_{0,y}^f)^2\varepsilon_r\right).$$

(Step 2) Consider (15). From Algorithm 2, we have $\hat{x}_r = x_{i,T_r}$. Thus, we obtain

$$\mathbb{E}\left[\|x_{i,k}-\hat{x}_r\|^2\right] \leq \gamma^2\sum_{t=T_r}^{k-1}\mathbb{E}\left[\left\|g_{i,t}^{\eta,\varepsilon_r}+\nabla_{i,t}^{\eta,d}\right\|^2\right].$$

Thus, for all $T_r+1 \leq k \leq T_{r+1}$ we have

$$\mathbb{E}\left[\hat{e}_k\right] = \frac{1}{m}\sum_{i=1}^{m}\mathbb{E}\left[\|x_{i,k}-\hat{x}_k\|^2\right] = \frac{\gamma^2(k-T_r)}{m}\sum_{t=T_r}^{k-1}\sum_{i=1}^{m}\mathbb{E}\left[\left\|g_{i,t}^{\eta,\varepsilon_r}+\nabla_{i,t}^{\eta,d}\right\|^2\right].$$

Following the steps in (Step 1), we obtain the same bound on $\mathbb{E}\left[\hat{e}_k\right]$.

(ii) The bound in (ii) is obtained by applying Lemma 10 on the inequality in part (i). $\qquad\square$

### B.4 Proof of Theorem 1

*Proof.* (i) From the $L$-smoothness property of the implicit function, where $L = \frac{(L_0^{\text{imp}}n+1)}{\eta}$, we have

$$\mathbf{f}^{\eta}(\bar{x}_{k+1}) \leq \mathbf{f}^{\eta}(\bar{x}_k) + \nabla\mathbf{f}^{\eta}(\bar{x}_k)^T(\bar{x}_{k+1}-\bar{x}_k) + \frac{(L_0^{\text{imp}}n+1)}{2\eta}\|\bar{x}_{k+1}-\bar{x}_k\|^2.$$

In view of the recursion $\bar{x}_{k+1} = \bar{x}_k - \gamma(\bar{g}_k^{\eta,\varepsilon_r}+\bar{\nabla}_k^{\eta,d})$, we have

$$\mathbf{f}^{\eta}(\bar{x}_{k+1}) \leq \mathbf{f}^{\eta}(\bar{x}_k) - \gamma\nabla\mathbf{f}^{\eta}(\bar{x}_k)^T(\bar{g}_k^{\eta,\varepsilon_r}+\bar{\nabla}_k^{\eta,d}) + \frac{(L_0^{\text{imp}}n+1)}{2\eta}\gamma^2\|\bar{g}_k^{\eta,\varepsilon_r}+\bar{\nabla}_k^{\eta,d}\|^2.$$

In view of $\bar{g}_k^{\eta,\varepsilon_r} = \bar{g}_k^{\eta}+\bar{\omega}^{\eta}{}_k+\bar{w}^{\eta}{}_k$, we have

$$\mathbf{f}^{\eta}(\bar{x}_{k+1}) \leq \mathbf{f}^{\eta}(\bar{x}_k) - \gamma\nabla\mathbf{f}^{\eta}(\bar{x}_k)^T(\bar{g}_k^{\eta}+\bar{\nabla}_k^{\eta,d}) - \gamma\nabla\mathbf{f}^{\eta}(\bar{x}_k)^T\bar{\omega}^{\eta}{}_k - \gamma\nabla\mathbf{f}^{\eta}(\bar{x}_k)^T\bar{w}^{\eta}{}_k$$

$$+\frac{(L_0^{\text{imp}}n+1)}{2\eta}\gamma^2\|\bar{g}_k^{\eta}+\bar{\nabla}_k^{\eta,d}+\bar{\omega}^{\eta}{}_k+\bar{w}^{\eta}{}_k\|^2$$

$$\leq \mathbf{f}^{\eta}(\bar{x}_k) - \gamma\nabla\mathbf{f}^{\eta}(\bar{x}_k)^T(\bar{g}_k^{\eta}+\bar{\nabla}_k^{\eta,d}) + \frac{\gamma}{8}\|\nabla\mathbf{f}^{\eta}(\bar{x}_k)\|^2 + 4\gamma\|\bar{\omega}^{\eta}{}_k\|^2 + 4\gamma\|\bar{w}^{\eta}{}_k\|^2$$

$$+\frac{3(L_0^{\text{imp}}n+1)}{2\eta}\gamma^2\left(\|\bar{g}_k^{\eta}+\bar{\nabla}_k^{\eta,d}\|^2+\|\bar{\omega}^{\eta}{}_k\|^2+\|\bar{w}^{\eta}{}_k\|^2\right),$$

where in the second inequality, we utilized the following relation twice: $|u_1^T u_2| \leq 0.5\left(\frac{1}{8}\|u_1\|^2+8\|u_2\|^2\right)$ for any $u_1, u_2 \in \mathbb{R}^n$. Taking expectations on both sides, and requiring that $\gamma \leq \frac{\eta}{24(L_0^{\text{imp}}n+1)}$, we obtain

$$\mathbb{E}\left[\mathbf{f}^{\eta}(\bar{x}_{k+1})\right] \leq \mathbb{E}\left[\mathbf{f}^{\eta}(\bar{x}_k)\right] - \gamma\mathbb{E}\left[\nabla\mathbf{f}^{\eta}(\bar{x}_k)^T(\bar{g}_k^{\eta}+\bar{\nabla}_k^{\eta,d})\right] + \frac{\gamma}{8}\mathbb{E}\left[\|\nabla\mathbf{f}^{\eta}(\bar{x}_k)\|^2\right]$$

$$+\frac{3(L_0^{\text{imp}}n+1)}{2\eta}\gamma^2\mathbb{E}\left[\|\bar{g}_k^{\eta}+\bar{\nabla}_k^{\eta,d}\|^2\right] + 5\gamma\mathbb{E}\left[\|\bar{\omega}^{\eta}{}_k\|^2+\|\bar{w}^{\eta}{}_k\|^2\right]. \qquad (17)$$

From Lemma 9 (vi) and (v), we have

$$-\gamma\mathbb{E}\left[\nabla\mathbf{f}^\eta(\bar{x}_k)^T(\bar{g}_k^\eta + \bar{\nabla}_k^{\eta,d})\right] \le -\tfrac{\gamma}{2}\mathbb{E}\left[\|\nabla\mathbf{f}^\eta(\bar{x}_k)\|^2\right] + \gamma\tfrac{(L_0^{\mathrm{imp}}n+1)^2}{2\eta^2}\mathbb{E}\left[\bar{e}_k\right], \qquad (18)$$

and

$$\mathbb{E}\left[\|\bar{g}_k^\eta + \bar{\nabla}_k^{\eta,d}\|^2\right] \le \tfrac{n^2}{m}(L_0^{\mathrm{imp}})^2 + 2\mathbb{E}\left[\|\nabla\mathbf{f}^\eta(\bar{x}_k)\|^2\right] + \tfrac{2(L_0^{\mathrm{imp}}n+1)^2}{\eta^2}\mathbb{E}\left[\bar{e}_k\right],$$

respectively. In view of $\gamma \le \frac{\eta}{24(L_0^{\mathrm{imp}}n+1)}$, multiplying both sides of the preceding inequality by $\frac{3(L_0^{\mathrm{imp}}n+1)}{2\eta}\gamma^2$, we obtain

$$\tfrac{3(L_0^{\mathrm{imp}}n+1)}{2\eta}\gamma^2\mathbb{E}\left[\|\bar{g}_k^\eta + \bar{\nabla}_k^{\eta,d}\|^2\right] \le \tfrac{3(L_0^{\mathrm{imp}}n+1)n^2}{2m\eta}\gamma^2(L_0^{\mathrm{imp}})^2 + \tfrac{\gamma}{8}\mathbb{E}\left[\|\nabla\mathbf{f}^\eta(\bar{x}_k)\|^2\right]$$
$$+ \gamma\tfrac{(L_0^{\mathrm{imp}}n+1)^2}{8\eta^2}\mathbb{E}\left[\bar{e}_k\right]. \qquad (19)$$

Also, from Lemma 9 (vii) and (viii), we have

$$5\gamma\mathbb{E}\left[\|\bar{\omega}^\eta{}_k\|^2 + \|\bar{w}^\eta{}_k\|^2\right] \le \left(\tfrac{L_{0,x}^{\nabla h}}{\mu_h}\right)^2 \tfrac{20\gamma n^2}{\eta^2}(L_{0,y}^f)^2\left(\mathbb{E}[\hat{e}_k] + \varepsilon_r\right). \qquad (20)$$

From (17)–(20), we obtain

$$\mathbb{E}\left[\mathbf{f}^\eta(\bar{x}_{k+1})\right] \le \mathbb{E}\left[\mathbf{f}^\eta(\bar{x}_k)\right] - \tfrac{\gamma}{4}\mathbb{E}\left[\|\nabla\mathbf{f}^\eta(\bar{x}_k)\|^2\right] + \tfrac{3(L_0^{\mathrm{imp}}n+1)n^2}{2m\eta}\gamma^2(L_0^{\mathrm{imp}})^2$$
$$+ \gamma\tfrac{5(L_0^{\mathrm{imp}}n+1)^2}{8\eta^2}\mathbb{E}\left[\bar{e}_k\right] + \left(\tfrac{L_{0,x}^{\nabla h}}{\mu_h}\right)^2 \tfrac{20\gamma n^2}{\eta^2}(L_{0,y}^f)^2\left(\mathbb{E}[\hat{e}_k] + \varepsilon_r\right).$$

This implies that

$$\mathbb{E}\left[\mathbf{f}^\eta(\bar{x}_{k+1})\right] \le \mathbb{E}\left[\mathbf{f}^\eta(\bar{x}_k)\right] - \tfrac{\gamma}{4}\mathbb{E}\left[\|\nabla\mathbf{f}^\eta(\bar{x}_k)\|^2\right] + \tfrac{\gamma^2\Theta_1}{m}$$
$$+ \gamma\max\{\Theta_2,\Theta_3\}(\mathbb{E}\left[\bar{e}_k\right] + \mathbb{E}[\hat{e}_k]) + \gamma\Theta_3\varepsilon_r.$$

where $\Theta_1 := \frac{3(L_0^{\mathrm{imp}}n+1)n^2}{2\eta}(L_0^{\mathrm{imp}})^2$, $\Theta_2 := \frac{5(L_0^{\mathrm{imp}}n+1)^2}{8\eta^2}$, and $\Theta_3 := \left(\frac{L_{0,x}^{\nabla h}}{\mu_h}\right)^2 \frac{20n^2}{\eta^2}(L_{0,y}^f)^2$. From Lemma 11, for $\gamma \le \frac{\max\{2,\sqrt{0.1\Theta_3}\}^{-1}}{4H}$, we have

$$\mathbb{E}\left[\bar{e}_k\right] + \mathbb{E}\left[\hat{e}_k\right] \le 96B_2^2H\gamma^2\sum_{t=T_r}^{k-1}\mathbb{E}\left[\|\nabla\mathbf{f}^\eta(\bar{x}_t)\|^2\right] + H^2\gamma^2\Theta_4\varepsilon_r + H^2\gamma^2\Theta_5,$$

where $\Theta_4 := \frac{96n^2}{\eta^2}(L_{0,y}^f)^2$ and $\Theta_5 := \frac{48B_1^2}{\eta^2} + (96B_2^2 + 1)(L_0^{\mathrm{imp}})^2n^2$. Since $\gamma \le \frac{\left(\sqrt{48}B_2\max\{\sqrt{\Theta_2},\sqrt{\Theta_3}\}\right)^{-1}}{4H}$, we have $96B_2^2H\gamma^2\max\{\Theta_2,\Theta_3\} \le \frac{1}{8H}$. Combining the two preceding inequalities, we obtain

$$\mathbb{E}\left[\mathbf{f}^\eta(\bar{x}_{k+1})\right] \le \mathbb{E}\left[\mathbf{f}^\eta(\bar{x}_k)\right] - \tfrac{\gamma}{4}\mathbb{E}\left[\|\nabla\mathbf{f}^\eta(\bar{x}_k)\|^2\right] + \tfrac{\gamma}{8H}\sum_{t=T_r}^{k-1}\mathbb{E}\left[\|\nabla\mathbf{f}^\eta(\bar{x}_t)\|^2\right] + \tfrac{\gamma^2\Theta_1}{m}$$
$$+ \gamma\left(H^2\gamma^2\max\{\Theta_2,\Theta_3\}\Theta_4 + \Theta_3\right)\varepsilon_r + H^2\gamma^3\max\{\Theta_2,\Theta_3\}\Theta_5.$$

Summing the preceding relation from $k := 0,\ldots,K := T_R - 1$ for some $R \ge 1$, we have

$$\mathbb{E}\left[\mathbf{f}^\eta(\bar{x}_{T_R})\right] \le \mathbb{E}\left[\mathbf{f}^\eta(\bar{x}_0)\right] - \sum_{k=0}^{T_R-1}\tfrac{\gamma}{4}\mathbb{E}\left[\|\nabla\mathbf{f}^\eta(\bar{x}_k)\|^2\right] + \tfrac{\gamma}{8H}\sum_{k=0}^{T_R-1}\sum_{t=T_r}^{k-1}\mathbb{E}\left[\|\nabla\mathbf{f}^\eta(\bar{x}_t)\|^2\right]$$
$$+ \tfrac{T_R\gamma^2\Theta_1}{m} + \gamma\left(H^2\gamma^2\max\{\Theta_2,\Theta_3\}\Theta_4 + \Theta_3\right)\sum_{r=0}^{R-1}(T_{r+1} - T_r - 1)\varepsilon_r$$
$$+ T_RH^2\gamma^3\max\{\Theta_2,\Theta_3\}\Theta_5.$$

We obtain

$$\sum_{k=0}^{T_R-1}\tfrac{\gamma}{8}\mathbb{E}\left[\|\nabla\mathbf{f}^\eta(\bar{x}_k)\|^2\right] \le \mathbb{E}\left[\mathbf{f}^\eta(\bar{x}_0)\right] - \mathbb{E}\left[\mathbf{f}^\eta(\bar{x}_{T_R})\right] + \tfrac{T_R\gamma^2\Theta_1}{m}$$
$$+ \gamma\left(H^2\gamma^2\max\{\Theta_2,\Theta_3\}\Theta_4 + \Theta_3\right)H\sum_{r=0}^{R-1}\varepsilon_r$$
$$+ T_RH^2\gamma^3\max\{\Theta_2,\Theta_3\}\Theta_5.$$

Multiplying both sides by $\frac{8}{\gamma T_R}$ and using the definition of $k^*$, we obtain

$$\mathbb{E}\left[\|\nabla \mathbf{f}^\eta(\bar{x}_{k^*})\|^2\right] \le 8(\gamma T_R)^{-1}(\mathbb{E}\left[\mathbf{f}^\eta(x_0)\right] - \mathbf{f}^{\eta,*}) + \frac{8\gamma\Theta_1}{m}$$
$$+ 8\left(H^2\gamma^2 \max\{\Theta_2, \Theta_3\}\Theta_4 + \Theta_3\right) H \frac{\sum_{r=0}^{R-1}\varepsilon_r}{T_R}$$
$$+ 8H^2\gamma^2 \max\{\Theta_2, \Theta_3\}\Theta_5.$$

(ii) Substituting $\gamma := \sqrt{\frac{m}{K}}$ and $H := \sqrt[4]{\frac{K}{m^3}}$ in the error bound in (i), we obtain

$$\mathbb{E}\left[\|\nabla \mathbf{f}^\eta(\bar{x}_{k^*})\|^2\right] \le \frac{8(\mathbb{E}[\mathbf{f}^\eta(x_0)]-\mathbf{f}^{\eta,*})}{\sqrt{mK}} + \frac{8\Theta_1}{\sqrt{mK}} + 8\left(\frac{\max\{\Theta_2,\Theta_3\}\Theta_4}{\sqrt{mK}} + \Theta_3\right)\frac{\sum_{r=0}^{R-1}\varepsilon_r}{(mK)^{\frac{3}{4}}} \quad (21)$$
$$+ 8\frac{\max\{\Theta_2,\Theta_3\}\Theta_5}{\sqrt{mK}}.$$

Invoking $\varepsilon_r := \tilde{\mathcal{O}}(\frac{1}{m\tilde{T}_{\tilde{R}_r}})$ where $\tilde{T}_{\tilde{R}_r} := \tilde{\mathcal{O}}\left(m^{-1}(r+1)^{\frac{2}{3}}\right)$, we have $\sum_{r=0}^{R-1}\varepsilon_r = \tilde{\mathcal{O}}(R^{\frac{1}{3}}) = \tilde{\mathcal{O}}\left(\frac{K^{\frac{1}{3}}}{H^{\frac{1}{3}}}\right)$. Substituting $H := \sqrt[4]{\frac{K}{m^3}}$, we obtain $\sum_{r=0}^{R-1}\varepsilon_r = \tilde{\mathcal{O}}\left((mK)^{\frac{1}{4}}\right)$. Substituting this bound in (21), we obtain the iteration complexity.

(iii) From (ii), we obtain $R = \mathcal{O}(\frac{K}{H}) = \mathcal{O}\left(\frac{K}{\sqrt[4]{K/m^3}}\right) = \mathcal{O}\left((mK)^{3/4}\right).$

$\square$

## C Overall Communication Complexity of FedRZO$_{bl}$ in Table 1

Consider FedRZO$_{bl}$ given by Algorithm 2. The overall communication complexity of FedRZO$_{bl}$ depends on what type of FL scheme is employed in step 4 in Algorithm 2. Here, we elaborate on the results summarized in Table 1. These include the following three cases:

**Case 1:** FedRZO$_{bl}$ employs Local SGD [26] with i.i.d. datasets in round $r$ to obtain $\varepsilon_r$-inexact solution to the lower-level problem (**FL**$_{bl}$). In this case, in view of Corollary 1 in [26] under suitable settings and terminating Local SGD after $\tilde{T}_{\tilde{R}_r}$ iterations, we have

$$\mathbb{E}\left[\|\hat{y}_{\varepsilon_r}(x) - y(x)\|^2\right] \le \varepsilon_r := \tilde{\mathcal{O}}\left(\frac{1}{\mu_h^2 m\tilde{T}_{\tilde{R}_r}}\right),$$

in $\tilde{R}_r = m$ round of communications (in the lower-level). Invoking Theorem 1 (iii), the overall communication complexity is $m \times \mathcal{O}\left((mK_\epsilon)^{3/4}\right)$ that is $\mathcal{O}\left(m^{7/4}K_\epsilon^{3/4}\right)$.

**Case 2:** FedRZO$_{bl}$ employs FedAC [54] with i.i.d. datasets in round $r$ to obtain $\varepsilon_r$-inexact solution to the lower-level problem (**FL**$_{bl}$). Similarly, invoking Theorem 3.1 in [54], the communication complexity of the lower-level calls is of the order $m^{1/3}$. Invoking Theorem 1 (iii) again, the overall communication complexity is $m^{1/3} \times \mathcal{O}\left((mK_\epsilon)^{3/4}\right)$ that is $\mathcal{O}\left(m^{13/12}K_\epsilon^{3/4}\right)$.

**Case 3:** FedRZO$_{bl}$ employs LFD [20] with heterogeneous datasets in round $r$ to obtain $\varepsilon_r$-inexact solution to the lower-level problem (**FL**$_{bl}$). In view of Theorem 4.2 [20], the number of local steps in the lower-level calls is of the order $m^{-\frac{1}{3}}\tilde{T}_{\tilde{R}_r}^{\frac{2}{3}}$, which implies the lower-level communication complexity of the order $m^{\frac{1}{3}}\tilde{T}_{\tilde{R}_r}^{\frac{1}{3}}$. Invoking Theorem 1 (iii) and $\tilde{T}_{\tilde{R}_r} := \tilde{\mathcal{O}}\left(m^{-1}(r+1)^{\frac{2}{3}}\right)$, the overall communication complexity is $\sum_{r=0}^{(mK_\epsilon)^{3/4}}\left(m^{\frac{1}{3}}\tilde{T}_{\tilde{R}_r}^{\frac{1}{3}}\right) = \sum_{r=0}^{(mK_\epsilon)^{3/4}}\tilde{\mathcal{O}}\left(r^{2/9}\right) = \tilde{\mathcal{O}}\left((mK_\epsilon)^{11/12}\right).$

**Remark 8.** In Theorem 1, if we take the problem dimension and smoothing parameter into account, we may write the iteration complexity as $K_\epsilon := \tilde{\mathcal{O}}\left(\frac{\frac{n^4}{\eta^8} + \frac{n^6}{\eta^6} + \frac{n^8}{\eta^4}}{m\epsilon^2} + \frac{\frac{n^4}{\eta^4}}{m^{1.8}\epsilon^{0.8}} + \frac{\left(\frac{n}{\eta}\right)^{8/3}}{m^{7/3}\epsilon^{4/3}}\right).$

# D   Additional Experiments

In this section, we provide some additional detail that was omitted in the numerical experiments section.

## D.1   Comparison of Algorithm 1 with other FL methods.

We compare FedRZO$_{nn}$ with other FL methods including FedAvg [34], FedProx [29], FedMSPP [55], and Scaffnew [35] when applied on neural networks with a smooth rectifier. Throughout this experiment, $\eta := 0.01$ for FedRZO$_{nn}$. For the other methods, we use the softplus rectifier defined as $\sigma^\beta(x) := \beta^{-1}\ln(1 + exp(\beta x))$ as a smoothed approximation of the ReLU function. Our goal lies in observing the sensitivity of the methods to $\beta$ when we compare them in terms of the original ReLU loss function. Figure 3 presents the results. We observe that the performance of the standard FL methods is sensitive to the choice of $\beta$. It is also worth noting that the smooth approximation of activation functions, such as softplus, hypertangent, or sigmoid, involves the computation of an exponential term which might render some scaling issues in large-scale datasets.

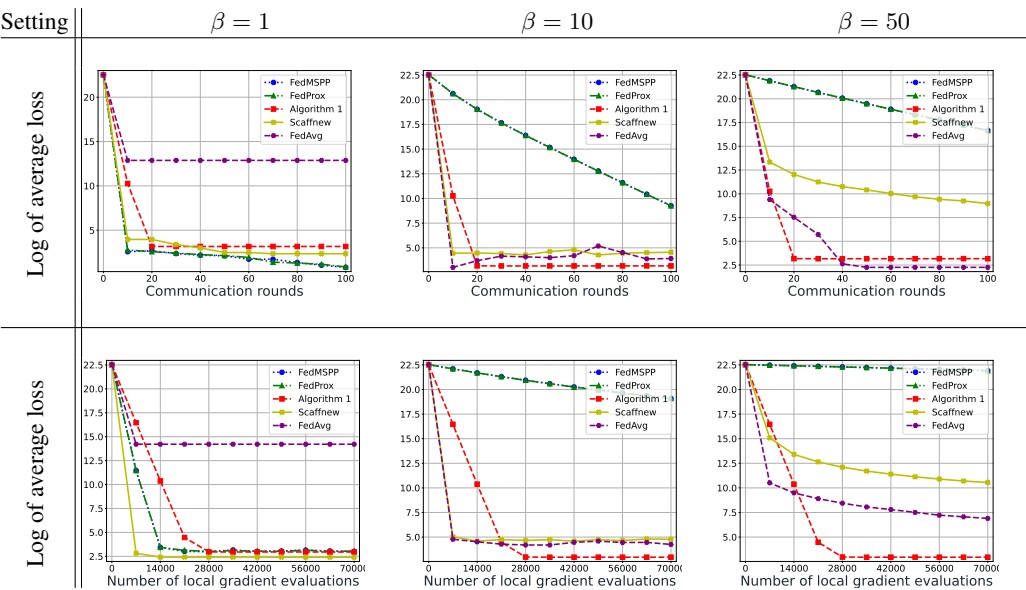

Figure 3: Comparison between FedRZO$_{nn}$ on ReLU NN and standard FL methods when they are implemented on an NN with a smoothed variant of ReLU, characterized by a parameter $\beta$. The performance of the standard FL methods appears to be sensitive to the choice of $\beta$.

## D.2   Additional experiments on Algorithm 1.

We implement Algorithm 1 on the Cifar-10 dataset to test its performance on problems with higher dimensions. We use the same objectives as defined in Section 5.1. We set the number of neurons in the layer to be 4, 20, and 100 in three sets of experiments, receptively. The results are presented in Figure 4.

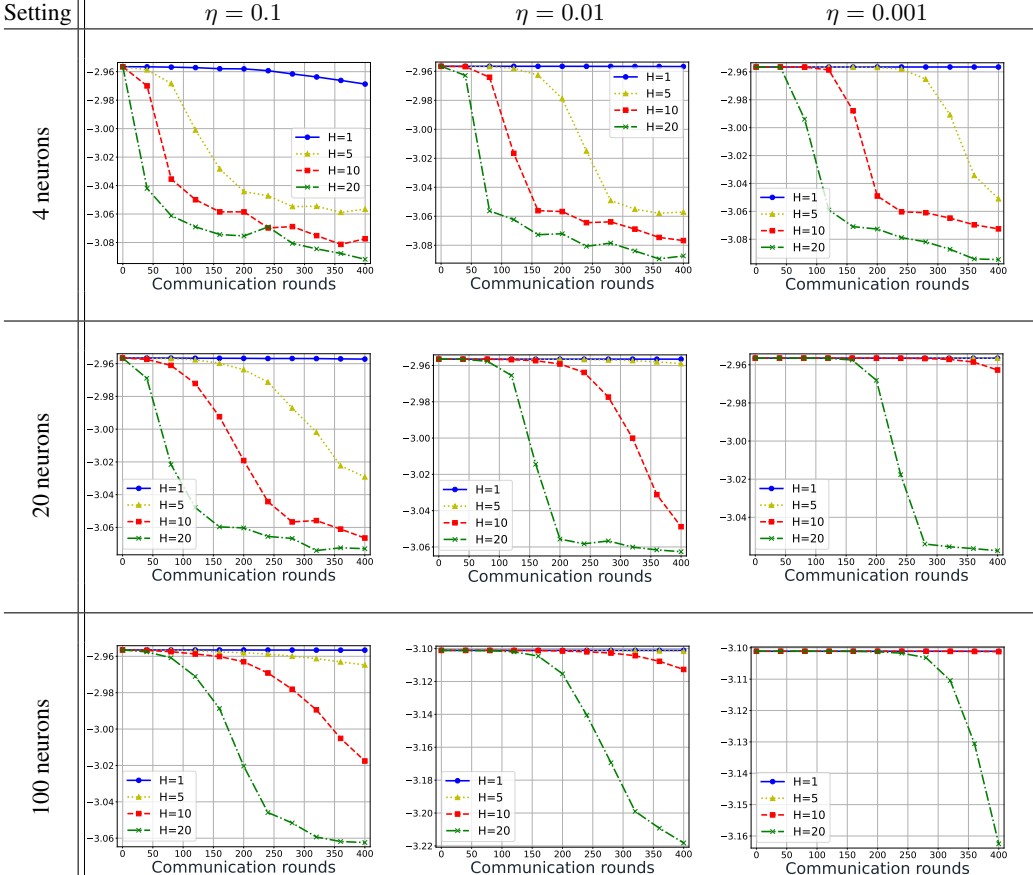

Figure 4: Performance of FedRZO$_{nn}$ on ReLU NN with different number of neurons in the layer and different values of the smoothing parameter $\eta$, using the Cifar-10 dataset. The method performs better with smaller number of neurons and less communication frequency given a certain communication rounds. We also observe that the smaller the smoothing parameter $\eta$, the longer it takes for the method to converge. This is aligned with our theory and analysis.