# OpenReview forum: "Zeroth-Order Methods for Nondifferentiable, Nonconvex, and Hierarchical Federated Optimization"
_NeurIPS.cc/2023/Conference — NeurIPS 2023 poster_

### Official Review · Reviewer_U9HM · 2023-06-20

**Soundness:** 3 good
**Presentation:** 3 good
**Contribution:** 3 good
**Rating:** 7
**Confidence:** 2

**Summary:**


The authors tackle the problem of nondifferentiable nonconvex locally constrained federated learning (Eq. $FL_{nn}$), and the bilevel variant (Eq. $FL_{bl}$). The minimax settings (Eq. $FL_{mm}$) is a special case of the latter.

The authors provide error, iteration and communication complexity for these settings where both non-convexity and non-differentiability is assumed.

**Strengths:**

The reasoning for obtaining the algorithm is clearly explained in Sec. 1.

The experiment section is relevant and provides a clear comparison with other methods.

**Weaknesses:**

Contrary to the reasoning for obtaining the algorithm, the results of Thm. 1 and 2 are not discussed. Doing so would help understand their significance. Typically, do lower bounds exist in some specific settings? What are the interpretation of the different terms? What are the dominant ones?

Of lower importance, the reading is a bit complicated: the content is dense, with lots of technicalities and equation in the main text.

**Questions:**

* Why is the final loss so big in Fig. 2?

**Clarity.**

* A reader not acquainted with Moreau smoothing cannot understand Eq. (1) without further research. Defining $\mathbb B$ and $I_X^\eta$ is advised (or at least point to `l. 158`).
* Likewise, a reference for Eq. (4) is advised (or at least point to `l. 147`).
* Lemma 1 in `l. 161`, Prop. 1 at `l. 171`, Thm. 1 at `l. 182`, Thm. 2 at `l. 209`: refer to the appendix' location of the proof.

**Presentation.**

* Tables 1 and 2 are useful summaries of the result. The font is however smaller than the rest of the text.
* Framed environments `l. 48` and `l. 90` have italic or bold titles. Choose one of them?
* It is a matter of taste, but I would emphasize the text of Definition and Proposition environments so that they stand out of the main text.
* Fig. 1 and 2 require bigger legends (or moved to the caption)
* `l. 270`: `\exp` instead of `exp`

**Limitations:**

Discussing the significance of the results of Thm. 1 and 2 would be helpful.

The comparison with other FL scheme in Table 2 is done with different metric of accuracy. Relating those metrics would provide insight on the theoretical benefits of Alg. 1, and would complements the finding of the experiment section. Notably, highlighting the role of $\eta$ in Table 2 and in Thm 1 and 2 would be welcomed too, as it is an important hyperparameter (complementary to the small remark `l. 263-265`).

---

> ### Author Rebuttal · Authors · 2023-08-10
>
> We thank you for your valuable suggestions and detailed comments on improving this work. Below, please see our response to each comment.
>
> $\textbf{Your comment:}$ Contrary to the reasoning for obtaining the algorithm, the results of Thm. 1 and 2 are not discussed. Doing so would help understand their significance. Typically, do lower bounds exist in some specific settings? What are the interpretation of the different terms? What are the dominant ones?
>
> $\textbf{Our response:}$ Your point is well-taken and we do agree that a remark on Theorem 1 would help with this clarification. Indeed, the communication complexity in Theorem 1, in terms of dependence on the number of clients denoted by $m$ and the parameter $\epsilon$, matches with the existing complexities for addressing smooth nonconvex FL. Importantly, this implies that the use of randomized smoothing does not lead to a degradation of the main complexity bounds in terms of the smoothed problem. Please note that we mention the utility of Theorem 2 in Remark 1. We would like to note that our major contribution in this work lies in the design and analysis of Algorithm 2 in addressing hierarchical FL and so, we attempted to emphasize on this matter in Remark 1.
>
> $\textbf{Your comment:}$ Of lower importance, the reading is a bit complicated: the content is dense, with lots of technicalities and equation in the main text.
>
> $\textbf{Our response:}$  Thank you for raising this issue. We will add more intuition to the  technical details to ease the reading.
>
> $\textbf{Your comment:}$ Why is the final loss so big in Fig. 2?
>
> $\textbf{Our response:}$ We believe this is mainly because of the starting point we choose, we use vectors of all ones.
>
>
> $\textbf{Your comment:}$ A reader not acquainted with Moreau smoothing cannot understand Eq. (1) without further research. Defining $\mathbb B$ and $I_X^\eta$ is advised (or at least point to l. 158). Likewise, a reference for Eq. (4) is advised (or at least point to l. 147). Lemma 1 in l. 161, Prop. 1 at l. 171, Thm. 1 at l. 182, Thm. 2 at l. 209: refer to the appendix' location of the proof.
>
> $\textbf{Our response:}$ Thank you for your suggestions on making the paper more reader-friendly. We will address these by adding some intuitive explanation and point the location of our import lemmas and propositions.
>
> $\textbf{Your comment:}$ Tables 1 and 2 are useful summaries of the result. The font is however smaller than the rest of the text. Framed environments l. 48 and l. 90 have italic or bold titles. Choose one of them? It is a matter of taste, but I would emphasize the text of Definition and Proposition environments so that they stand out of the main text. Fig. 1 and 2 require bigger legends (or moved to the caption). l. 270: $\exp$ instead of exp.
>
> $\textbf{Our response:}$ We agree with your helpful suggestions on the presentation of this paper. We will address these to make the important parts of this paper more clear and consistent.
>
> $\textbf{Your comment:}$ Discussing the significance of the results of Thm. 1 and 2 would be helpful.
>
> $\textbf{Our response:}$  Thank you for your suggestion. We may provide the following note to clarify the significance of the results in our paper and how they benefit the research on FL.
>
> This work appears to be the first paper that provides an FL method with complexity guarantees for solving bilevel optimization problems where the $\textbf{lower-level problem may be constrained.}$ There have been several recent works that have highlighted the challenges in solving this type of hierarchical problems (i.e., Stackelberg games), even in centralized settings. For example, consider $\min_{x \in [-1,1]}\ \max_{y \in [-1,1],\ x+y\leq 0}\ x^2+y$. The solution is $(x^*,y^*)=(0.5,-0.5)$. The same problem, but with a reversed order of min and max, $\max_{y \in [-1,1]}\ \min_{x \in [-1,1], \ x+y\leq 0}\  x^2+y$, has the solution $(x^*,y^*)=(-1,1)$. One of the major contributions in this work is that Algorithm 2 can be employed to address this challenging class of problems complicated with the need for federated learning in both levels. To provide more details on this, see Remark 1 and Table 1 where we have provided explicit communication complexity results for addressing bilevel FL problems under the use of suitable FL methods for the lower level problem with different assumptions. These are only a few instances of the breadth of FL problems that we can provably address using Algorithm 2.
>
> $\textbf{Your comment:}$ The comparison with other FL scheme in Table 2 is done with different metric of accuracy. Relating those metrics would provide insight on the theoretical benefits of Alg. 1, and would complements the finding of the experiment section. Notably, highlighting the role of $\eta$ in Table 2 and in Thm 1 and 2 would be welcomed too, as it is an important hyperparameter (complementary to the small remark l. 263-265).
>
> $\textbf{Our response:}$ Thank you for this important observation. We would like to note that naturally, in view of Lemma 1(iv), the dependence on $\eta$ would be similar to the dependence on inverse of the Lipschitzian parameter in smooth nonconvex FL. However, we can add the dependence on $\eta$ in the convergence rate in Thm. 1 and 2, as the random variable $v$ for zeroth-order method depends on it. As we have shown explicitly in Thm. 1 and 2, the rate depends on $\eta$ and we also validated this in our experiment in sec. 5.1.

---

> > ### Comment · Reviewer_U9HM · 2023-08-18
> >
> > Thank to the authors for answering my questions. I maintain my score.

---

> > > ### Author Response · Authors · 2023-08-19
> > > **Response to Reviewer U9HM**
> > >
> > > We are very grateful to the referee for taking the time in reviewing our work and providing us with detailed and constructive feedback.

---

### Official Review · Reviewer_LNaU · 2023-07-06

**Soundness:** 2 fair
**Presentation:** 2 fair
**Contribution:** 2 fair
**Rating:** 5
**Confidence:** 4

**Summary:**

The paper considers the federated optimization problem with the non-smooth non-convex target function. The authors use a zero-order oracle, which allows to approximate the gradient. This approximation is related not only to the original target function, but also to its smoothed version (an additional theoretical object obtained using the random smoothing technique). For a new smooth target function, there are methods in the literature for solving it, and this is what the authors use. Mini-max problems and bilevel optimization are also considered. Experiments are given.

**Strengths:**

For me, the article is nicely written and easy to follow.

**Weaknesses:**

I ask the authors to pay attention to this work:

Gasnikov, A., Novitskii, A., Novitskii, V., Abdukhakimov, F., Kamzolov, D., Beznosikov, A., ... & Gu, B. (2022). The power of first-order smooth optimization for black-box non-smooth problems. arXiv preprint arXiv:2201.12289.

It gives a more or less unified scheme of how by taking any practical first-order stochastic method for smooth problems to obtain a gradient-free method for non-smooth problems. It seems that using this scheme one can obtain the results of this paper as well. Moreover, this scheme is not new; it has been used for 10 years in most papers on gradient-free optimization.  In particular, the facts that are proven in the article under review are also used in the article I cited and there are links from where they are taken.

Please also note the refs within the article I cited. For example, there are references to works with gradient-free methods of solving saddle point problems.

Finally, the contribution of this paper looks rather technical and at the moment I am not ready to accept the work. But perhaps the authors will change my mind in the process of rebuttal.

**Questions:**

No

**Limitations:**

The paper is theoretical, therefore there is no need to discuss the social negative impact.

---

> ### Author Rebuttal · Authors · 2023-08-09
>
> Thank you for bringing this paper to our attention. We will
> include this work in our references and comment on it in section 2. With respect, this paper has significant differences in scope and treatment with our work. We clarify these in the following.
>
> (i) First, note that this paper studies zeroth-order schemes for $\textbf{nonsmooth convex}$ optimization problems, while we study $\textbf{nonsmooth nonconvex}$ federated stochastic optimization problems. The progression from convex nonsmooth to nonconvex nonsmooth leads to significant challenges and necessitates the introduction of Clarke-stationarity. As we have noted in lines 144--147, our work in designing Algorithm 1 is motivated by recent findings where it is shown that for a subclass of nonsmooth nonconvex functions, computing an $\epsilon$-stationary point is impossible in finite time. In Algorithm 1, we employ randomized smoothing to develop a zeroth-order FL method and then, in Proposition 1, we leverage a relationship between stationarity with respect to the smoothed problem and $\eta$-Clarke stationarity. The result in Proposition 1 is complemented in Theorem 1 where we provide explicit performance guarantees for computing a stationary point to the smoothed problem. To our knowledge, these guarantees did not exist for nondifferentiable nonconvex FL.
>
> (ii) Second, the paper you mentioned does not address hierarchical optimization problems. However, one of the key contributions in our work is the design of Algorithm 2 and its complexity analysis in Theorem 2 for addressing hierarchical problems in FL. To highlight our contributions, consider the (centralized) bilevel minimization of $f(x,y(x))$ with respect to $x$ where $y(x) \in \hbox{arg}\min_{y \in \mathcal{Y}(x)}\ h(x,y)$. First, note that even when $f$ is smooth and convex in $(x,y)$, the implicit function $f(\bullet,y(\bullet))$ is often nondifferentiable nonconvex in $x$. Second, the analytical form of $y(\bullet)$ is unavailable in most ML applications, such as hyperparameter ML. As such, $\textbf{the zeroth-order information of the implicit function $f(\bullet,y(\bullet))$ is not available.}$ In view of this challenge, it is not clear how one can develop a provably convergent zeroth-order method for computing a stationary point to the nonsmooth nonconvex and hierarchical problem $\min_{x}\ f(x,y(x))$. A naive idea is to inexactly compute $y(x)$. However, an inexact computation of $y(x)$ leads to a bias in the zeroth-order information of $f$ and consequently, a bias in the approximation of its zeroth-order gradient. This $\textbf{bias further propagates}$ throughout the implementation (please see Theorem 2(i) where we manage to derive the aggregated bias as $\sum_{r=0}^R{\varepsilon_r}$). Our work precisely addresses this challenge in Theorem 2. Importantly the bound in Theorem 2(i) implies that even when we inexactly compute $y(x)$ using a standard FL scheme, we are able to derive complexity guarantees for solving bilevel FL problems.
>
> Please note that a major technical challenge we faced in designing Algorithm 2 is that $\textbf{inexact evaluations of $y(x)$ must be avoided during the local steps.}$ This is because we consider bilevel problems where both the levels are distributed. Because of this, the inexact evaluation of $y(x)$ by each client in the local step in the upper level would require significant communications and is counter-intuitive to the nature of the FL framework. We carefully address this challenge by introducing delayed inexact computation of $y(x)$. Please see step 8 in Algorithm 2 and note how $y_{\varepsilon}$ is evaluated at $\hat x_r +v_{T_r}$ that is a different vector than the vector used by the client, i.e., $x_{i,k} +v_{T_r}$. At each communication round in the upper level, we only compute $y(x)$ inexactly twice in the global step and then use this $\textbf{delayed information}$ in the local steps. $\textbf{This delayed inexact computation of $y$ renders some technical challenges}$ in the proofs and so, we hope that the design and analysis of Algorithm 2 are not viewed as simple extensions of existing zeroth-order methods.
>
> Lastly, we thank you for considering to reevaluate our work. Although our work may seem technical, we hope to clarify on the significance of the results in our paper and how they benefit the research on FL. Our work appears to be the first paper that provides an FL method with complexity guarantees for solving bilevel optimization problems where the $\textbf{lower-level problem may be constrained.}$ There have been several recent works that have highlighted the challenges in solving this type of hierarchical problems (i.e., Stackelberg games), even in centralized settings. For example, consider $\min_{x \in [-1,1]}\ \max_{y \in [-1,1],\ x+y\leq 0}\ x^2+y$. The solution is $(x^*,y^*)=(0.5,-0.5)$. The same problem, but with a reversed order of min and max, $\max_{y \in [-1,1]}\ \min_{x \in [-1,1], \ x+y\leq 0}\  x^2+y$, has the solution $(x^*,y^*)=(-1,1)$. While we are unaware of zeroth-order methods for solving such problems even in centralized settings, one of the major contributions in our work is that Algorithm 2 can be employed to address this challenging class of problems complicated with the need for federated learning in both levels. Indeed, Algorithm 2 only requires solving the inner level problem for a fixed $x$ inexactly. In the case where the inner level problem is constrained, as it is the case in the above example as well as adversarial problems in FL, we may use a suitable method for solving the lower level problem for a fixed value of $x$ (see Algorithm 2, step 4). To provide more details on this, please see Remark 1 and Table 1 in our paper. We have provided explicit communication complexity results for addressing bilevel FL problems under use of different FL methods for the lower level problem. These are only a few instances of the breadth of FL problems that we can provably address using Algorithm 2.

---

> > ### Comment · Reviewer_LNaU · 2023-08-17
> >
> > Thanks to the authors for the response!
> >
> > (i) I agree that the non-convex case is more complicated, but the scheme for analyzing it is also a unified recipe. Yes, there is no such paper in the literature that clearly describes this scheme. But this scheme has already appeared in other papers in the non-distributed case (in particular, Clarke-stationarity etc).
> >
> > (ii) I agree with the authors here and apologize for not noticing this earlier. It seemed to me to be a part that is not emphasized much (including in the experiments - much less space is devoted to it). Despite the fact that the contribution here looks a bit technical, but for me it is sufficient.
> >
> > I thank the authors again for the rebuttal, and raise my score.

---

> > > ### Author Response · Authors · 2023-08-19
> > > **Response to Reviewer LNaU**
> > >
> > > We thank the referee for taking the time in reviewing our response in detail. We are truly grateful for the reevaluation of our work. Please see our responses in the following.
> > >
> > > (i) With regard to Algorithm 1 and Theorem 1, the referee is making a fair point and we do agree that the randomized smoothing technique has been employed in development of zeroth-order schemes for standard (single-level, non-FL, and mostly differentiable) optimization problems. In Section 3 in lines 154 to160, we have commented on some of the key references related to the zeroth-order and smoothing methods. To address your concern and clarify further, we will add a note in Section 1.2 where we summarize the main contributions and emphasize that the major contribution of our work lies in devising Algorithm 2 and its analysis in Theorem 2 in addressing hierarchical problems in FL.
> > >
> > > (ii) Thank you for this feedback. This is also a valid point. We have three numerical experiments. The first one (Section 5.1) is to validate Algorithm 1 and the other two experiments (Sections 5.2 and 5.3) are to validate Algorithm 2 in addressing a bilevel FL and a minimax FL problem. To address your concern, we will revise the numerical experiments section, reallocate the space between Sections 5.1, 5.2, and 5.3 to provide more details about the two implementations for bilevel and minimax FL problems. This will be done by moving a portion of the discussion in Section 5.1 to the Supplementary Material document.
> > >
> > > We thank the referee once again. Please let us know if you have any further questions or comments.

---

### Official Review · Reviewer_CtdY · 2023-07-07

**Soundness:** 3 good
**Presentation:** 4 excellent
**Contribution:** 3 good
**Rating:** 5
**Confidence:** 4

**Summary:**

**Summary:** The paper develops zero-th order methods for solving non-smooth and non-convex federated problems. In addition, the paper also develops zero-th order methods for solving bilevel and minimax optimization problems in a federated setting. The authors develop a randomized smoothing-based approach and present convergence guarantees for the proposed algorithms. Experiments on training ReLU neural networks and hyperparameter optimization tasks are presented to evaluate the performance of the proposed algorithms.

**Strengths:**

**Strengths:** The paper considers an important problem class of solving non-convex and non-smooth problems with zero-th order oracle. The paper is very well written with ideas explained clearly.

**Weaknesses:**

**Weaknesses:** Here, I list some points the authors should consider addressing:

1.	There is some existing literature on federated learning where randomized smoothing-based zero-th order methods have already been developed. The authors have not mentioned/or compared their approach against such methods Please see [R1], [R2] (strongly convex) below.


     [R1] Fang et al., Communication-Efficient Stochastic Zeroth-Order Optimization for Federated Learning

     [R2] Li et al., Communication-efficient decentralized zeroth-order method on heterogeneous data


2.	The authors should make dependence on the problem dimension explicit in the tables and the communication complexities listed in all the theorems. The current presentation gives an indication to the reader that zero-th order methods can achieve the same guarantees as first-order methods which is certainly not true.

3.	The authors should consider including some discussion on the bounded set dissimilarity since the assumption is relatively new in the context of FL.

4.	Is the notion of Clarke stationarity point related to Goldstein stationary point (or are the two notions the same)? Also, the discussion on Clarke's generalized gradient of the original problem compared to the gradient of the smoothed objective appears before the definition of Clarke's stationarity point. Please consider moving the definition before the discussion.

5.	In Line 172 and Prop. 1 please characterize sufficiently small $\eta$.

6.	Theorem 2 ignores the additional communication complexity because of solving the lower-level problem to $\epsilon_r$ accuracy. The authors should report the complete communication complexity incurred by the algorithm for solving the bilevel optimization problem not only the upper-level complexity.

7.	Finally, the experiments considered to evaluate Algorithm 1 are weak. Specifically, the authors have considered a very low-dimensional problem. It would be advisable to include training on Cifar-10/MINST datasets with practically sized neural networks to evaluate the actual performance of the proposed scheme. Moreover, for hyperparameter learning there are many baseline bilevel algorithms including BSA, TTSA, ALSET, SUSTAIN, stocBiO, etc. whose performance should be compared with the proposed scheme.


**Questions:**

Please see the weakness section above.

---

> ### Author Rebuttal · Authors · 2023-08-10
>
> Thank you for your detailed review. We address the comments point by point as follows.
>
> $\textbf{Response to weakness 1:}$ Note that [R1] considers smooth, but possibly nonconvex, problems while [R2] considers smooth and strongly convex problems. Several distinctions persist. (I) While some FL methods use a zeroth-order framework, they require differentiability or $L$-smoothness of the objective function in the convergence theory. (II) We do agree that these schemes leverage a similar smoothing technique, however, they don't address bilevel problems. One of our key contributions is the design of Algorithm 2 and its complexity analysis in Theorem 2. To clarify, consider the (centralized) bilevel minimization of $f(x,y(x))$ with respect to $x$ where $y(x) \in \hbox{arg}\min_{y \in \mathcal{Y}(x)}\ h(x,y)$. Even when $f$ is smooth and convex in $(x,y)$, the implicit function $f(\bullet,y(\bullet))$ is often nondifferentiable nonconvex in $x$. Also, the analytical form of $y(\bullet)$ is unavailable in most ML applications. As such, $\textbf{the zeroth-order information of the implicit function $f(\bullet,y(\bullet))$ is not available}$ and it is not clear how one can develop a provably convergent zeroth-order method for computing a stationary point to the nonsmooth nonconvex problem $\min_{x}\ f(x,y(x))$. A naive idea is to inexactly compute $y(x)$. However, this leads to a bias in the approximation of the zeroth-order gradient. This $\textbf{bias further propagates}$ throughout the implementation (see Theorem 2(i) where we manage to derive the aggregated bias as $\sum_{r=0}^R{\varepsilon_r}$). Importantly the bound in Theorem 2(i) implies that even when we inexactly compute $y(x)$ using a standard FL scheme, we can derive complexity guarantees for bilevel FL.
>
> A major technical challenge we faced in designing Algorithm 2 is that $\textbf{inexact evaluations of $y(x)$ must be avoided during the local steps.}$ This is because we consider bilevel problems where both the levels are distributed. So, the inexact evaluation of $y(x)$ by each client in the local step in the upper level would require significant communications and is counter-intuitive to the nature of the FL framework. We carefully address this challenge by introducing delayed inexact computation of $y(x)$. See step 8 in Algorithm 2 and note how $y_{\varepsilon}$ is evaluated at $\hat x_r +v_{T_r}$ that is different than $x_{i,k} +v_{T_r}$. $\textbf{This delayed inexact computation of $y$ renders some technical challenges}$ in the proofs and so, we hope that the design and analysis of Algorithm 2 are not viewed as simple extensions of existing zeroth-order methods.
>
> $\textbf{Response to weakness 2:}$ Your point on the dependence on $n$ is well-taken. We can also present the dependence on the smoothing parameter $\eta$ in the complexity results. We will be happy to make this change if the opportunity is provided. However, we would like to emphasize that our major contribution in this work lies in addressing bilevel FL problems in absence of the analytical form of the lower level solution. This problem, even in very low dimensions, has remained challenging in FL.
>
>  $\textbf{Response to weakness 3:}$ Regarding eq (3), as we have noted in the paper, this is an instance of the bounded gradient dissimilarity in SCAFFOLD. Indeed, when the bounded gradient dissimilarity assumption in SCAFFOLD is written for the local functions $0.5\|x-\mathcal{P}_{X_i}(x)\|^2$ we reach to eq. (3). Further, this assumption holds when for example the generated iterate by the algorithm remains bounded. We imposed this assumption to weaken the boundedness assumption.
>
>  $\textbf{Response to weakness 4:}$ Thank you. Goldstein also refers to Clarke stationary points. His contribution lies in showing that if $x$ is stationary with respect to the smoothed problem, then $x$ is $\eta$-Clarke stationary with respect to the original problem.
>
>  $\textbf{Response to weakness 5:}$ Thank you for pointing this out. Our proof for this result has been omitted. In the revised version of our manuscript, we will be happy to make this change and provide the detailed proof.
>
>  $\textbf{Response to weakness 6:}$  We actually did provide the explicit total communication complexity in Table 1 when three different lower-level algorithms are employed. The proof can be found in Appendix C. Also, in Thm. 2, we provide upper-level communication complexity because it is the algorithm for upper-level. Note that in Algorithm 2 line 4, "FedAvg" denotes the suitable federated averaging method for solving the lower level problem.
>
>  $\textbf{Response to weakness 7:}$ We thank the reviewer's suggestion on strengthening our experimental part. We will apply our methods on datasets with higher dimension. These algorithms are not federated and whether they can be applied to nondifferentiable nonconvex setting is not known. Indeed, our work appears to be the first paper that provides an FL method with complexity guarantees for solving bilevel optimization problems where the $\textbf{lower-level problem may be constrained.}$ There have been several recent works that have highlighted the challenges in solving this type of hierarchical problems (i.e., Stackelberg games), even in centralized settings. For example, consider $\min_{x \in [-1,1]}\ \max_{y \in [-1,1],\ x+y\leq 0}\ x^2+y$. The solution is $(x^*,y^*)=(0.5,-0.5)$. The same problem, but with a reversed order of min and max, $\max_{y \in [-1,1]}\ \min_{x \in [-1,1], \ x+y\leq 0}\  x^2+y$, has the solution $(x^*,y^*)=(-1,1)$. One of our major contributions is that Algorithm 2 can be employed to address this challenging class of problems in FL. To elaborate, please see Remark 1 and Table 1 where we provide explicit communication complexity results for addressing bilevel FL  under use of different FL methods for the lower level problem. These are only a few instances of the breadth of FL problems that we can provably address by Algorithm 2.

---

> > ### Comment · Reviewer_CtdY · 2023-08-16
> > **Response to Authors**
> >
> > I thank the authors for their detailed response. Most of my concerns have been addressed. A few general comments that the authors may consider addressing.
> >
> > - I agree with the authors that the contribution is not an extension of previous federated zeroth order algorithms, I just wanted to make sure that the authors include the discussions on these papers in the related work.
> > - Regarding the sufficiently small value of $\eta$, please characterize the value in the proposition's statement as well.
> > -  I believe the experiments on higher dimensional problems of practical interest will significantly strengthen the paper.

---

> > > ### Author Response · Authors · 2023-08-18
> > > **Response to Reviewer CtdY**
> > >
> > > We thank the referee for taking the time to review our response and providing us with further feedback. Please see our responses to your comments in the following.
> > >
> > > $\bullet$ Definitely, we will be happy to add the discussions on the references [R1] and [R2] in the related work in the revised paper.
> > >
> > > $\bullet$ The threshold on $\eta$ is given as $\eta \leq \frac{\delta}{\max (2,nL_0)}$ where $\delta >0$ is an arbitrary scalar to achieve a stationary point w.r.t. the $\delta$-Clarke generalized gradient, $n$ is the dimension, and $L_0$ is the Lipschitz continuity parameter of $f$. To be clear, we provide a revised version of Proposition 1 and a proof sketch as follows.
> > >
> > > $\textbf{Proposition 1}$ Consider problem (4) and let Assumption 1 hold.
> > >
> > >  (i) For any $\eta>0$, we have $\nabla f^{\eta}(x) \in \partial_{2\eta} f(x)$.
> > >
> > >  (ii)  Let $\delta>0$ be given. If $\nabla \mathbf{f}^{\eta}(x) =0$ and $\eta \leq \frac{\delta}{\max (2,nL_0)}$, then $0_n \in \partial_{\delta} \left(  f + \mathbb{I}_X \right)(x).$
> > >
> > > $\textbf{Proof Sketch}$
> > >
> > > (i) The proof for this part follows from Proposition 2.2 in [51].
> > >
> > > (ii) From $\nabla \mathbf{f}^\eta(x) =0$, we have $\nabla f^\eta (x)+\frac{1}{\eta}(x-P_X(x)) =0$. This implies that $\|x-P_X(x)\| \leq \eta \|\nabla f^{\eta}(x)\|$. Next, we obtain a bound as follows.
> > >
> > > $\left\|\nabla f^{\eta}(x) \right\|^2 \leq \tfrac{1}{m}\textstyle\sum_{i=1}^m\left\|\nabla f_i^{\eta}(x) \right\|^2 = \left(\tfrac{n^2}{\eta^2}\right)  \tfrac{1}{m}\textstyle\sum_{i=1}^m\left\|\mathbb{E}_{v_i \in \mathbb{\eta S}}  [  f_i(x+ v_i)  \tfrac{v_i}{\|v_i\|}  ]\right\|^2$
> > >
> > > $ =\left(\tfrac{n^2}{\eta^2}\right)  \tfrac{1}{m}\textstyle\sum_{i=1}^m\left\|\mathbb{E} [(  f_i(x+ v)-f_i(x))  \tfrac{v_i}{\|v_i\|}  ]\right\|^2 $
> > > $\stackrel{\tiny \mbox{Jensen's ineq.}}{\leq}\left(\tfrac{n^2}{\eta^2}\right)  \tfrac{1}{m}\textstyle\sum_{i=1}^m\mathbb{E} [|  f_i(x+ v)-f_i(x)) |^2 ] $
> > >
> > > $\stackrel{\tiny \mbox{Assumption~1 (i)}}{\leq}\left(\tfrac{n^2}{\eta^2}\right)  \tfrac{1}{m}\textstyle\sum_{i=1}^m\mathbb{E} [L_0^2\|v_i\|^2 ] = n^2L_0^2.$
> > >
> > >  Thus, the infeasibility of $x$ is bounded as $\|x-P_X(x)\| \leq \eta nL_0$. Recall that the $\delta$-Clarke generalized gradient of ${\mathbb{I}_X}$ at $x$  is defined as
> > >
> > > $$\partial_\delta \mathbb{I}_X(x) \triangleq \text{conv} ( \zeta: \zeta \in \mathcal{N}_X(y), \|x-y\| \leq \delta ),$$
> > >
> > > where $\mathcal{N}_X$ denotes the normal cone of $X$. In view of $\|x-P_X(x)\| \leq \eta nL_0$, for $y:=P_X(x) $ and $\eta \leq \frac{\delta}{\max\{2,nL_0\}}$, we have $\|x-y\| \leq \delta$. Next we show that for  $\zeta:= \frac{1}{\eta}(x-P_X(x))$ we have $\zeta \in \mathcal{N}_X(y)$. From the projection theorem, we may write
> > >
> > > $$(x-P_X(x))^T(P_X(x)-z) \geq 0, \quad \hbox{for all } z \in X. $$
> > >
> > > This implies that $\zeta^T(y-z) \geq 0$ for all $z \in X$. Note that $y=P_X(x) \in X $. Thus, we have
> > >
> > > $\zeta \in \mathcal{N}_X (y) $
> > >
> > > which implies that $\frac{1}{\eta}(x-P_X(x)) \in \partial_\delta \mathbb{I}(x)$.
> > >
> > > From (i) and that $2\eta \leq \delta$, we have $\nabla f^{\eta}(x) \in \partial_{\delta} f(x)$. Adding the preceding relations and invoking $\nabla \mathbf{f}^{\eta}(x) =0$, we obtain $ 0 \in \partial_{\delta} \left(  f + \mathbb{I}_X \right)(x)$.
> > >
> > > $\bullet$ We thank the referee for the suggestion with regard to implementing the proposed methods on higher dimensional problems of practical interest. Your point is well-taken and we plan on addressing this as follows. (1) By increasing the number of the neurons in our experiments (currently we considered 4 neurons and there are 3140 variables), and (2) By implementing our schemes on the Cifar-10 dataset as you had suggested. We will be happy to incorporate the results in the revised version of the paper.
> > >
> > > Thank you once again and please let us know if you have any further questions or comments.

---

### Official Review · Reviewer_Lumm · 2023-07-07

**Soundness:** 3 good
**Presentation:** 2 fair
**Contribution:** 3 good
**Rating:** 5
**Confidence:** 3

**Summary:**

Existing federated optimization algorithms usually rely on the assumption of differentiability and smoothness, which may fail to hold in practical settings. To this end, this paper employs randomized smoothing approach and zeroth-order optimization techniques for the development of FedRZO algorithm to address this kind of problem. Theoretical analyses on convergence, iteration complexity and communication complexity have also been provided. This paper further extend the idea behind the newly developed algorithm to bilevel and minimax federated optimization problems with sound theoretical guarantees.

**Strengths:**

1. This paper has studied three different federated setting that existing works seldomly consider with both theoretical guarantees and empirical justifications.
2. The combination of randomized smoothing technique with zeroth-order optimization appeals to me, which may inspires the future work on non-smooth zeroth-order optimization.
3. Empirical results are interesting with adequate interpretation.

**Weaknesses:**

1. This paper may give more intuitive explanation or interpretation for its equations to ease reader's understanding. E.g., the authors may need to provide certain motivations for the study of the three different cases in the introduction section instead of directly putting out these cases without any explanation. Besides, the authors may also need to provide intuitive explanation for eq. 3 to help justify the reasonability of this assumption as well as its connection with real-world examples.
2. This paper may need to compare with more federated optimization algorithms to verify the efficacy of their algorithms, e.g., SCAFFOLD, FedZO [R1], from both theoretical and empirical perspectives.

[R1] Fang, Wenzhi, Ziyi Yu, Yuning Jiang, Yuanming Shi, Colin N. Jones, and Yong Zhou. 2022. “Communication-Efficient Stochastic Zeroth-Order Optimization for Federated Learning.”.

**Questions:**

1. What's the difference between line 6 of Algorithm 1 and the gradient estimation (or even the algorithm) in FedZO [R1]? They seem to share similar calculation with only different scale that has been multiplied.
2. Are the baselines in Figure 2 able to access the gradient? If so, shouldn't they converge much faster compared with your zeroth-order optimization algorithm?

**Limitations:**

Limitations haven't been mentioned.

---

> ### Author Rebuttal · Authors · 2023-08-09
>
> Thank you for your helpful suggestions and comments on improving this work.
>
> $\textbf{Response to weakness 1:}$ Thank you for pointing this out. To be clear, the three formulations are all nondifferentiable nonconvex FL problems. The motivation for the three formulations is mentioned in lines 112--120 and 123--126 in section 2. The federated training of ReLU neural networks has coupled nondifferentiability and nonconvexity. In bilevel and minimax FL, the nondifferentiability and nonconvexity come from the implicit objective function. If the opportunity is provided, we will add more intuitive explanations on technical equations to smooth reader's experience.
>
> Regarding eq. (3), as we have noted in the paper, this is an instance of the bounded gradient dissimilarity in SCAFFOLD. Indeed, when the bounded gradient dissimilarity assumption in SCAFFOLD is written for the local functions $0.5\Vert x-\mathcal{P}_{X_i}(x)\Vert^2$ we reach to eq. (3). Further, this assumption holds when for example the generated iterate by the algorithm remains bounded. We imposed this assumption to weaken the boundedness assumption.
>
>
> $\textbf{Response to weakness 2:}$ We would like to point out that both these references assume $L$-smoothness of the local objectives. This is a strong assumption and it often fails to hold when considering bilevel problems. We don't make this assumption and only consider Lipschitz continuity of the objective function. Please note that the progression from smooth nonconvex to nonsmooth nonconvex leads to some challenges and necessitates the introduction of Clarke-stationarity. As we have noted in lines 144--147, our work in designing Algorithm 1 is motivated by recent findings where it is shown that for a subclass of nonsmooth nonconvex functions, computing an $\epsilon$-stationary point is impossible in finite time.
>
> $\textbf{Response to question 1:}$ We do agree that FedZO leverages a similar smoothing technique, however, it does not address bilevel problems. One of the key contributions in our work is the design of Algorithm 2 and its complexity analysis in Theorem 2. To clarify, consider the (centralized) bilevel minimization of $f(x,y(x))$ with respect to $x$ where $y(x) \in \hbox{arg}\min_{y \in \mathcal{Y}(x)}\ h(x,y)$. Even when $f$ is smooth and convex in $(x,y)$, the implicit function $f(\bullet,y(\bullet))$ is often nondifferentiable nonconvex in $x$. Second, the analytical form of $y(\bullet)$ is unavailable in most ML applications. As such, $\textbf{the zeroth-order information of the implicit function $f(\bullet,y(\bullet))$ is not available}$ and it is not clear how one can develop a provably convergent zeroth-order method for computing a stationary point to the nonsmooth nonconvex problem $\min_{x}\ f(x,y(x))$. A naive idea is to inexactly compute $y(x)$. However, an inexact computation of $y(x)$ leads to a bias in the approximation of the zeroth-order gradient. This $\textbf{bias further propagates}$ throughout the implementation (see Theorem 2(i) where we manage to derive the aggregated bias as $\sum_{r=0}^R{\varepsilon_r}$). Importantly the bound in Theorem 2(i) implies that even when we inexactly compute $y(x)$ using a standard FL scheme, we are able to derive complexity guarantees for solving bilevel FL problems.
>
> Please note that a major technical challenge we faced in designing Algorithm 2 is that $\textbf{inexact evaluations of $y(x)$ must be avoided during the local steps.}$ This is because we consider bilevel problems where both the levels are distributed. Because of this, the inexact evaluation of $y(x)$ by each client in the local step in the upper level would require significant communications and is counter-intuitive to the nature of the FL framework. We carefully address this challenge by introducing delayed inexact computation of $y(x)$. See step 8 in Algorithm 2 and note how $y_{\varepsilon}$ is evaluated at $\hat x_r +v_{T_r}$ that is different than $x_{i,k} +v_{T_r}$. $\textbf{This delayed inexact computation of $y$ renders some technical challenges}$ in the proofs and so, we hope that the design and analysis of Algorithm 2 are not viewed as simple extensions of existing zeroth-order methods.
>
> Lastly, we hope to clarify the significance of the results in our paper and how they benefit the research on FL. Our work appears to be the first paper that provides an FL method with complexity guarantees for solving bilevel optimization problems where the $\textbf{lower-level problem may be constrained.}$ There have been several recent works that have highlighted the challenges in solving this type of hierarchical problems (i.e., Stackelberg games), even in centralized settings. For example, consider $\min_{x \in [-1,1]}\ \max_{y \in [-1,1],\ x+y\leq 0}\ x^2+y$. The solution is $(x^*,y^*)=(0.5,-0.5)$. The same problem, but with a reversed order of min and max, $\max_{y \in [-1,1]}\ \min_{x \in [-1,1], \ x+y\leq 0}\  x^2+y$, has the solution $(x^*,y^*)=(-1,1)$. One of the major contributions in our work is that Algorithm 2 can be employed to address this challenging class of problems complicated with the need for federated learning in both levels. To provide more details on this, please see Remark 1 and Table 1 in our paper. We have provided explicit communication complexity results for addressing bilevel FL problems under use of different FL methods for the lower level problem. These are only a few instances of the breadth of FL problems that we can provably address using Algorithm 2.
>
>
> $\textbf{Response to question 2:}$ Thank you for the question. They do have access to the gradient, however they seem to be sensitive to the choice of the parameter $\beta$ in the scaled softplus function (differentiable) defined in line 270, larger $\beta$ means more accurate approximation of the ReLU (nondifferentiable), and ReLU is what we used as the activation function. As shown in Figure 2, our algorithm shows more robustness.

---

### Author Rebuttal · Authors · 2023-08-10

We sincerely appreciate all the reviewers for their time and thoughtful reviews on improving this paper. The following is a summary of major changes that would appear in the revision if accepted.

1. In response to reviewer "Lumm" and "LNaU", we will add the following note to emphasize the significance of our contributions in addressing bilevel FL problems.

This work appears to be the first paper that provides an FL method with complexity guarantees for solving bilevel optimization problems where the $\textbf{lower-level problem may be constrained.}$ There have been several recent works that have highlighted the challenges in solving this type of hierarchical problems (i.e., Stackelberg games), even in centralized settings. For example, consider $\min_{x \in [-1,1]}\ \max_{y \in [-1,1],\ x+y\leq 0}\ x^2+y$. The solution is $(x^*,y^*)=(0.5,-0.5)$. The same problem, but with a reversed order of min and max, $\max_{y \in [-1,1]}\ \min_{x \in [-1,1], \ x+y\leq 0}\  x^2+y$, has the solution $(x^*,y^*)=(-1,1)$. One of the major contributions in this work is that Algorithm 2 can be employed to address this challenging class of problems complicated with the need for federated learning in both levels. To provide more details on this, see Remark 1 and Table 1 where we have provided explicit communication complexity results for addressing bilevel FL problems under the use of suitable FL methods for the lower level problem with different assumptions. These are only a few instances of the breadth of hierarchical FL problems that we can provably address using Algorithm 2.

2. In response to reviewer "Lumm", "CtdY" and "U9HM", we will add more interpretation of technical terms to smooth reader's experience, such as a note on the bounded set dissimilarity (eq. 3) that is an instance of the "bounded gradient dissimilarity" condition, and we will present the definition of Clarke stationary point earlier in section 1.

3. In response to reviewer "Lumm", "CtdY" and "LNaU", we will add a paragraph to discuss and compare other zeroth-order methods in FL  and emphasize that they have made stronger assumptions than in our work.

4. In response to reviewer "CtdY" and "U9HM", we will provide all the complexity results explicitly in terms of both problem dimension $n$ and smoothing parameter $\eta$.

5. In response to reviewer "CtdY", we will apply our method on other datasets such as Cifar-10.

6. In response to reviewer "U9HM", we will add a note to discuss the significance of Thm. 1 and Thm. 2.

Please see the $\textbf{detailed responses}$ to the reviewers.

---

### Decision · Program_Chairs · 2023-09-21

**Decision:**

Accept (poster)

**Comment:**

The paper considers the federated optimization problem with the non-smooth non-convex target function. In general, the reviewers are not against the acceptance of this paper. However, they pointed out that the authors need to carefully take their suggestions and incorporate the discussions into the final version. For example, the reviewer found the contribution of Section 4 interesting and would advise the authors to emphasize Section 4 in terms of experimentation. Please consider all the suggestions in the detailed reviews and from the discussions with the reviewers.